# An atypical Arp2/3 complex is required for *Plasmodium* DNA segregation and malaria transmission

Franziska Hentzschel [1,2,3] ✉, David Jewanski[2,5], Yvonne Sokolowski[2,5], Pratika Agarwal[2], Anna Kraeft [2], Kolja Hildenbrand [2], Lilian P. Dorner [2], Mirko Singer [2], Matthias Marti [1,4,6] ✉ & Friedrich Frischknecht [2,3,6]

*Plasmodium* parasites, the causative agents of malaria, undergo crucial developments within the mosquito vector, initiated by the formation of male and female gametes. Male gametogenesis involves three rapid rounds of mitosis without nuclear or cell division, followed by a single round of DNA segregation and nuclear division during gamete budding. How the cell organizes the segregation of eight genomes from a single octoploid nucleus into eight haploid gametes is currently unknown. Here we discovered an atypical Arp2/3 complex in *Plasmodium* important for DNA segregation during male gametogenesis. Unlike the canonical Arp2/3 complex found in other eukaryotes, *Plasmodium* Arp2/3 localizes to endomitotic spindles and interacts with a kinetochore-associated protein. Disruption of Arp2/3 subunits or actin polymerization interferes with kinetochore–spindle association, causes the formation of subhaploid gametes, and blocks transmission. Our work identified an evolutionary divergent Arp2/3 complex in malaria parasites, provides insights into gametogenesis, and reveals potential targets for transmission-blocking interventions.

*Plasmodium* are evolutionarily distant single-celled apicomplexan parasites that cause malaria and undergo a complex life cycle between vertebrate and mosquito hosts (Fig. 1a). Although transcription factors that regulate differentiation have been identified[1], the molecular mechanisms that drive differentiation processes are not well understood. One reason for these knowledge gaps is the limited sequence conservation of *Plasmodium* proteins[2]. In addition, evolutionarily conserved proteins and protein complexes governing fundamental biological processes are often difficult to identify on the basis of homology. Hence, many proteins are currently marked as absent in *Plasmodium*, such as the spindle assembly checkpoint[3–5] or the Arp2/3 actin nucleator complex[6–9]. *Plasmodium* parasites form gametes that undergo fertilization in the mosquito. Male gametogenesis is a particularly complex process: within just 15 min, a progenitor gametocyte undergoes

three rounds of endomitosis, resulting in the formation of an octoploid nucleus that is then divided as eight flagellated microgametes that emerge simultaneously from the mother cell[10–13]. In the absence of a known canonical spindle assembly checkpoint[4,5,14], it is unclear how the parasite safeguards chromosome segregation during the rapid rounds of endomitosis and gamete formation.

The Arp2/3 complex nucleates actin filaments[6,7,15–18]. It consists of Arp2, Arp3 and five supporting subunits, ARPC1–ARPC5, and is present across the eukaryotic kingdom[15]. Functional studies in model organisms revealed that the Arp2/3 complex mediates lamellipodia formation and endocytic trafficking in the cytoplasm, among other processes[16,17]. Nuclear functions of Arp2/3 include DNA damage repair, nucleation of spindle actin during mitosis and meiosis, and chromosome capture and segregation, but are less well studied and differ

¹Wellcome Centre for Integrative Parasitology, University of Glasgow, Glasgow, UK. ²Integrative Parasitology, Centre for Infectious Diseases, Heidelberg University Medical Faculty, Heidelberg, Germany. ³German Center for Infection Research, DZIF, partner site Heidelberg, Heidelberg, Germany. ⁴VetSuisse and Medical Faculties, University of Zurich, Zurich, Switzerland. ⁵These authors contributed equally: David Jewanski, Yvonne Sokolowski. ⁶These authors jointly supervised this work: Matthias Marti, Friedrich Frischknecht. ✉e-mail: Franziska.hentzschel@med.uni-heidelberg.de; Matthias.marti@uzh.ch

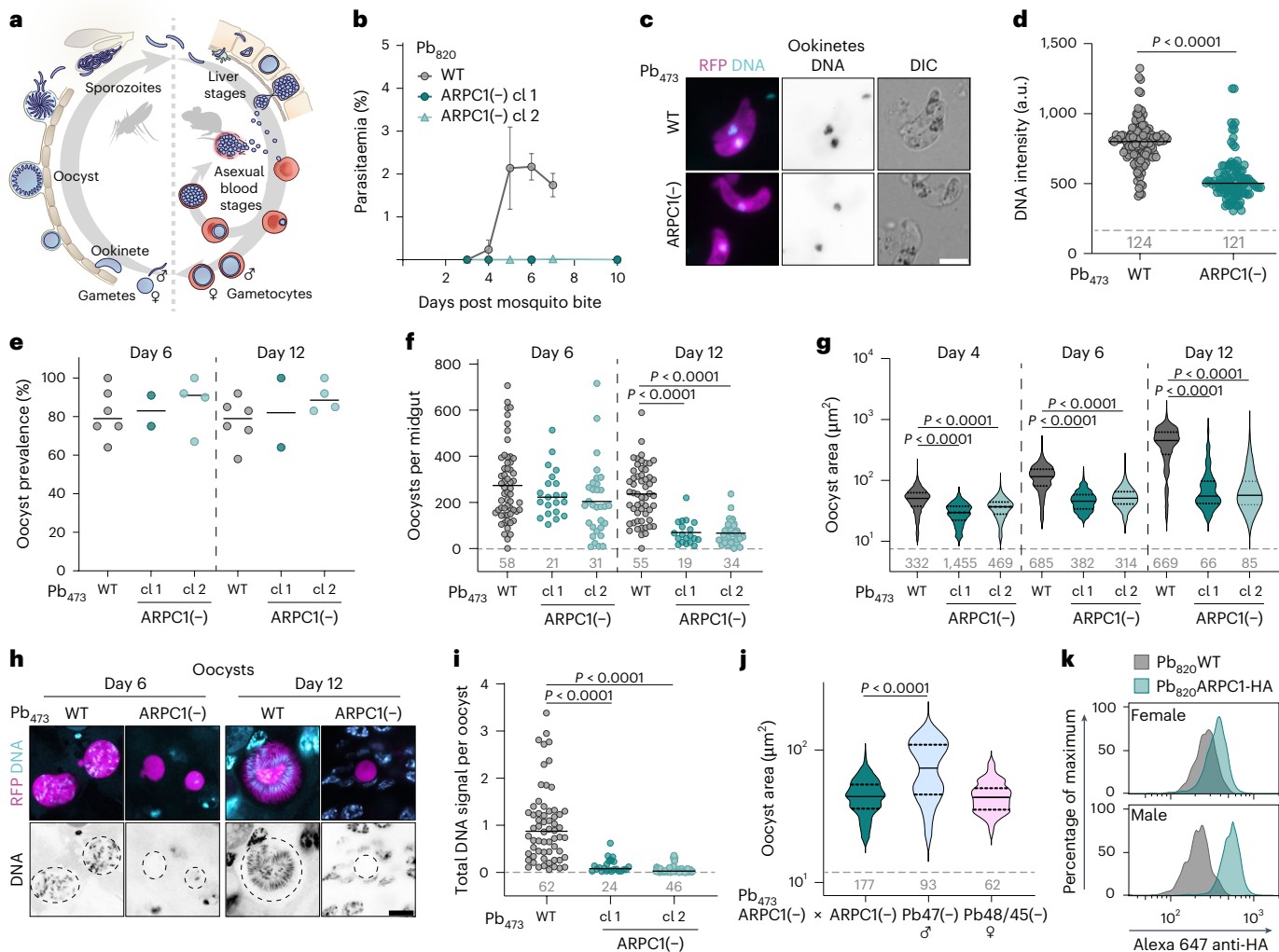

**Fig. 1 | ARPC1 is essential for parasite transmission. a**, Simplified life cycle of *Plasmodium*. **b**, Parasitaemia after natural transmission by mosquito bite. Note that mice bitten by mosquitoes infected with two clones of ARPC1(−) parasites are not infected. WT, wild type. Mean ± s.d. of 4 mice per group. **c**, Ookinete morphology and DNA staining. Scale bar, 5 μm. **d**, DNA intensity of ookinetes. a.u., arbitrary units. **e**, Mosquito infection rate as proportion of midguts carrying oocysts. Line indicates mean. Each data point corresponds to an independent cage feed. **f**, Oocyst numbers per midgut at days 6 and 12 after mosquito infection. **g**, Oocyst area at days 4, 6 and 12 after mosquito infection. **h**, Oocyst morphology and DNA content at days 6 and 12 after mosquito infection. Dashed circles highlight cell circumference. Scale bar, 10 μm. **i**, Total DNA content of oocysts 6 days after mosquito infection, normalized to the mean DNA content of wild-type Pb$_{473}$. **j**, Oocyst area at day 6 after crossing Pb$_{473}$ARPC1(−) with itself, with Pb47(−) (female-deficient) or with Pb48/45(−) (male-deficient) *P. berghei*. **k**, ARPC1–HA signal in female (top) and male (bottom) gametocytes as determined by flow cytometry. Representative histogram of 3 biological replicates. **d,f,i**, Line indicates median. **g,j**, Centre and dotted lines indicate median and quartiles, respectively. Pooled data from at least three (**d,f,g,i**) or two (**j**) independent experiments. Grey numbers above *x* axis indicate total numbers of cells/midguts analysed. **c,h**, Representative images of at least 10 images. Statistics: unpaired two-sided *t*-test (**d**); Kruskal–Wallis Test, Dunn's post test (**f,g,i,j**).

between species[18–25]. Despite the evolutionary conservation of Arp2/3 (refs. [17,26]), it was assumed that the complex has been lost in Apicomplexans, except for a single subunit, annotated as ARPC1/ARC40 in *Plasmodium* (PF3D7_1118800 in *P. falciparum*, PBANKA_0929300 in the rodent malaria parasite *P. berghei*)[8,9]. Here we discovered that *Plasmodium* ARPC1 constitutes part of a divergent Arp2/3 complex, which associates with mitotic spindles in activated male gametocytes and is essential for genome segregation into budding gametes. Disruption of the complex results in a delayed-death phenotype in subsequent oocyst stages leading to a complete transmission block within mosquitoes.

## ARPC1 is essential for male gamete fertility

Phylogenetic studies identified *Plasmodium* ARPC1/ARC40 (named ARPC1 hereafter) as the sole conserved subunit of the Arp2/3 complex

in *Plasmodium*[8,9]. *Plasmodium* ARPC1 is a 41-kDa protein with a predicted WD40 repeat domain with two alpha-helical loops extending from the doughnut-shaped β-propeller[27,28] (Extended Data Fig. 1a). The ARPC1 protein sequence is conserved across *Plasmodium* species but shows less than 20% identity to other ARPC1 proteins (Extended Data Fig. 1b,c). *Plasmodium* ARPC1 has also low identity to a predicted ARPC1 homologue of *Cryptosporidium parvum*, the only other ARPC1 predicted in apicomplexans[8]. Tagging endogenous ARPC1 with green fluorescent protein (GFP) in *P. berghei* (Extended Data Fig. 2a,b) revealed that the protein is predominantly expressed in gametocytes and ookinetes (the motile zygote that forms in the mosquito midgut) and localizes to the nucleus (Extended Data Fig. 2c). A weak ARPC1–GFP signal was also observed in late-stage oocysts. ARPC1–GFP exhibited no phenotypic defects, indicating that the C-terminal tag did not affect the function of ARPC1.

To interrogate ARPC1 function, we generated knockout lines of the *arpc1* gene in two different *P. berghei* reporter lines that facilitate phenotypic characterization either across the life cycle (Pb$_{473}$ARPC1(−)) or specifically within male and female gametocytes (Pb$_{820}$ARPC1(−))[29] (Extended Data Fig. 3). Infection experiments revealed no difference in asexual growth or formation of female and male gametocytes between wild-type and ARPC1(−) lines (Extended Data Fig. 3e–g), but a complete block of transmission from ARPC1(−)-infected mosquitoes to mice, indicating that ARPC1 is essential for parasite development in the mosquito (Fig. 1b and Supplementary Table 1). We next probed the formation and motility of ARPC1(−) ookinetes and found no difference from wild-type ookinetes (Fig. 1c and Extended Data Fig. 3h,i). We noted, however, that the DNA content of ookinete nuclei was reduced by 30% in ARPC1(−) compared with wild-type lines (Fig. 1d). Notably, ookinetes were still infective, as mosquitoes infected with ARPC1(−) had wild-type-like infection rates (that is, prevalence) and oocyst numbers at early development (6 days after infection) (Fig. 1e,f). However, oocyst numbers dropped significantly during later oocyst development (12 days after infection) (Fig. 1f). None of the oocysts produced sporozoites explaining why none of the mice bitten by ARPC1(−)-infected mosquitoes became infected (Supplementary Table 1). Imaging oocysts at 4, 6 and 12 days after infection showed ARPC1(−) oocysts to be significantly smaller than wild-type oocysts at all time points (Fig. 1g,h) and to contain less DNA than wild-type oocysts (Fig. 1h,i). Complementing ARPC1(−) by reintroducing the *arpc1* gene into the same locus fully restored oocyst size, sporulation and parasite transmission, confirming that the phenotype is caused by the absence of ARPC1 (Supplementary Fig. 1 and Supplementary Table 1). In conclusion, we found that ARPC1 is required for normal oocyst growth and sporozoite development, and deletion of ARPC1 leads to a complete block in transmission.

In the mosquito, *Plasmodium* parasites undergo sexual replication and many gene functions essential for mosquito-stage development are provided by one sex only[10,30–32]. To investigate whether ARPC1 function is sex specific, we crossed ARPC1(−) with either Pb47(−) parasites that do not produce fertile females[32] or Pb48/45(−) parasites that do not produce fertile males[31]. Only crossing with female-deficient Pb47(−) restored oocyst size at 6 days after infection, while oocysts of the cross with male-deficient Pb48/45(−) parasites remained small (Fig. 1j). ARPC1 is thus required for male, but not female fertility, in line with a recent genetic screen for fertility traits in *Plasmodium*[33]. To test for sex-specific expression of ARPC1, we tagged ARPC1 with a haemagglutinin (HA) in the gametocyte wild-type reporter line Pb$_{820}$ (Extended Data Fig. 4a,b). Both red fluorescent protein (RFP)-positive female and GFP-positive male gametocytes expressed ARPC1–HA in the nucleus (Extended Data Fig. 4c), confirming the original observation with the ARPC1–GFP line (Extended Data Fig. 2c). However, we detected a stronger ARPC1–HA signal in males compared with females (Fig. 1k and Extended Data Fig. 4c–e), indicating a higher protein expression in the male lineage, in line with the male-specific phenotype of ARPC1(−). As the ARPC1–HA signal observed in females was not significantly higher than that in the wild-type background (Extended Data Fig. 4e) and crossing ARPC1(−) with a female-deficient line restored the phenotype (Fig. 1j), it is likely that the residual expression of ARPC1 in females reflects the shared developmental pathway with males rather than a specific biological function.

## ARPC1 is needed for proper DNA segregation into male gametes

During male gametogenesis in the mosquito midgut, the DNA is replicated three times by endomitosis and in parallel, eight axonemes are formed in the cytoplasm. Subsequently, eight flagellated microgametes bud off from the parental cell (Fig. 2a). Imaging ARPC1–GFP during gametogenesis revealed that ARPC1 relocalizes from the nucleoplasm to an area surrounding the mitotic spindle at 3 min post activation (mpa) (Fig. 2b, full figure in Supplementary Fig. 2). ARPC1 then follows the spindle dynamics, localizing to two spindles at 7–8 mpa and to four

spindles at 12 mpa. At 15 mpa, ARPC1 localized either at eight distinct foci surrounding the DNA, or within the residual body surrounded by either eight condensed DNA foci or the emerging flagella. To investigate whether spindle formation, DNA replication or DNA condensation are impaired in ARPC1(−), we next imaged ARPC1(−) gametocytes at 3 mpa and 15 mpa. At 3 mpa, both wild-type and ARPC1(−) gametocytes formed a spindle (Fig. 2c). At 15 mpa, ARPC1(−) gametocytes, similar to wild-type gametocytes, had formed axonemes and replicated their DNA (Fig. 2d). However, we observed differences in DNA localization. In wild-type male gametocytes, around 50% of the population contained an enlarged nucleus with a homogeneous DNA distribution, 30% showed DNA condensation into eight foci, and 20% of cells were exflagellating (gametes beating their flagella and leaving the mother cell), with DNA localizing to the budding flagella (Fig. 2d,e). In the few remaining cells, DNA was localized in an irregular pattern, probably representing an intermediate step in the process of DNA segregation. By contrast, only few ARPC1(−) gametocytes contained condensed DNA foci while the majority exhibited an irregular DNA pattern (Fig. 2d,e). In exflagellating ARPC1(−) gametocytes, a large proportion of the DNA was retained in the residual body and only a part of the DNA localized to the flagella. Altogether, our data suggest that ARPC1 is not required for spindle or gamete formation but for correct DNA condensation right before the flagellated gamete starts budding off.

To investigate whether the change in DNA localization also results in defects in DNA segregation into the eight progeny cells, we quantified the DNA content of activated male gametocytes and free male gametes (Fig. 2f–h). DNA signal was normalized to that of female gametocytes/gametes on the same slide. Only around 30–60% of all microgametes contained any detectable DNA signal, supporting previous observations that DNA segregation into male gametes is an error-prone process[34,35] (Fig. 2g). Activated male gametocytes of both wild-type and ARPC1(−) lines had the expected octoploid (8C) DNA content (Fig. 2h). DNA-positive wild-type microgametes had an average DNA content of 1C, indicating that DNA segregation during exflagellation is an all-or-nothing process. The proportion of DNA-positive microgametes did not differ between ARPC1(−) and wild-type lines, but the DNA content was reduced to an average of 0.4C in those ARPC1(−) gametes that contained DNA (Fig. 2g,h). We therefore conclude that deletion of *arpc1* results in subhaploid microgametes, suggesting that ARPC1 is required for proper DNA segregation into developing male gametes.

In the mosquito midgut, male gametes fertilize females, forming a zygote that directly undergoes DNA replication and meiosis to further develop into the ookinete[11] (Supplementary Fig. 3a). Accordingly, in wild-type parasites, the DNA content roughly doubles from non-activated gametocytes to zygotes at 1 h after activation and then further increases to 4C at 4 h after activation (after meiosis) to remain at that level in ookinetes (Supplementary Fig. 3b). In ARPC1(−) parasites, the C value only increased to 1.2C after fertilization, consistent with the decreased DNA content contributed by the male gamete. Nevertheless, the DNA content approximately doubled to around 2.8C at 4 h after activation and remained at this level in ookinetes. To further investigate whether nuclear architecture or meiosis are affected by the deletion of *arpc1*, we imaged wild-type and ARPC1(−) ookinetes by electron tomography and reconstructed their nuclei in 3D (Supplementary Fig. 3c). In both wild-type and ARPC1(−) lines, four centrosomes nucleating hemispindles were present, demonstrating that duplication of the centrosomes and spindle formation has occurred in ARPC1(−) ookinetes despite the decreased DNA content. Thus, ARPC1-deficient, subhaploid gametes are still fertile and ARPC1(−) parasites can undergo meiosis and ookinete formation.

## ARPC1 is part of a functional atypical Arp2/3 complex

As WD40 domains are known to mediate protein–protein interactions, we performed immunoprecipitation of ARPC1–GFP from purified

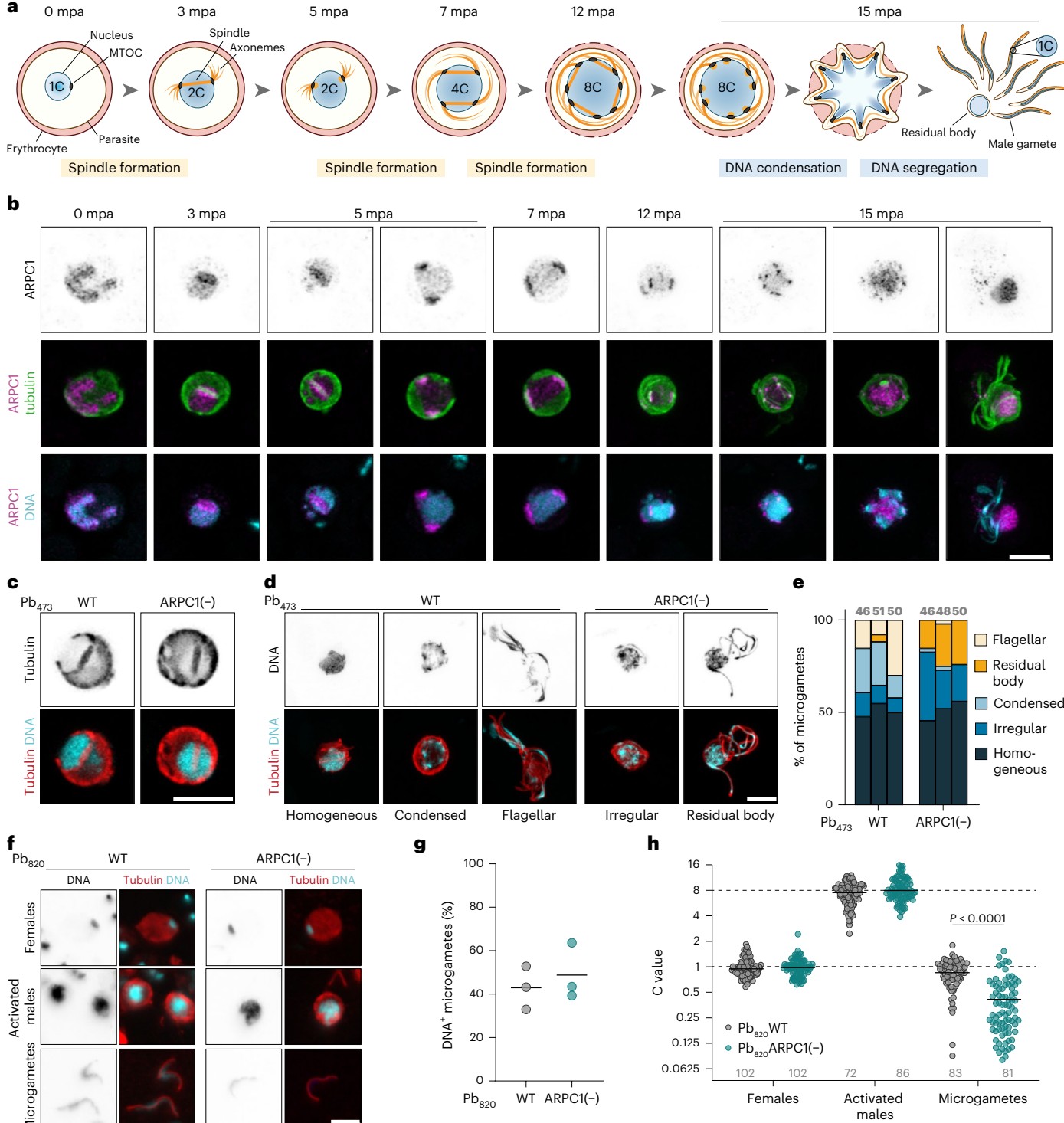

**Fig. 2 | ARPC1 mediates DNA segregation into male gametes. a**, Scheme of male gametogenesis. Key events of the three rounds of mitosis are indicated. **b**, ARPC1–GFP localization during gametogenesis. 0–7 mpa, single slice; 12–15 mpa, maximum *Z* projection. **c**, Spindle formation in activated male gametes at 3 mpa. **d**, Different patterns of DNA localization in activated male gametocytes at 15 mpa. **e**, Quantification of DNA localization observed in activated male gametocytes at 15 mpa. Each bar represents an individual biological replicate. Total numbers of investigated cells indicated above bar. **f**, Images of female gametocytes, activated male gametocytes and microgametes

at 20 mpa. **g**, Relative abundance of microgametes with detectable DNA signal. Line indicates mean. Each dot represents one independent experiment. **h**, DNA content of females, activated males and microgametes, normalized to the mean DNA content of females imaged on the same slide. Dotted lines, expected C value of activated males (8C) and microgametes (1C). Pooled data from three independent experiments. Grey numbers indicate total numbers of cells analysed. **b–d,f**, Representative images from at least 5–10 images. Scale bars, 5 μm. Statistics: Kruskal–Wallis test, Dunn's post test (**h**).

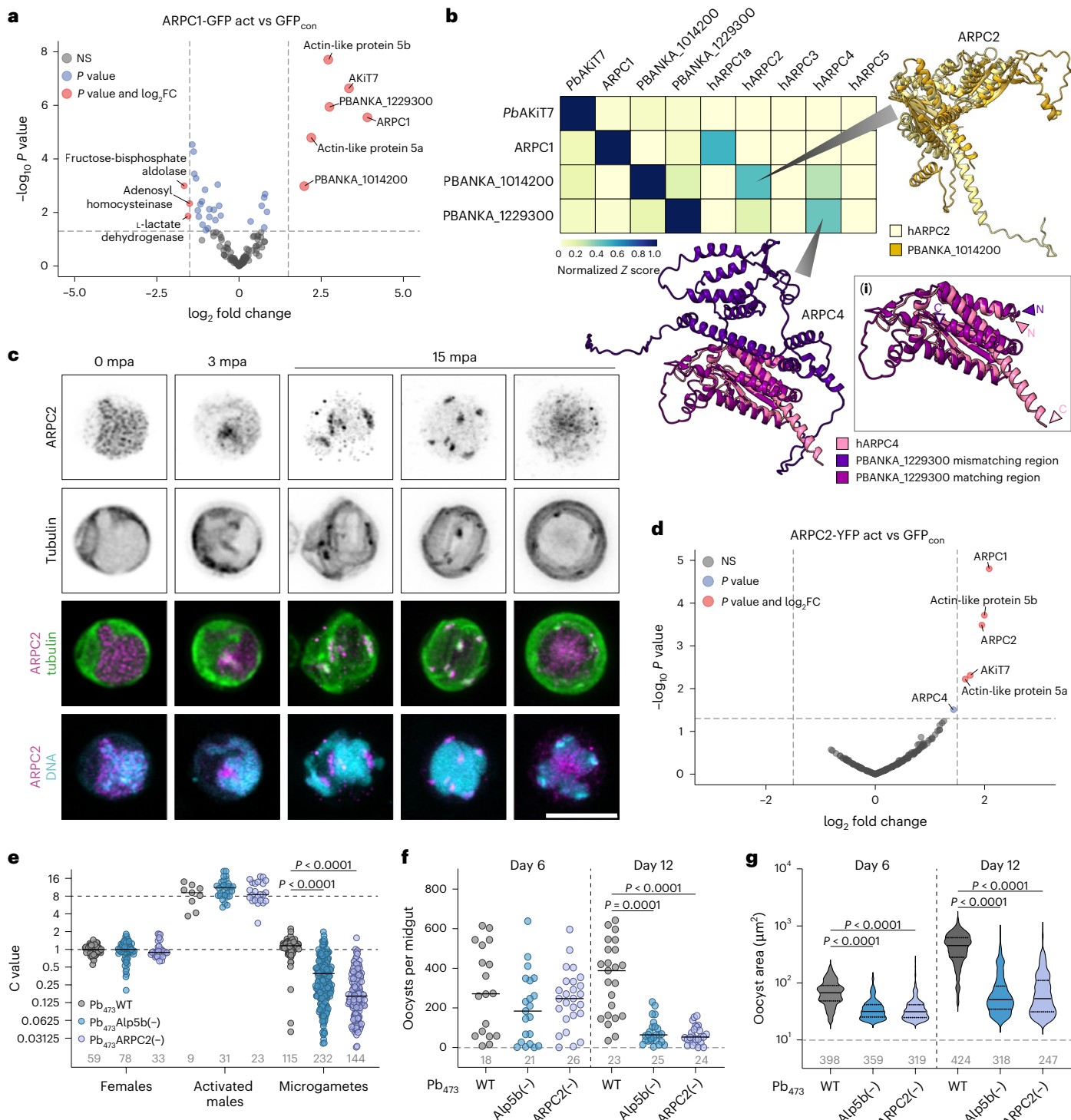

**Fig. 3 | ARPC1 constitutes part of an atypical Arp2/3 complex. a**, Enriched proteins after co-immunoprecipitation of ARPC1–GFP versus GFP_con from activated gametocytes followed by mass spectrometry. NS, not significant. **b**, Structural comparisons of *Plasmodium* ARPC1 interaction partners to human ARPC subunits using DALI[40]. *Z* scores are normalized per row to the *Z* score of self-comparison to account for the fact that different protein sizes result in different maximal *Z* scores. Structure predictions of PBANKA_1014200 overlaid with hARPC2 and PBANKA_1229300 overlaid with hARPC4 are shown in yellow and pink/violet, respectively. Inset (**i**) shows a close-up view of the C-terminal region of PBANKA_1229300. **c**, Localization of ARPC2–YFP in non-activated and activated gametocytes. 0–3 mpa, single slice; 15 mpa, maximum *Z* projection. Scale bar, 5 μm. Representative images from at least five images. **d**, Enriched proteins after co-immunoprecipitation of ARPC1–GFP versus GFP_con from activated gametocytes followed by mass spectrometry. **e**, DNA content of females, activated males and microgametes of Pb_473Alp5b(−) and Pb_473ARPC2(−). Values normalized to the mean DNA content of females imaged on the same slide. Dotted lines, expected C value of activated males (8) and microgametes (1). **f**, Oocyst numbers per midgut at days 6 and 12 after mosquito infection with Pb_473Alp5b(−) and Pb_473ARPC2(−). **g**, Oocyst area at days 6 and 12 after mosquito infection with Pb_473Alp5b(−) and Pb_473ARPC2(−). Centre and dotted lines indicate median and quartiles, respectively. **e**–**g**, Pooled data from at least two independent experiments. Grey numbers indicate total numbers of cells/midguts analysed. Statistics: Kruskal–Wallis test, Dunn's post test.

non-activated and activated gametocytes to identify interaction partners. *Pb*GFP_con gametocytes that express nucleocytosolic GFP served as control[36]. In pulldowns from non-activated ARPC1–GFP gametocytes, we only identified ARPC1 itself (Extended Data Fig. 5a). In activated gametocytes, we found five additional proteins enriched in ARPC1–GFP compared with *Pb*GFP_con (Fig. 3a and Supplementary Table 2). These proteins included two actin-like proteins, annotated as Alp5a (PBANKA_0811800) and Alp5b (PBANKA_1007500), two proteins of unknown function (PBANKA_1014200 and PBANKA_1229300) and the apicomplexan-specific kinetochore protein 7 (AKiT7, PBANKA_0612300)[37]. All the identified proteins are conserved across *Plasmodium* and are specifically expressed in male gametocytes according to a single-cell RNA-seq resource[38] (Extended Data Fig. 5b). Both Alp5b and AKiT7 were recently reported to be required for male fertility, whereas there are no data available for the other pulldown hits[33].

Structure prediction of all identified proteins using AlphaFold 3 (ref. [39]) suggests that Alp5a and Alp5b are structurally similar to human ARP3 and ARP2, respectively (Extended Data Fig. 6a,b). We also compared the predicted structures of ARPC1, the two *Plasmodium* proteins of unknown function and AKiT7 to those of the human (h) ARPC subunits using the DALI algorithm[40]. We found structural conservation between *Plasmodium* ARPC1 and hARPC1, as suggested by previous annotations[8]. Moreover, our analysis also revealed structural conservation between PBANKA_1014200 and hARPC2 (as already identified in a recent study for the *P. falciparum* homologue[2]) and between PBANKA_1229300 and hARPC4 (Fig. 3b and Extended Data Fig. 6c–e). Notably, sequence alignment and superimposition of PBANKA_1229300 with hARPC4 revealed that the *Plasmodium* protein contains an additional domain. Whereas its C-terminal domain is predicted with high confidence by AlphaFold 3 (Extended Data Fig. 6f) and aligns very well with the structure of hARPC4, the large N-terminal domain of ~40 kDa in size is predicted with very low confidence and consists of disordered loop regions with little secondary structure (Fig. 3b). Including ARPC1 itself, we thus identified structural homologues to five out of seven subunits of the Arp2/3 complex, including the core proteins Arp2 and Arp3.

To test whether the identified hits are valid ARPC1 interaction partners, we first tagged *P. berghei* ARPC2 (PBANKA_1014200) by integrating YFP into a non-conserved internal loop (Supplementary Fig. 4a,b). Imaging Pb_473ARPC2–YFPint revealed gametocyte-specific expression and localization mirroring that of ARPC1 in space and time (Fig. 3c). A pulldown of ARPC2–YFP from non-activated and activated gametocytes yielded the same set of interaction partners in activated gametocytes as observed for ARPC1 (Fig. 3d, Supplementary Fig. 4c and Supplementary Table 3). Deletion of either ARPC2 or Alp5b (*Plasmodium* Arp2) in wild-type Pb_473 (Supplementary Fig. 5a–d) did not impact blood-stage growth or gametocyte production, but it resulted in a significant decrease in the DNA content of male gametes and in the formation of small oocysts that arrested early in development (Fig. 3e–g, Supplementary Fig. 5e,f and Supplementary Table 1). Together, these findings demonstrate that ARPC1 interaction partners phenocopy ARPC1, suggesting that these proteins interact in a *Plasmodium* Arp2/3 complex to facilitate male DNA segregation.

In the canonical Arp2/3 complex, Arp2 and Arp3 serve as nucleation sites for the new actin filament, while ARPC1–5 form a scaffold to interact with nucleation-promoting factors and the actin mother filament[41,42]. To investigate whether the putative *Plasmodium* Arp2/3 subunits could arrange in a similar complex, we predicted the complex formed by the five *Plasmodium* subunits together with different numbers of actin 2 monomers using AlphaFold 3 (ref. [39]). The arrangement of subunits varied depending on the number of actin monomers used in the prediction (Extended Data Fig. 7a). Structure predictions had the highest prediction score with three to five actin monomers, coinciding with a subunit arrangement similar to the experimentally solved structure of the bovine Arp2/3 complex in the branch junction[41]

(Fig. 4a–c and Extended Data Fig. 7b–d). The modelling predicted a direct interaction between one actin subunit each and Alp5a and Alp5b, respectively, closely resembling the interaction of bovine Arp2 and Arp3 with actin[43,44] and suggesting a role for *Plasmodium* Arp2/3 in actin polymerization. Given the predicted structural conservation of the *Plasmodium* Arp2/3 complex, we hypothesized that it could be inhibited by the known Arp2/3 inhibitors CK-666 or CK-869 (ref. [45]). However, upon drug treatment of activated gametocytes, we observed a different phenotype: CK-666 and CK-869 both impaired overall exflagellation rates in a titratable manner compared with the non-active control CK-689, and the few gametes that formed often lacked DNA entirely (Extended Data Fig. 7e). Nevertheless, those microgametes that were DNA positive contained a complete genome, indicating that neither CK-666 nor CK-869 did affect DNA segregation itself (Fig. 4d and Extended Data Fig. 7f). The phenotypic difference between drug treatment and knockout of Arp2/3 subunits suggests that CK-666 and CK-869 do not inhibit the *Plasmodium* Arp2/3 complex itself, possibly because the binding site for these drugs in the *Plasmodium* complex is not conserved. The impact of CK-666 and CK-869 on exflagellation may instead be an off-target effect.

To test whether actin filaments co-localize with the *Plasmodium* Arp2/3 complex in vivo, we expressed an actin-filament binding chromobody fused to GFPemerald (CBE)[46,47] together with mCherry-tagged ARPC1 in *P. berghei* (Extended Data Fig. 8a,b). Live imaging showed dynamic co-localization of ARPC1 and F-actin throughout all three rounds of endomitosis in male gametocytes (Fig. 4e,f and Extended Data Fig. 8c). Treatment of activated gametocytes with two inhibitors of actin polymerization: cytochalasin D which is active in *Plasmodium* and impacts gliding motility of *Plasmodium* ookinetes and sporozoites, and latrunculin B which does not interfere with *Plasmodium* actin dynamics[47], revealed that cytochalasin D but not latrunculin B treatment led to the formation of subhaploid gametes, while DNA replication in male gametocytes and overall proportion of DNA-positive gametes were not affected (Fig. 4g and Extended Data Fig. 8d). Together, these data fit a model in which actin polymerization is facilitated by a *Plasmodium* Arp2/3 complex to mediate genome segregation.

The only non-Arp2/3 complex protein found in the interactome of ARPC1 and ARPC2 was AKiT7, previously described as being kinetochore associated[37] (Fig. 3a,d). Endogenous tagging of AKiT7 with YFP showed that AKiT7–YFP was distributed around the nucleoplasm, with some enrichment at the spindles during male gametogenesis, as previously observed[37] (Extended Data Fig. 9a–c). Importantly, AKiT7–YFP-expressing parasites did not show a phenotype during male genome segregation (Extended Data Fig. 9d). Immunoprecipitation of AKiT7–YFP from activated gametocytes revealed the five putative Arp2/3 subunits among the most significant hits (Extended Data Fig. 9e,f). In addition, many nuclear proteins that are highly expressed in male gametocytes were significantly enriched, such as proteins of the DNA replication machinery and chromatin organization, reflecting the more pan-nuclear localization of AKiT7 (Supplementary Table 4). Among the interaction partners was also actin 2, further supporting a link between *Plasmodium* Arp2/3 and this actin isoform. Curiously, we did not find any of the previously described kinetochore proteins[37] interacting with AKiT7 in our conditions. Thus, the AKiT7 interactome further supports the formation of an Arp2/3–AKiT7 complex, yet it suggests that there is no or no strong direct link between Arp2/3–AKiT7 and the core kinetochore.

## *Plasmodium* Arp2/3 safeguards kinetochore–spindle attachment

We lastly tested whether the Arp2/3 complex affects the kinetochore–spindle connection by tagging the core kinetochore protein NDC80 with YFP in wild-type and ARPC1(−) parasites and imaging NDC80 localization during gametogenesis (Extended Data Fig. 10a,b). In wild-type parasites, NDC80 localized exclusively to the spindles at all time points,

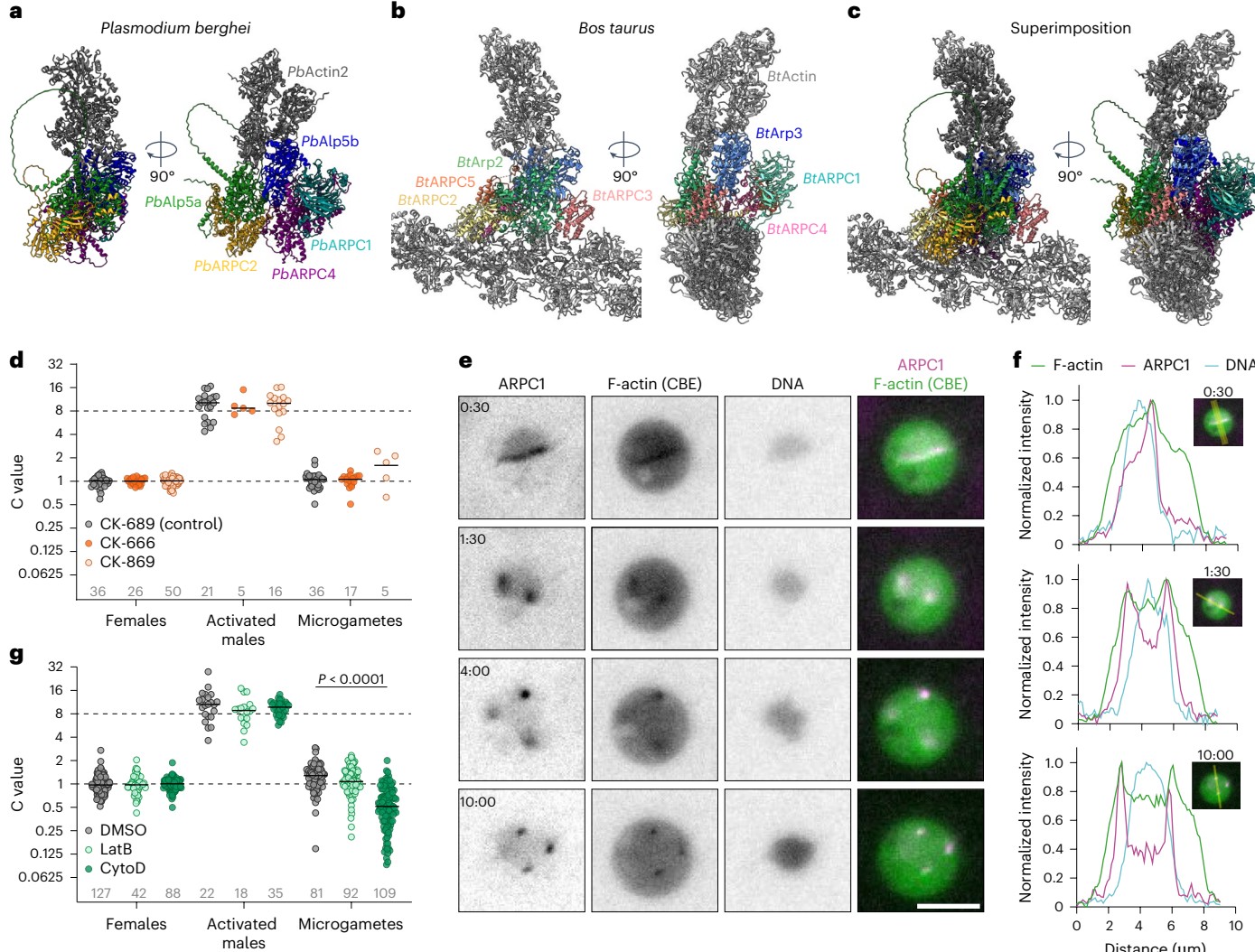

**Fig. 4 | *Plasmodium* Arp2/3 co-localizes with F-actin and actin polymerization is essential for male DNA segregation. a**, First rank of AlphaFold 3 multimer prediction of *Pb*Alp5a, *Pb*Alp5b, *Pb*ARPC1, *Pb*ARPC2, *Pb*ARPC4 and three monomers of *Pb*Act2. **b**, Experimentally determined structure of the bovine (*Bos taurus*) Arp2/3 complex in the branch junction[41]. **c**, Superimposition of *Pb*Arp2/3 complex structure prediction and *Bt*Arp2/3 structure. **a**–**c**, Additional side views provided in Extended Data Fig. 7b–d. **d**, DNA content of females, activated males and microgametes in the presence of 100 μM CK-689 (inactive control), CK-666 or CK-869. **e**, Live-cell imaging of activated *Pb*ARPC1–mCherry/ CBE male gametocyte. Time points in minutes after start of the movie indicated in top left corner. Scale bar, 5 μm. Representative images from at least five

movies. **f**, Intensity profile plots of cross-sections of selected images of the time series shown in **e**. Intensity was normalized to minimal and maximal value per channel. Insets show images and line used for generating the profile plots. **g**, DNA content of females, activated males and microgametes in the presence of dimethylsulfoxide (DMSO; solvent control), 1 μM latrunculin B (LatB) or 1 μM cytochalasin D (CytoD). **d**,**g**, Values normalized to the mean DNA content of females imaged on the same slide. Dotted lines, expected C value of activated males (8) and microgametes (1). Pooled data from at least two independent experiments. Grey numbers above *x* axis indicate total numbers of cells analysed. Statistics: Kruskal–Wallis test, Dunn's post test.

in line with published data[48] (Extended Data Fig. 10c,d and Supplementary Fig. 6a). At 15 mpa, NDC80 localized either to the spindle or to the distinct foci of condensed DNA before each one is pulled towards an emerging flagellum (Fig. 5a–c and Supplementary Fig. 6a). In ARPC1(−), NDC80 also localized to the first spindle at 3 mpa, but this association was partially lost in subsequent mitosis steps, resulting in the appearance of free NDC80 foci at 7 mpa and 12 mpa (Extended Data Fig. 10c,d and Supplementary Fig. 6a). At 15 mpa, NDC80 was only partially associated with the spindles with additional free foci or dispersed in a speckled pattern throughout the nucleus (Fig. 5a–c and Supplementary Fig. 6a). We observed the same dispersed NDC80 signal when treating wild-type gametocytes with cytochalasin D, which also resulted in aberrant DNA condensation and segregation during exflagellation (Fig. 5d,e and Supplementary Fig. 6b). While the *Plasmodium* Arp2/3 complex and actin

polymerization are thus not required for the initial kinetochore–spindle attachment, they seem to be important for maintaining this connection during the subsequent rounds of rapid endomitosis.

## Discussion

In this study, we present the discovery of an atypical *Plasmodium* Arp2/3 complex required for maintaining kinetochore–spindle attachment during male gametogenesis and consequently correct DNA segregation into emerging gametes (Fig. 6). The complex localizes to the spindles during male endomitosis and remains in the residual body upon gamete budding. In the absence of individual Arp2/3 complex subunits, male gametes are subhaploid. Although they can still fertilize females to form zygotes and motile ookinetes, these mutant parasites arrest at the oocyst stage, leading to a complete block in transmission.

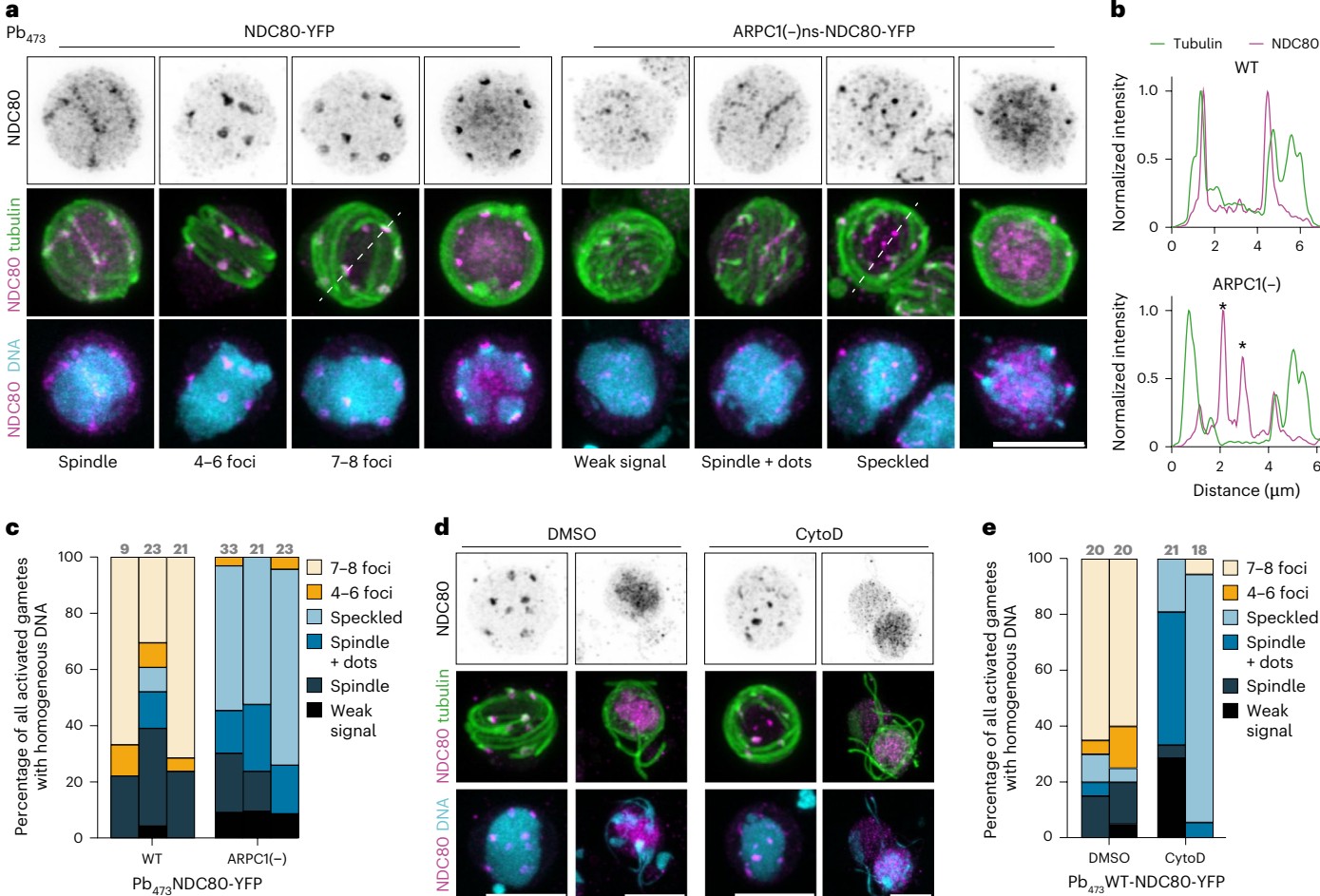

**Fig. 5 | Arp2/3 and actin are required for maintaining kinetochore attachment to the spindle. a**, Localization of NDC80–YFP in activated male wild-type Pb$_{473}$ or Pb$_{473}$ARPC1(−) gametocytes at 15 mpa. White dashed lines indicate cross-sections used for intensity profile plots in **b**. **b**, Intensity profile plots of cross-sections of selected images of **a**. Intensity was normalized to minimal and maximal value per channel. Asterisks indicate NDC80 foci not attached to a tubulin spindle. **c**, Quantification of observed NDC80–YFP localization patterns in **a**.

**d**, Localization of NDC80–YFP in activated male wild-type Pb$_{473}$ gametocytes at 15 mpa exposed to either solvent (DMSO) or 1 μM CytoD. **e**, Quantification of observed NDC80–YFP localization patterns in **d**. **a,d**, Images represent maximum Z projections. Scale bars, 5 μm. Representative images from at least five images per time point. **c,e**, Each bar represents an independent experiment. Total numbers of investigated cells indicated above bar.

Arp2/3 subunits co-localize with F-actin, and inhibition of actin polymerization during gamete formation phenocopies their deficiency. These findings are consistent with a model in which *Plasmodium* Arp2/3 functions as an actin nucleator, but direct biochemical evidence is still required to support this hypothesis. It also remains to be investigated whether the observed actin filaments are linear or branched. *Plasmodium* encodes for two actin isoforms—ubiquitously expressed actin 1, and actin 2, which is expressed in mosquito stages only. Actin 2 is essential for male gametogenesis, localizes both to the cytoplasm and the nuclear spindle of the male gametocyte and in its absence, gametocytes do not exflagellate[43,49]. Although complementation of actin 2 with actin 1 restores gamete formation and exflagellation, parasites still arrest in early oocysts, suggesting that actin 1 complements cytoplasmic, but not nuclear functions of actin 2 (ref. 50). We thus hypothesize that Arp2/3 facilitates polymerization of actin 2 along the spindle during male gametogenesis. Similar to Arp2/3, actin 2 is only found in the genus *Plasmodium*, suggesting a specialized function of this actin isoform together with the *Plasmodium* Arp2/3 complex.

In the absence of Alp5b (*Plasmodium* Arp2), ARPC1 or ARPC2, parasites form male gametes containing only half of the genome. Our data suggest that this phenotype is caused by the partial detachment of NDC80-positive kinetochores from the spindles during the second

and third round of mitosis, while the initial attachment of kinetochores to the spindle is not affected. During *Plasmodium* male gametogenesis, kinetochores remain attached to the spindle throughout three consecutive, rapid rounds of mitosis[34,48]. Arp2/3-mediated actin polymerization might be required to stabilize the attachment of kinetochores to the spindle against the mechanical forces occurring during mitosis (Fig. 6). In the absence of Arp2/3, the pulling forces from the retracting spindles may cause the dissociation of individual kinetochores, leading to the retention of a subset of chromosomes in the residual body upon gamete emergence (Fig. 6). The localization of Arp2/3 subunits and F-actin to the mitotic spindle and the disrupted kinetochore localization in exflagellating ARPC1(−) or CytoD-treated parasites suggest that Arp2/3 and F-actin act near the kinetochore–spindle interphase. Yet, we did not find evidence for a direct protein–protein interaction with core kinetochore components.

Recent studies have implicated the Arp2/3 complex in the mitosis of metazoan cells, where Arp2/3 localizes to the centrosome to nucleate actin filaments permeating the mitotic spindle[20–22,25]. Inhibition of Arp2/3 perturbed centrosomal microtubule organization and impaired mitotic spindle formation and chromosome congression, leading to chromosomal segregation defects[20,21,23,25]. Spindle actin is also required for correct chromosome segregation during meiosis[51]. In contrast to

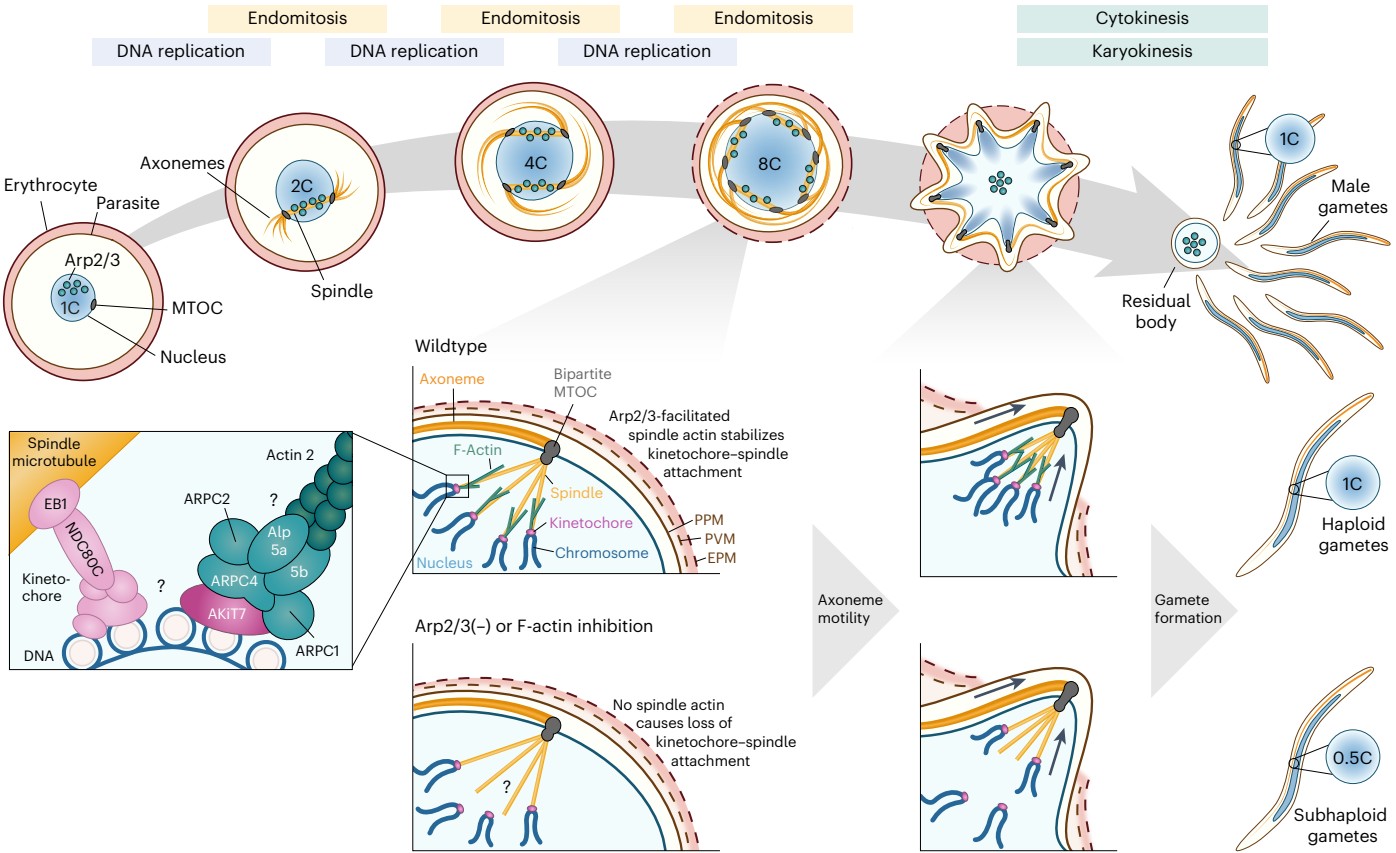

**Fig. 6 | Proposed model of the *Plasmodium* Arp2/3 complex during DNA replication in male gametes.** The Arp2/3 complex localizes to the spindles during the three rounds of endomitosis and then remains in the residual body upon gamete budding. The complex interacts with AKiT7 and localizes near the spindle, possibly nucleating actin 2 filaments that localize along the spindle. During the three rapid rounds of mitosis, this spindle actin may stabilize the kinetochore–spindle attachment, resulting in the formation of haploid gametes. In the absence of Arp2/3 or after inhibition of actin polymerization, no spindle

actin is nucleated. The forces mediated by the repeated shortening of the spindle possibly lead to a loss of kinetochore–spindle attachment for a subset of chromosomes, resulting in the formation of subhaploid gametes. Question marks indicate open questions regarding the precise link between Arp2/3 and the kinetochore and whether Arp2/3 indeed nucleates spindle actin. EPM, erythrocyte plasma membrane; MTOC, microtubule-organizing centre; PVM, parasitophorous vacuolar membrane; PPM, parasite plasma membrane.

metazoan Arp2/3, *Plasmodium* Arp2/3 localizes directly near the spindle itself. Even though the specific mechanisms vary, a role for Arp2/3 and F-actin in chromosome segregation during mitosis thus appears to be a common theme across the eukaryotic kingdom. Of note, in animal cells, the Arp2/3 complex is also implicated in DNA damage repair and safeguarding replication forks during replicative stress[18,19,52]. While our current data favour a role in kinetochore–spindle attachment, it cannot be excluded that *Plasmodium* Arp2/3 is also important to safeguard DNA integrity during the three rapid rounds of DNA replication.

A core protein connecting the kinetochores to the spindle during *Plasmodium* male gametogenesis is end-binding protein 1 (EB1)[34,53,54]. Deletion of EB1 in *P. berghei* resembles the phenotype of ARPC1(−), with an arrest in early oocysts[54]. Interestingly, deletion of EB1 in *P. falciparum* and *P. yoelli* leads to the formation of gametes that lack DNA entirely and are incapable of fertilizing females, highlighting species-specific differences[34,53]. A paternal phenotype followed by an early oocyst arrest has also been described for *Plasmodium* parasites deficient in MISFIT, the microtubule motor kinesin-8X (ref. 55), the aurora kinase Ark2 (ref. 54), the SUN domain protein SUN1 (ref. 33), and for a parasite line that expresses actin 1 in place of actin 2 (ref. 50). Notably, MISFIT is a formin-like protein[56], and in other systems, Arp2/3 and formins are known to work in concert to regulate the formation of actin networks[57,58]. Also, immunoprecipitation of Ark2 and EB1 in *P. berghei* revealed many interacting proteins including AKiT7 but not a single subunit of the Arp2/3 complex[54], suggesting additional AKiT7

functions and interactions including as a linker of the Arp2/3 complex to other complexes.

Although for the above-mentioned proteins a mechanistic link with the Arp2/3 complex or Arp2/3-mediated actin polymerization remains to be determined, a unifying feature of these paternal defects is that they do not affect parasite development until the oocyst stage. This delayed-death-like developmental arrest remains puzzling. Arp2/3-deficient oocysts are equipped with a complete genome provided by the female gamete, regardless of the nature of the missing male DNA. It is possible that the aneuploidy of the oocyst leads to activation of a post-meiotic replication checkpoint that causes a cell cycle arrest. Alternatively, absence of the male genome could cause gene dosage effects that affect oocyst development. A set of paternally imprinted genes could also be expressed from the paternal genome only, as described in some plants where paternal expression of specific genes is essential after fertilization[59].

Phylogenomic analyses across apicomplexan parasites for actin-related proteins and the apparent lack of branched actin in *Plasmodium* previously led to the prevailing assumption that the Arp2/3 complex had been lost in this phylum[8,9]. While the canonical Arp2/3 complex consists of seven subunits, we have so far only identified five subunits. We did not find evidence for putative *Plasmodium* ARPC3 and ARPC5 subunit orthologues in any of our three pulldowns or by structure-based search. Their existence cannot be excluded, as they may exhibit weaker interactions with the complex and thus be

undetectable in our experimental conditions and/or be too divergent for identification by structure-based search. Instead of ARPC3 and ARPC5, we consistently identified AKiT7 as interaction partner, suggesting the formation of a hybrid complex.

Canonical Arp2/3 complexes are recruited and activated by nucleation-promoting factors (NPFs) that either belong to the WASP or the WISH/DIP1/SPIN90 family[17]. However, we could not identify any orthologues to these proteins in the *Plasmodium* genomes[60]. The mode of *Plasmodium* Arp2/3 activation thus remains to be determined, although it is tempting to speculate that AKiT7 could act as an activator. Intriguingly, *Plasmodium* ARPC4 contains a long, N-terminal domain that does not align to any Arp2/3 subunit. This extension may compensate for the possibly missing Arp2/3 subunits ARPC3 or ARPC5, or it may directly connect the complex to the spindle and compensate for the apparent lack of NPFs. The discovery of *Plasmodium* Arp2/3 raises the intriguing possibility that additional aberrant Arp2/3 complexes may be present in other apicomplexan lineages. Indeed, an ARPC1 orthologue has been annotated in *Cryptosporidium*[8], and we found ARPC2 orthologues by structure search[61] in *Toxoplasma*, *Eimeria* and *Neospora*. It is therefore likely that the Arp2/3 complex has assumed specialized functions in Apicomplexa that resulted in major sequence divergence while structural features remained conserved.

In conclusion, we present the discovery and characterization of a divergent *Plasmodium* Arp2/3 complex that is essential for malaria parasite transmission to the mosquito. The study sheds light on the unconventional cell division in the male gamete of a major eukaryotic pathogen. While further work is required to elucidate the precise mode of action and evolution of this atypical Arp2/3 complex, interfering with its function could provide a building block to break the vicious cycle of *Plasmodium* transmission.

## Methods

### Ethics statement and mice
All experiments were performed in accordance with GV-SOLAS and FELASA guidelines and have been approved by German authorities (Regierungspräsidium Karlsruhe), or according to the guidelines defined by the Home Office and UK Animals (Scientific Procedures) Act 1986 and approved by the UK Home Office (project license P6CA91811) and the University of Glasgow animal welfare and ethical review body. Female Theiler Original (TO) mice weighing 25–30 g were purchased from Envigo and were aged 5–8 weeks at the time point of infection. Female Swiss mice weighing 20–25 g were purchased from JANVIER and were aged 5–8 weeks at the time point of infection. Female C57BL/6 mice weighing 18–20 g were purchased from Charles River Laboratories. For each experiment, mice were age matched and were allocated randomly to each group. Mice were kept in groups of 2–4 mice per cage under specific pathogen-free (SPF) conditions within the animal facilities at Heidelberg University or the University of Glasgow on a 12 h light/dark cycle at 22 °C (±2 °C) and 50–60% humidity with ad libitum access to food and water.

### General maintenance of parasites and mosquitoes
The following *P. berghei* lines were used in this study as parental lines to generate transgenic parasites: wild-type Pb, *P. berghei* ANKA; wild-type Pb$_{473}$, a *P. berghei* ANKA line expressing RFP from the *hsp70* promoter (gift from K. Huges and A. Waters); and wild-type Pb$_{820}$, a *P. berghei* ANKA line expressing GFP from a male promoter and RFP from a female reporter (gift from K. Huges and A. Waters[29]). *P. berghei* parasite infections were initiated by intraperitoneal (i.p.) injection of cryostabilates (100 µl parasitic blood and 200 µl freezing solution (Alsever solution + 10% glycerol)). Parasitaemia was monitored by Giemsa-stained blood smears. Mice were generally killed by cardiac puncture after isoflurane- or ketamine/xylazine-induced full anaesthesia. Mosquitoes (*Anopheles stephensi*) were reared and maintained according to standard procedures.

### Generation of transgenic parasites
**Cloning and preparation of plasmids for transfection.** Cloning was performed according to standard procedures using enzymes purchased from NEB, unless stated otherwise. Gibson assembly was performed using either the Infusion kit (Takara Bio) or the NEB HiFi Assembly kit (NEB). All primers are listed in Supplementary Table 5. All final plasmids were checked by sequencing before transfection.

To tag ARPC1 with GFP by single crossover, a targeting vector was cloned by amplifying the 3′ region of *arpc1* using primers P1/P2 and cloning it via Gibson assembly into the EcoRI/BamHI-digested plasmid pL18 (ref. [62]). The final plasmid pL18-ARPC1–GFP was linearized using NheI for transfection into wild-type Pb.

To tag ARPC1 by double crossover with a triple HA-tag, the cloning strategy involved three steps. First, the GFP cassette was removed from the vector pBAT-SIL6 (ref. [63]) by digesting with PvuII/EcoRI and relegation of the backbone to obtain pBAT-SIL6-mCherry. Then, 5′ and 3′ homology regions of the *arpc1* locus were PCR amplified with primers P3/P4 or P5/P6, respectively, and cloned via Gibson assembly into pBAT-SIL6-mCherry using SpeI/XbaI and AvrII/HindIII, respectively, to generate pBAT-SIL6-ARPC1-mCherry. Finally, this vector was digested with PmlI/HpaI, and a 3×HA sequence, generated by overlap extension PCR using the primers P7/P8/P9/P10, was inserted using Gibson assembly. The final plasmid pBAT-SIL6-ARPC1-HA-tag was linearized using KpnI/XbaI and transfected into wild-type Pb$_{820}$.

To tag ARPC2 internally with an sYFP2, pBAT-SIL6-mCherry was digested with PmlI/NaeI. The 3′ region of *arpc2* was amplified from genomic (g)DNA in two fragments using primers P13/14 and P15/16, and the *yfp* gene was amplified from the plasmid pSYFP2-C1 (gift from D. Gadella (Addgene, plasmid 22878)) using primers P17/18. Fragments were assembled via Gibson assembly to generate pBAT-SIL6-ARPC2-YFPint. The final plasmid was linearized using EcoRI to transfect into wild-type Pb$_{473}$.

To generate a line expressing mCherry-tagged ARPC1 and the chromobody-GFPemerald under the *actin 1*-promoter, the vector pBAT-SIL6-ARPC1-mCherry was cut with SacI and ligated with the chromobody-GFPemerald expression cassette, which was amplified using primers P19/20 from a previously published vector[47]. The vector was linearized with XbaI/HindIII before transfection.

To tag AKiT7, the C-terminal region of *akit7* was amplified using primers P60/61 from gDNA, and *yfp* was amplified from pSYFP2-C1 using primers P62/63. pBAT-SIL6-mCherry was digested with PmlI/NaeI. Fragments were assembled via Gibson assembly to generate pBAT-SIL6-AKiT7-YFP. The plasmid was linearized using BbsI to transfect into wild-type Pb. Owing to an oversight in cloning, the plasmid led to a frameshift at the C terminus of AKiT7, changing the sequence from RSILFKI to GVYFSKY. However, as the resulting parasite showed no phenotype, we concluded that this did not affect parasite life cycle progression.

To delete *arpc1*, the targeting vector PbGEM-290656 was obtained from the PlasmoGEM resource[64]. The vector was linearized using NotI for transfection into wild-type Pb$_{820}$ or Pb$_{473}$.

To complement the ARPC1(−) line, the *arpc1* gene was amplified from gDNA using primers P11/P12 and cloned into the BamHI/XbaI-digested vector pBAT-SIL6 to obtain pBAT-SIL6-ARPC1. To complement with an HA-tagged ARPC1, the *arpc1* gene was then excised again by SacII/XhoI digestion and cloned into the equally digested pBAT-SIL6-ARPC1-HA-tag plasmid to obtain the plasmid pBAT-SIL6-ARPC1-HAcomp. For transfection, the vector was digested with SacII/KpnI.

To delete *alp5b*, 5′ and 3′ homology regions were amplified from gDNA using primers P21/22 and P23/24, respectively. To delete *arpc2*, 5′ and 3′ homology regions were amplified using primers P25/26 and P27/28, respectively. The plasmid pBAT-SIL6-mCherry was digested with NaeI/SacI and AvrII/AatII to insert the 3′ and 5′ homology regions via four-fragment Gibson assembly, resulting in the plasmids

pBAT-Alp5b-KO and pBAT-ARPC2-KO. The final plasmids were digested using NotI for transfection into wild-type $Pb_{473}$.

To tag NDC80, we first generated an mCherry tagging vector by amplifying mCherry with primers P29/30 from the vector pBAT-SIL6-mCherry and cloning it into the EcoRV/XbaI-digested pL18-ARC40-GFP vector. This plasmid was digested with HpaI/SpeI, and the *ndc80* 3′ region, amplified with primers P31/32, was inserted using Gibson assembly. Finally, mCherry was replaced with *yfp* by inserting the *yfp* gene, amplified with primers P33/34 from pSYFP2-C1, into the XbaI/MscI-digested vector via Gibson assembly. The resulting final plasmid pL18-NDC80-YFP was digested with EcoRV before transfection into wild-type $Pb_{473}$ or $Pb_{473}$ARC40(−)ns.

**Transfection.** Transgenic *P. berghei* parasite lines were generated largely as previously described[65,66]. In brief, for all transfections, 10 μg DNA were digested overnight, precipitated using ethanol and resuspended in 10 μl PBS. Schizonts were obtained by culturing 500 μl blood containing >1% parasitaemia in schizont medium (RPMI-1640 (Gibco, 52400-025) supplemented with 20% fetal calf serum (FCS) (Gibco, 26140-079)) and 1 μg ml$^{-1}$ gentamycin (PAA) for ~20 h at 37 °C, and were then purified over a 55% Nycodenz (Axis-Shield Diagnostics) gradient. DNA and schizonts were mixed with 100 μl Nucleofector solution (either from the parasite transfection kit or from the human T cell Nucleofector kit), electroporated using the Amaxa Nucleofector II device (Lonza) and immediately injected intravenously (i.v.) into a mouse. Transgenic parasites were selected by administering pyrimethamine (7 μg ml$^{-1}$, Sigma-Aldrich) to the drinking water 1 day after transfection. Blood-stage-positive mice were bled by cardiac puncture and parasites were genotyped as described below. If correct transgenesis was observed, single clones were obtained by limiting dilution for all lines except for ARPC2-YFPint, ARPC1-mCherry/CBE and both $Pb_{473}$NDC80-YFP and $Pb_{473}$ARPC1(−)ns-NDC80-YFP. For this purpose, blood from an infected donor mouse was collected and serially diluted to contain a single parasite per 100 μl, which was injected intravenously into 4–6 mice. Mice were followed up for up to 14 days and bled upon reaching parasitaemia over 1% for genotyping and cryostabilates. One to two clones of each transgenic line was used for further phenotypic analysis.

To obtain a marker-free line by negative selection, a $Pb_{473}$ ARPC1(−)-infected mouse was treated with 5-fluorocytosine in the drinking water (1 mg l$^{-1}$, Sigma-Aldrich). Upon reappearance of parasites in the blood, loss of the human dihydrofolate reductase (hDHFR) selection cassette was verified by genotyping as described below and a single clone was obtained by limiting dilution.

**Genotyping.** For genotyping of transgenic *P. berghei* lines, blood containing at least 1% parasitaemia was lysed using 0.093% saponin and the parasite pellet was resuspended in 200 μl PBS. gDNA was isolated using the DNeasy Blood and Tissue kit (Qiagen) according to manufacturer instructions. Parasites were genotyped by PCR for amplification of the wild-type locus (locus) or whole locus (WL), integration (int) and negative selection (ns) sites using the following primer sets: ARPC1-GFP: P35/36 (locus), P35/37 (int); ARPC1-HA and ARPC1-mSc/CB-EME: P35/36 (locus), P35/38 (int); ARPC1(−): P39/40 (locus), P41/42 (int); ARPC1(−)ns and ARPC1(−)compl: P41/43 (WL), P41/42 (int), P44/43 (ns), P39/40 (locus). ARPC2-YFP: P45/46 (locus), P47/48 (int); Alp5b(−): P49/50 (WL), P49/51 (5′ int), Pb50/52 (3′ int). ARPC2(−): P53/P54 (WL), P53/P55 (5′ int), P54/56 (3′ int); $Pb_{473}$NDC80-YFP/$Pb_{473}$ARPC1(−) ns-NDC80-YFP: P57/P58 (locus), P57/P59 (int), P44/43 (ns). *Pb*AKiT7-YFP: P64/65 (locus), P64/P38 (5′ int), P66/65 (3′ int).

### Imaging localization of Arp2/3 subunits and interaction partners
#### Blood stage, gametocyte and ookinete immunofluorescence assays. To detect ARPC1 expression in ARPC1-GFP or in $Pb_{820}$

ARPC1-HA, we performed immunofluorescence of blood stages and ookinetes largely as previously described[67]. Mixed blood stages were obtained from mice infected 3 days before bleeding with $2.5 \times 10^7$ infected red blood cells (iRBC) i.p. Schizonts were obtained by culturing 500 μl blood containing >1% parasitaemia in schizont medium (see above) for ~20 h at 37 °C. Ookinetes were obtained by infecting a mouse with $2 \times 10^7$ iRBC i.p. and 3 days later culturing ~500 μl blood in 10 ml ookinete medium (RPMI supplemented with 20% (v/v) FCS, 50 μg ml$^{-1}$ hypoxanthine and 100 μM xanthurenic acid, pH 7.8–8.0) for 20 h at 19 °C, followed by purification over a 63% Nycodenz gradient.

For fixing gametocytes at various time points after activation, mice were infected either with $2.5 \times 10^7$ iRBC i.p. and bled 3 days later, or with $1.5 \times 10^7$ iRBC i.p. and bled 4 days later. The blood was immediately stored at 37 °C and 100 μl of blood were transferred to 400 μl pre-warmed schizont medium for the non-activated sample. To activate gametogenesis, 100 μl of blood were incubated with 400 μl ookinete medium at 19 °C. In case of drug treatment, drugs were added to the ookinete medium at indicated concentrations. At indicated time points (0, 3, 5, 7, 12 and/or 15 mpa), gametogenesis was stopped by adding PFA fixative (see below).

All cell suspensions were fixed by adding equal volumes of 4% PFA/0.0075% glutaraldehyde in PBS and incubating for 30 min at 37 °C. Cells were washed once in 1 ml PBS and permeabilized for 15 min at room temperature in 125 mM glycine/0.1% Triton-X-100 in PBS. After blocking in 3% BSA/PBS for at least 1 h at room temperature, cells were incubated in primary antibody for 4 h at room temperature or overnight at 4 °C. Cells were washed three times for 10 min with 1% BSA/PBS and secondary antibody was added for 1–2 h incubation at room temperature. Cells were washed thrice for 10 min in 1% BSA/PBS, adding Hoechst 33342 at a final concentration of 10 μg ml$^{-1}$ to the second wash step. Cells were pelleted and 2 μl of cell pellet placed on a glass slide and covered with a cover slip. Cells were imaged either on a Zeiss Axiovert 200M fluorescence microscope using a ×63 objective, on a Zeiss CellDiscoverer 7 fluorescence microscope using a ×50 water immersion objective, or on a Zeiss LSM900 microscope equipped with an Airyscan detector using a Plan-Apochromat ×63/1.4 oil immersion objective. To detect ARPC1-HA expression in male and female gametocytes of $Pb_{820}$ARPC1-HA, the stained samples were additionally analysed by flow cytometry on a BD FACSCelesta cell analyser.

Primary antibodies used were rabbit anti-GFP (1:50 dilution, Invitrogen, G10362), mouse anti-tubulin (1:1,000 dilution, Sigma-Aldrich, T5168-2ML) and rat anti-HA (1:1,000 dilution, Roche, 11867423001). Secondary antibodies used were Alexa-546 anti-rabbit (Invitrogen, A11035), Alexa-488 anti-mouse (Invitrogen, A11029), Alexa-488 anti-rabbit (Invitrogen, A11008) and Alexa-647 anti-rat (BD Pharmingen, 51-9006589), all at a 1:1,000 dilution. For all immunofluorescence assays, matching wild-type Pb samples were stained in parallel to control for unspecific staining of the anti-GFP or the anti-HA antibody, and no major unspecific staining was detected.

**Live-cell imaging of gametocytes.** For live-cell imaging of ARPC1-mCherry/CBE, a drop of blood from a highly parasitaemic mouse was mixed with 3 ml ookinete medium and Hoechst 33342 at a final dilution of 10 μg ml$^{-1}$. Cells were immediately placed on a glass slide, covered with a cover slip and imaged on Zeiss Axiovert 200M fluorescence microscope using a ×63 objective.

**Oocysts.** To detect ARPC1-GFP expression in ARPC1-GFP oocysts, mosquitoes were infected as described below and midguts were dissected at 5 and 12 days after feeding. Midguts were stained with Hoechst at a final concentration of 10 μg ml$^{-1}$ in PBS for 30 min at 37 °C and imaged on a Zeiss Axiovert 200M fluorescence microscope using a ×63 objective. Wild-type Pb oocysts imaged with the same settings did not display a GFP signal.

**Sporozoites.** Mosquitoes were infected with ARPC1–GFP as described below and salivary glands were dissected and crushed in RPMI medium on day 18 to isolate sporozoites. Sporozoites were seeded into 4 wells of an 8-well ibiTreat LabTek slide (ibidi) ($4 \times 10^5$ sporozoites per well) and centrifuged for 3 min at 800$g$. After a 10 min incubation at room temperature, the supernatant was removed and wells were washed twice with RPMI medium. Cells were fixed with 250 µl 4% PFA/0.0075% glutaraldehyde for 1 h at room temperature, washed thrice with PBS and permeabilized for 1 h with 0.5% Triton-X-100/PBS at room temperature. After additional 3 washes, cells were blocked for 1 h in 3% BSA/PBS and incubated with rabbit anti-GFP (1:50 in 3% BSA/PBS, 200 µl per well, Invitrogen, G10362) for 1 h at 37 °C. Cells were washed three times and incubated with Alexa-546 anti-rabbit secondary antibody (1:1,000 in 3% BSA/PBS, 200 µl per well, Invitrogen, A11035) for another hour at 37 °C. Cells were finally washed three times with PBS, adding Hoechst at a final concentration of 10 µg ml$^{-1}$ to the second wash step and incubating for 10 min at room temperature. Sporozoites were imaged on a Zeiss Axiovert 200M fluorescence microscope using a ×63 objective.

**Liver stages.** HepG2 cells, cultured under standard conditions in DMEM supplemented with 10% FCS, 1 mM glutamine and 1% antibiotica-antimycotica (Gibco), were seeded into 8-well slides (Nunc) at a density of $2 \times 10^4$ cells per well. Two days later, sporozoites were isolated from mosquitoes as described above, diluted to 100 sporozoites per µl in 3% BSA/RPMI and 200 µl of sporozoite mixture was added per well. Cells were incubated for 1 h at 37 °C to allow invasion before washing once with complete DMEM. Cells were maintained in 400 µl medium per well at 37 °C, fixed at 48 h after infection in 200 µl ice-cold methanol for 10 min at −20 °C, washed with 1% FCS/PBS and blocked overnight in 10% FCS/PBS. Cells were stained with rabbit anti-GFP (1:50 dilution, Invitrogen, G10362) and mouse anti-PbHsp70 (ref. 68) diluted 1:300 for 2 h at 37 °C, followed by three washes with 1% FCS/PBS and secondary staining using Alexa-546 anti-rabbit (Invitrogen, A11035), Alexa-488 anti-mouse (Invitrogen, A11029) at 1:1,000 dilution for 1 h at 37 °C. Hoechst was added to a final concentration of 2.5 µg ml$^{-1}$ and cells were incubated for another 15 min at room temperature before washing three times with 1% FCS/PBS and mounting with 10% glycerol/PBS. Cells were imaged on a Zeiss Axiovert 200M fluorescence microscope using a ×63 objective.

### Characterization of parasite development across the life cycle

**Asexual growth and gametocyte formation.** To determine parasite growth and gametocyte formation, 4 TO mice per parasite line were infected intravenously with 1,000 iRBC. Parasitaemia was assessed from days 4 to 10 after infection by staining a drop of blood in DRAQ5 fluorescent probe (1:1,000 diluted in PBS) for 10 min at room temperature, washing the cells once with PBS and analysing them by flow cytometry on a MACSQuant VYB flow cytometer.

**Ookinete formation and motility assays.** For ookinete formation and motility assays, TO mice were pretreated with 200 µl phenylhydrazine (6 mg ml$^{-1}$ in PBS) intraperitoneally to induce reticulocytosis, before infecting them 2 days later by i.p. injection of a parasite cryostock. Upon reaching a parasitaemia above 5, asexual parasites were killed by supplementing drinking water with sulphadiazine (30 mg l$^{-1}$). Two days later, mice were bled and the gametocyte-containing blood transferred to 10 ml ookinete medium. Parasites were cultured at 19 °C for 20–24 h. Ookinetes were purified using a 63% Nycodenz gradient, washed once with ookinete medium, and 2 µl of the pellet transferred onto a glass slide and covered with a cover slip. Ookinetes were imaged for 15 min at a frame rate of 20 s on a Nikon A1R inverted confocal microscope using a ×60 oil objective.

**Mosquito infections.** To infect mosquitoes, two mice were infected with either $2 \times 10^7$ iRBC i.p. or $2 \times 10^6$ iRBC i.v. Three days later, mice were anaesthetized by administering ketamine/xylazine solution (20 mg ml$^{-1}$

ketamine, 0.6 mg ml$^{-1}$ xylazine in PBS) i.p. at 5 µl g$^{-1}$ body weight and placed onto a mosquito cage containing ~400–500 mosquitoes. Mosquitoes were allowed to feed for ~30 min, with a change in mouse position after 15 min. After feeding, mosquitoes were immediately kept at 21 °C and 80% humidity, and fed with 10% (v/v) saccharose with 0.05% (w/v) para-aminobenzoic acid and 1% (w/v) NaCl.

**Oocyst quantification and size determination.** Infection intensity was assessed at 4, 6 and 12 days after blood meal by dissecting 10–30 midguts in PBS. Dissected midguts were placed on a glass slide, covered with a cover slip and imaged using a Leica AF6000 LX or a Zeiss Axiovert 200M fluorescence microscope with a ×10 objective to determine prevalence and oocysts per midgut. To measure oocyst area, infected midguts were imaged either on a Nikon A1R inverted confocal microscope using a ×25 objective or on a Zeiss Axiovert 200M fluorescence microscope using a ×25 objective. The area of oocysts was determined in FIJI[69] by thresholding oocysts on the basis of their fluorescence and measuring the size.

**Sporozoite numbers.** To determine sporozoite production, 15–40 mosquitoes were dissected at 17–18 days after blood meal, and salivary glands transferred into 100 µl PBS. Organs were disrupted mechanically using a plastic pestle to release sporozoites. Sporozoites were counted using a haemocytometer on a Zeiss Axiostar light microscope under a ×40 objective with phase contrast.

**By-bite infections.** Natural transmissions by mosquito bite to mice were performed at 18–20 days after blood meal. TO or C57BL/6 mice were anaesthetized by administering ketamine/xylazine solution (20 mg ml$^{-1}$ ketamine, 0.6 mg ml$^{-1}$ xylazine in PBS) i.p. at 5 µl g$^{-1}$ body weight and placed either on a full infected mosquito cage or on cups containing 10 female infected mosquitoes. Mice were exposed to mosquito bites for 10–15 min and were bitten by at least 7 mosquitoes. Mice were monitored from days 3 to 14 after infection by daily blood smears.

### Crossing of parasite lines

To determine sex specificity of the ARPC1(−) phenotype, we crossed Pb$_{473}$ARPC1(−) with Pb48/45(−) or Pb47(−) parasites[31,32] by mixing parasite lines at equal ratio and injecting two TO mice with $2 \times 10^6$ mixed parasites each. At 3 days after infection, mice were anaesthetized and fed to mosquitoes as described above. Oocyst size was determined at 6 days after blood meal from red-fluorescent oocysts as described above.

### Ookinete EM reconstructions

Purified ookinetes were fixed overnight at 4 °C in 2% PFA/2% glutaraldehyde/0.1 M cacodylate buffer and processed as previously described[70]. Embedded ookinetes were sectioned to 200-nm-thick sections and imaged on a transmission electron microscope (at 200 kV; Tecnai F20 TEM, FEI) equipped with an Eagle 4k × 4k CCD camera (FEI). Bidirectional tilt series were acquired from −60° to +60° in 2° increments, with a magnification of ×9,600 (pixel size 1.118 nm) and ×14,500 (pixel size 0.74 nm) for the wild-type and ARPC1(−) ookinete nuclei, respectively. Tomograms were reconstructed, joined and segmented using IMOD[71]. The inner nuclear membrane, microtubule-organizing centres and spindle microtubules were rendered using 3dmod[71].

### DNA content of parasites

**Quantification of DNA content.** For all DNA content analysis, wildtype and mutant or drug- and solvent-treated parasites were prepared as described below and stained in parallel using the same dilution of Hoechst 33342. Images were taken, focusing on the widest area of the nucleus, and analysed in FIJI[69] using a self-written macro. In brief, nuclei were segmented using an automatic thresholding function, and the DNA signal was measured as the total fluorescence intensity of the nucleus area minus the average background fluorescence signal. Where

appropriate, DNA signal was normalized to wildtype DNA signal measured in parallel or to single nucleated cells imaged on the same slide.

**Ookinetes.** Ookinetes were produced in vitro as described for motility assays but without sulphadiazine supplementation, as we found that adding sulphadiazine to the drinking water impedes with DNA replication in wildtype ookinetes. Nycodenz-purified ookinetes were stained with Hoechst (final concentration 10 µg ml$^{-1}$) for 10 min and imaged on a Zeiss Axiovert 200M microscope using the ×63 objective.

**Oocysts.** Mosquito midguts were dissected at 6 days after blood meal. Midguts were incubated in Hoechst (3 µM in 3% BSA/PBS) for 30 min at 37 °C and washed twice in 3% BSA/PBS. Midguts were imaged on a Nikon A1R inverted confocal microscope using the ×100 objective, taking Z-stack images. Image analysis was performed on sum Z projections.

**Microgametes.** Mice were infected with $2 \times 10^7$ iRBC i.p., bled 3 days later and gametocytes were purified using a 49% Nycodenz gradient kept at 37 °C. Purified gametocytes were resuspended in 500 µl of ookinete medium (containing drugs as indicated), incubated for 20 min at 19 °C and pelleted to obtain blood smears on glass slides. Smears were fixed with ice-cold methanol for 5 min, rehydrated for 1 h in 3% BSA/PBS and stained with mouse anti-tubulin (1:500–1,000 in 3% BSA/PBS, Sigma-Aldrich, T5168-2ML), followed by secondary antibody Alexa-546 anti-mouse (Invitrogen, A111030, 1:500–1,000 in 3% BSA/PBS) and Hoechst at 10 µg ml$^{-1}$. Cells were mounted in 10% glycerol/PBS and imaged on a Zeiss CellDiscoverer 7 fluorescence microscope using a ×50 objective. Microgametes were identified by their tubulin signal and scored as DNA positive or DNA negative before proceeding to determination of DNA content.

**DNA content during fertilization and ookinete development.** Mice were infected with $2 \times 10^7$ iRBC i.p. or $2 \times 10^6$ iRBC i.v. and bled 3 days later. The blood was immediately transferred to ookinete medium (6 wells, each with 100 µl blood and 2 ml ookinete medium) and incubated at 19 °C to induce gametogenesis and fertilization. As non-activated sample, 100 µl of blood were kept at 37 °C and fixed immediately. At 1, 2, 4, 8 and 24 h after induction each, one well containing 100 µl blood was collected and fixed by adding equal volumes of 4% PFA/0.0075% glutaraldehyde/PBS for 10 min at room temperature, followed by one wash in PBS. After collection of all time points, cells were stained with Hoechst (final concentration 10 µg ml$^{-1}$) for 15 min at 37 °C and imaged on a Zeiss CellDiscoverer 7 fluorescence microscope using a ×50 objective. Female gametocytes, zygotes and ookinetes were identified on the basis of their red fluorescence in the reporter line Pb$_{820}$.

**Co-immunoprecipitation and mass spectrometry**
**Co-immunoprecipitation of ARPC1–GFP, ARPC2–YFP and AKiT7–YFP.** *Pb*ARPC1–GFP, *Pb*ARPC2–YFPint or *Pb*AKiT7–YFP gametocytes, each time along with *Pb*GFP$_{con}$ gametocytes, were purified from the blood of highly parasitaemic mice as described above. For each replicate, gametocytes from two mice were pooled and then divided into two samples, with one being fixed immediately (non-activated sample) and the other being activated in ookinete medium for 4 min at 19 °C (activated sample). Cells were pelleted, immediately fixed by resuspension in 1% (v/v) formaldehyde for 10 min and then quenched with 0.125 M glycine in PBS for 5 min. ARPC1–GFP, ARPC2–YFP and AKiT7–YFP were co-immunopurified using the GFP-Trap Agarose kit (gtak-20) by ChromoTek according to manufacturer instructions. In brief, the lysis buffer was supplemented with 2.5 mM MgCl$_2$ and 100 U DNaseI, and both lysis and dilution buffer were supplemented with 1× concentration of Halt Protease and Phosphatase Inhibitor Cocktail (Invitrogen). Gametocytes were pelleted, resuspended in 400 µl lysis buffer and lysed for 60 min on ice with regular vortexing. The GFP-trap beads were added to the supernatant and incubated rotating over end

for 60 min. The beads were rinsed three times with wash buffer and proteins were finally released by resuspending beads in 2× Laemmli SDS-sample buffer and denaturation at 95 °C for 5 min. Samples were separated on an SDS–PAGE to be analysed for mass spectrometry. For each pulldown sample, four replicates were collected.

**In-gel tryptic digestion.** Upon SDS–PAGE, Coomassie-stained bands (2–3 per analysed sample) were manually excised from the gel. The in-gel digestion was performed as previously described[72,73]. Peptides from corresponding gel lanes were combined (1–3, 4–6 and so on).

**LC–MS/MS analysis.** Nanoflow LC–MS/MS analysis for ARPC1–GFP and ARPC2–YFP was performed with an Ultimate 3000 liquid chromatography system coupled to an Orbitrap QE HF mass spectrometer (Thermo Fisher). An in-house packed analytical column (75 µM × 200 mm, 1.9 µM ReprosilPur-AQ 120 C18 material; Dr Maisch) was used. Mobile-phase solutions were prepared as follows: solvent A: 0.1% formic acid, 1% acetonitrile; solvent B: 0.1% formic acid, 89.9% acetonitrile. Peptides were separated in a 60 min linear gradient starting from 3% B and increased to 23% B over 50 min and to 38% B over 10 min, followed by washout with 95% B. The mass spectrometer was operated in data-dependent acquisition mode, automatically switching between MS and MS2. MS spectra ($m/z$ 400–1,600) were acquired in the Orbitrap at 60,000 ($m/z$ 400) resolution and MS2 spectra were generated for up to 15 precursors with normalized collision energy of 27 and isolation width of 1.4 $m/z$. Nanoflow LC–MS/MS analysis for AKiT7–YFP was performed using a Vanquish Neo system coupled to an Orbitrap Tribrid Eclipse mass spectrometer (Thermo Fisher). An in-house packed analytical column (75 µM × 200 mm, 1.9 µM ReprosilPur-AQ 120 C18 material; Dr Maisch) was used. The mobile-phase solutions were prepared as follows: solvent A: 0.1% formic acid; solvent B: 0.1% formic acid and 80% acetonitrile. Peptide separation was achieved using a 30 min linear gradient, starting at 4% solvent B and gradually increasing to 32% over the first 25 min, followed by a further increase to 49% over the next 5, and finally a wash step with 99% solvent B. The Orbitrap Tribrid Eclipse was operated in data-dependent acquisition mode, with similar settings to those described previously[74]. MS1 spectra were acquired in the Orbitrap with 120,000 resolution, AGC target = $1.2 \times 10^6$, MaxIT = 50 ms, RF lens = 30% and mass range = 400–1,600. Dynamic exclusion was used for 10 s excluding all charge states for a given precursor. Monoisotopic peak determination was switched on with the isolation window centre set to the most abundant peak. Singly charged ions and ions with charge states above 6 were filtered out. The intensity threshold filter was set to $5 \times 10^3$. MS2 spectra were collected in the Orbitrap with 30,000 resolution, custom AGC target = $1.25 \times 10^5$, MaxIT = 54 ms, mass range 150–1,350 and NCEHCD = 30%. Quadrupole was used for the precursor isolation with the isolation window set to 0.5 $m/z$.

**Database search.** The generated data were searched using Proteome Discoverer 2.5 (ARPC1–GFP and ARPC2–YFP) or 3.1 (AKiT7–YFP) with Sequest HT (Thermo Fisher) and Inferys Rescoring[75]. For ARPC1–GFP and ARPC2–YFP, the fragment ion mass tolerance was set to 0.6 Da and the parent ion mass tolerance to 10 ppm. For AKiT7–YFP, the fragment ion mass tolerance was set to 0.02 Da and the parent ion mass tolerance to 5 ppm. Trypsin was specified as an enzyme. Carbamidomethyl was set as a fixed modification of cysteine, and oxidation (methionine) and deamidation (asparagine, glutamine) as variable modifications of peptides. Acetylation, methionine loss and a combination of acetylation and methionine loss were set as variable modifications of protein termini. Peptide quantification was done using the precursor ion quantifier node with the Top N Average ($n$ = 3) method set for protein abundance calculation.

The MS/MS spectra were searched against the following databases: the customized contaminant database (part of MaxQuant, MPI Martinsried[76]), UniProt *M. musculus* protein database (17,202

sequences, UP000000589) and the UniProt *P. berghei* protein database (4,927 sequences, UP000074855).

**Data analysis.** For each immunoprecipitation experiment, only *P. berghei* proteins detected in at least two out of four ARPC1–GFP-activated samples, two out of four ARPC2–YFP-activated samples, or two out of four AKiT7–YFP-activated samples were included in further data analysis. The dataset was processed in Perseus (v.2.0.10)[77] and R (v.4.2.2). Protein abundance was $\log_2$ transformed and missing values were imputed in Perseus (width 0.3, downshift 1.8). The data were normalized by Z transformation on each column separately before calculating fold changes. P values were calculated using a two-sided Students t-test and corrected for multiple comparisons using the Benjamini–Hochberg procedure.

### In silico structural analysis

Protein structure and interface predictions were modelled by Alpha-Fold 3 (ref. 39) and aligned against each other using UCSF ChimeraX (v.1.9). Protein structure prediction alignments between *Plasmodium* and human orthologues were generated by the 'all against all' feature of the Dali server[40].

### Statistics and reproducibility

Unless stated otherwise, experiments were repeated at least three independent times, and exact sample sizes are given in the figures. Genotyping experiments were performed only once due to their qualitative nature (detection of presence or absence of a genetic modification), using appropriate negative controls in parallel. All microscopy images shown are representative of at least five images taken per condition. Where possible, images were analysed and quantified in a single-blinded manner to minimize bias. Error bars indicate s.e.m. unless stated otherwise.

### Software

Most data were plotted and analysed in GraphPad Prism (v.10.0.3), except for mass spectrometry data, which were analysed using Proteome Discoverer 2.5 and Perseus (v.2.0.10) and plotted with R (v.4.2.2). Flow cytometry data were analysed using FlowJo (v.10.7.2). ZEN Blue 3.01 software was used for the post-2D or 3D Airyscan processing with automatically determined default Airyscan settings. All images were further processed in FIJI (v.2.14.0). Structures were predicted using AlphaFold 3 and visualized using UCSF ChimeraX (1.9). EM tomograms were joined and reconstructed using IMOD (v.4.11.12).

### Reporting summary

Further information on research design is available in the Nature Portfolio Reporting Summary linked to this article.

### Data availability

The *Mus musculus* and *Plasmodium berghei* protein databases that were used in this study can be accessed at Uniprot (UP000000589 and UP000074855, respectively). Raw mass spectrometry proteomics data have been deposited to the ProteomeXchange Consortium via the PRIDE partner repository[78] with the dataset identifiers PXD046181, PXD051260 and PXD060997. All other data are included in the Article and the supplementary information. Source data are provided with this paper.

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

## Acknowledgements

We thank M. Reinig and A. McIlhoney for mosquito rearing; A. P. Waters and K. Huges for sharing parasite lines and plasmids; M. Ganter, R. Douglas and M. Meissner for discussion and critical comments to the manuscript; and M. Luzarowski and S. Merker from the Core Facility for Mass Spectrometry Proteomics (CFMP) at the ZMBH Heidelberg for technical support, performing mass spectrometry, and advice regarding data analysis. We further acknowledge the microscopy support from the Infectious Diseases Imaging Platform (IDIP) at the Center for Integrative Infectious Disease Research, Heidelberg; S. Gold and C. Funaya and the Electron Microscopy Core Facility (EMCF) Heidelberg for support in preparing and imaging EM samples. The *Plasmodium* database PlasmoDB facilitated this work. This study was funded by the following sources: WT Investigator award 110166 (F.H., M.M.), Wellcome center award 104111 (F.H., M.M.), Royal Society Wolfson Merit award (M.M.), German Center for Infection Research, DZIF (TTU 03.813) (F.F., F.H.), German Research Foundation grant: SPP 2225 (F.H. D.J., Y.S.), German Research Foundation grant: SPP 2332 (M.S., A.K., K.H., L.P.D.), and German Research Foundation grant: SFB 1129, project number 240245660 (M.S., P.A.). The CFMP is funded by the ZMBH and partially funded by the CellNetworks Core Technology Platform (CCTP) of Heidelberg University. The CCTP is funded in part by the Federal Ministry of Education and Research (BMBF) and the Ministry of Science Baden Württemberg within the framework of the Excellence Strategy of the Federal and State Governments of Germany. The purchase of the Orbitrap Tribrid Eclipse used in this study was partially funded by the German Research Foundation (DFG) - project number: 538758380.

## Author contributions

F.H., M.M. and F.F. conceptualized the project. F.H., M.M., F.F and M.S. developed the methodology. F.H., D.J., Y.S., P.A., A.K., K.H., L.P.D. and M.S. conducted investigation. F.H., D.J. and Y.S. performed formal analysis. F.H. and D.J. performed visualization. F.H., M.M. and F.F. acquired funding. F.H., M.S., M.M. and F.F. supervised the project. F.H. wrote the original manuscript draft. F.H., M.M. and F.F. reviewed and edited the manuscript.

## Competing interests

The authors declare no competing interests.

## Additional information

**Extended data** is available for this paper at https://doi.org/10.1038/s41564-025-02023-6.

**Correspondence and requests for materials** should be addressed to Franziska Hentzschel or Matthias Marti.

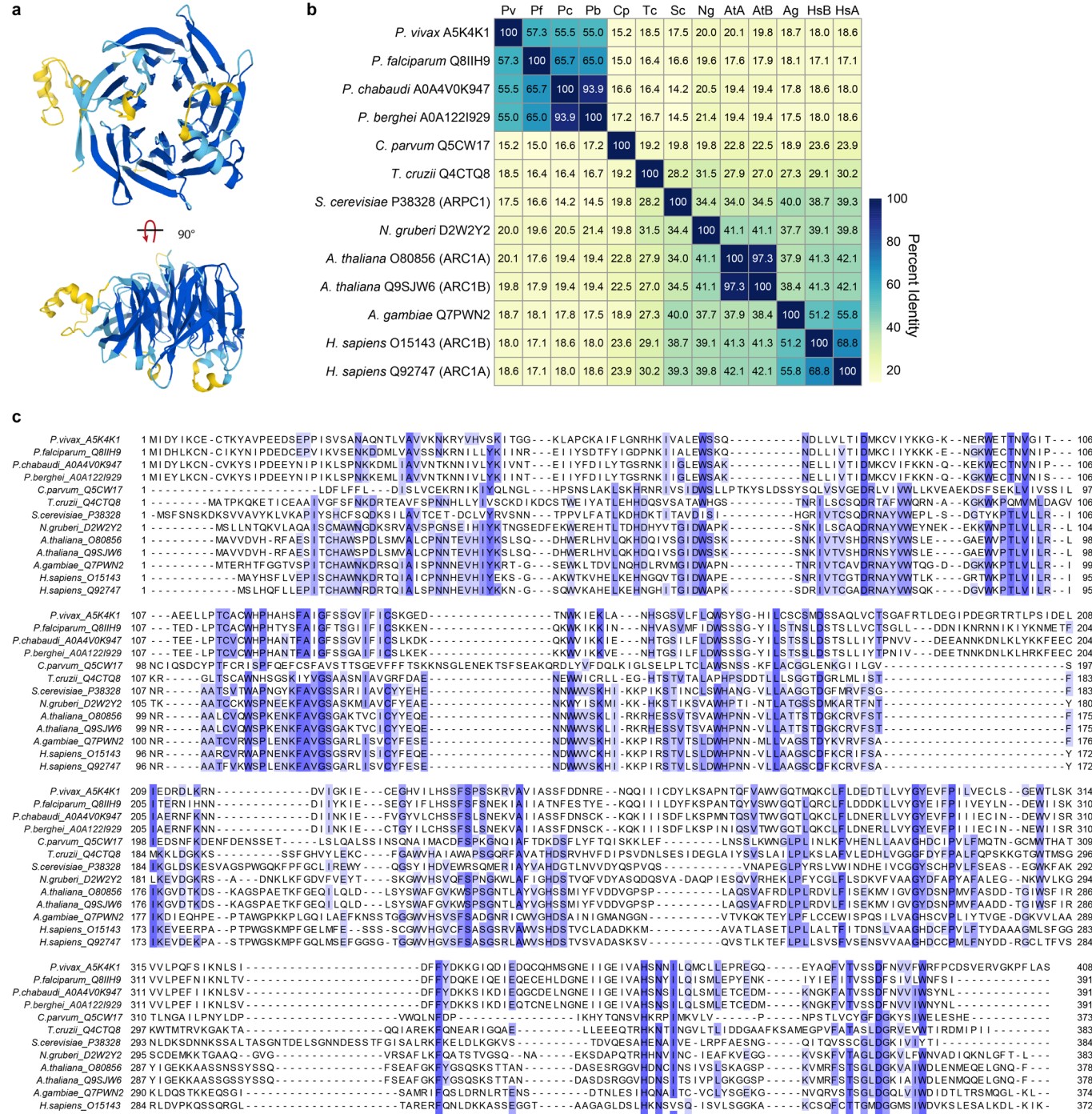

**Extended Data Fig. 1 | ARPC1 is a *Plasmodium*-specific WD40 domain protein. a**, Alphafold prediction of *P. berghei* ARPC1 structure. **b**, Percent Identity Matrix of *Plasmodium* ARPC1 and the ARPC1 subunit from other organisms, based on Clustal Omega alignments. **c**, Clustal Omega Alignment of putative ARPC1 sequences from *Plasmodium* and selected organisms.

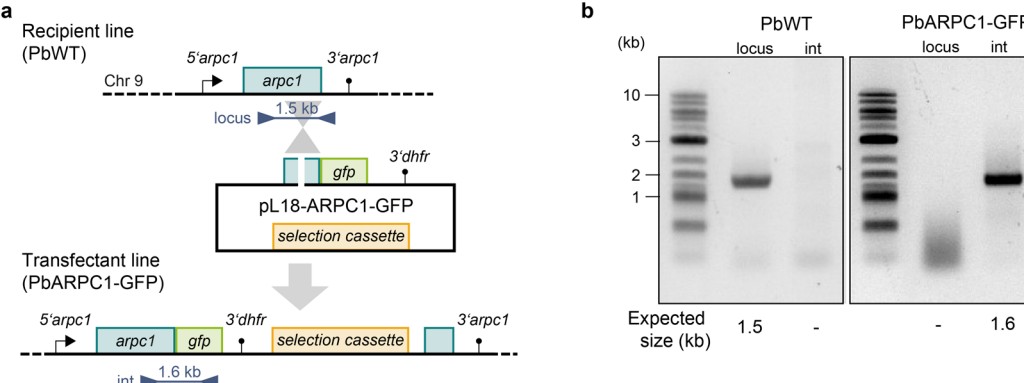

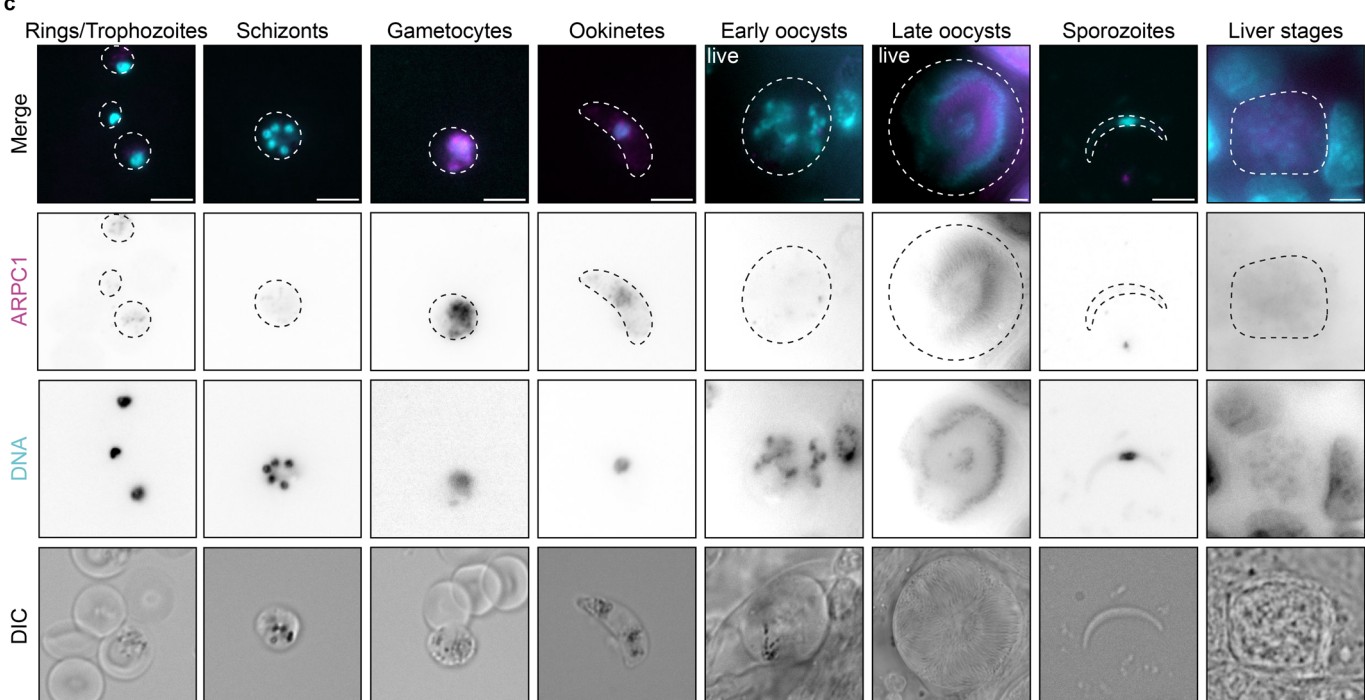

**Extended Data Fig. 2 | Generation of PbARPC1-GFP and localisation of ARPC1-GFP. a**, Scheme of genetic strategy. Primers (triangles) and expected amplicon sizes used for genotyping (see **b**) are indicated. Not drawn to scale. **b**, Genotyping PCR of PbARPC1-GFP. The expected size of the product is indicated below the gel images. **c**, Expression and localisation of ARPC1-GFP across the life cycle of *P. berghei*. Cells were fixed for immunofluorescent staining, except for oocysts, which were imaged live. Dashed lines indicate parasite circumference as determined from DIC images. Representative images of at least 10 images taken. Scale bars, 5 μm.

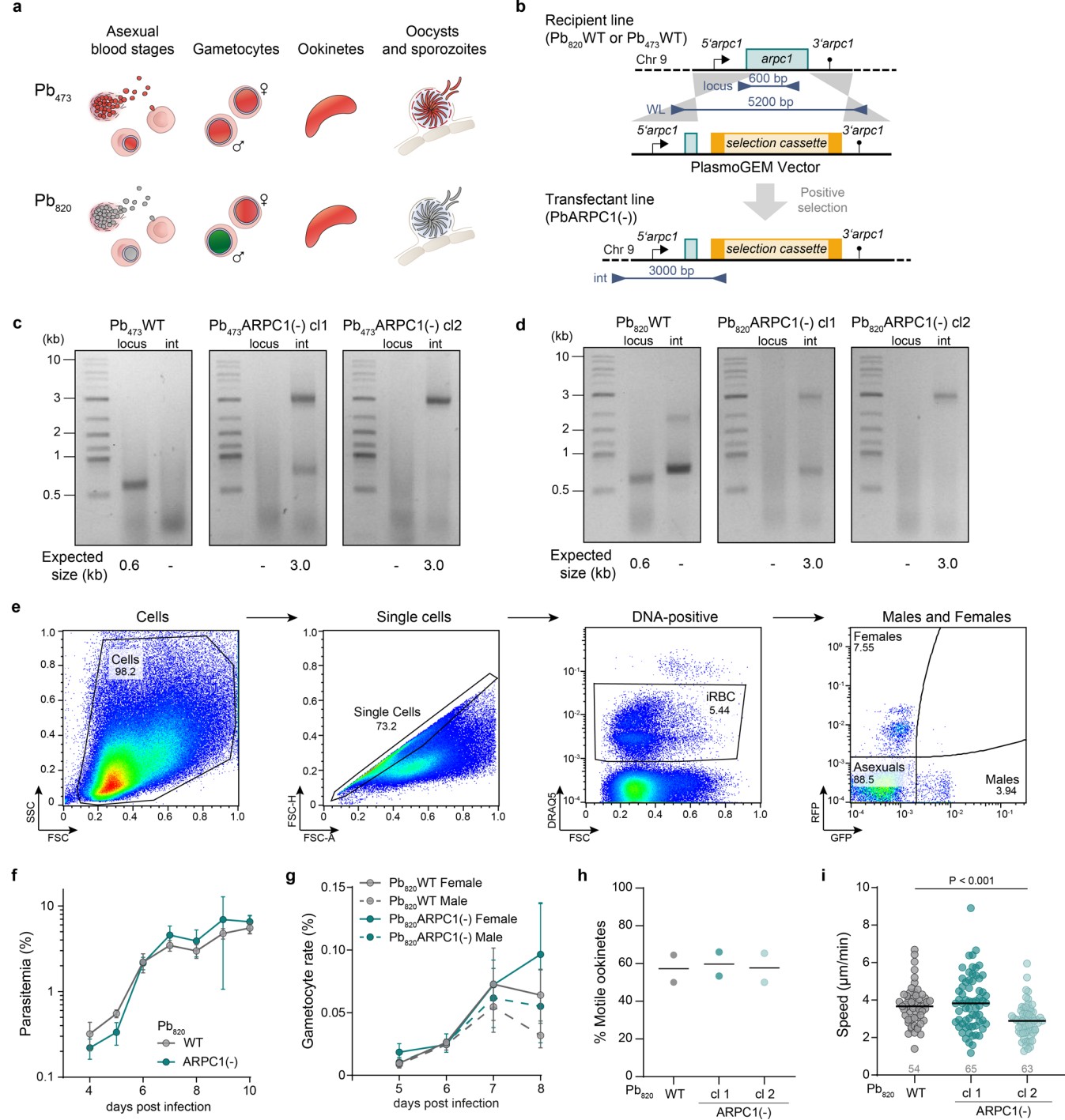

**Extended Data Fig. 3 | ARPC1 is not required for asexual growth, gametocyte formation or ookinete motility. a**, Schematic depiction of reporter lines used in this study. Pb473 expresses RFP under a strong promoter throughout the life cycle. Pb820 expresses GFP in male gametocytes and RFP in female gametocytes, gametes and ookinetes. **b**, Scheme of genetic strategy. Primers (triangles) and expected amplicon sizes used for genotyping (see **c,d**) are indicated. Not drawn to scale. **c,d**, Genotyping PCR of (**c**) Pb473ARPC1(−) and (**d**) Pb820ARPC1(−). Binding sites of the respective primers are indicated in **b**. The expected size of the product is indicated below the gel images. **e**, Gating strategy to determine

asexual growth and gametocyte formation in Pb820ARPC1(-). **f,g**, Asexual growth (**f**) and gametocyte rate (**g**) after intravenous inoculation of mice with 1000 iRBC. Mean ± s.d. of 4 mice per group. **h**, Proportion of motile ookinetes. Each data point corresponds to an independent experiment. **i**, Speed of moving ookinetes. Line indicates median. Pooled data from two independent experiments. Grey numbers above *x* axis indicate total number of observed ookinetes. Note that one of the two clones exhibited slightly but significantly lower ookinete motility. This is likely due to clonal variation and not caused by ARPC1 deletion. Statistics: One-Way ANOVA, Dunn's post test.

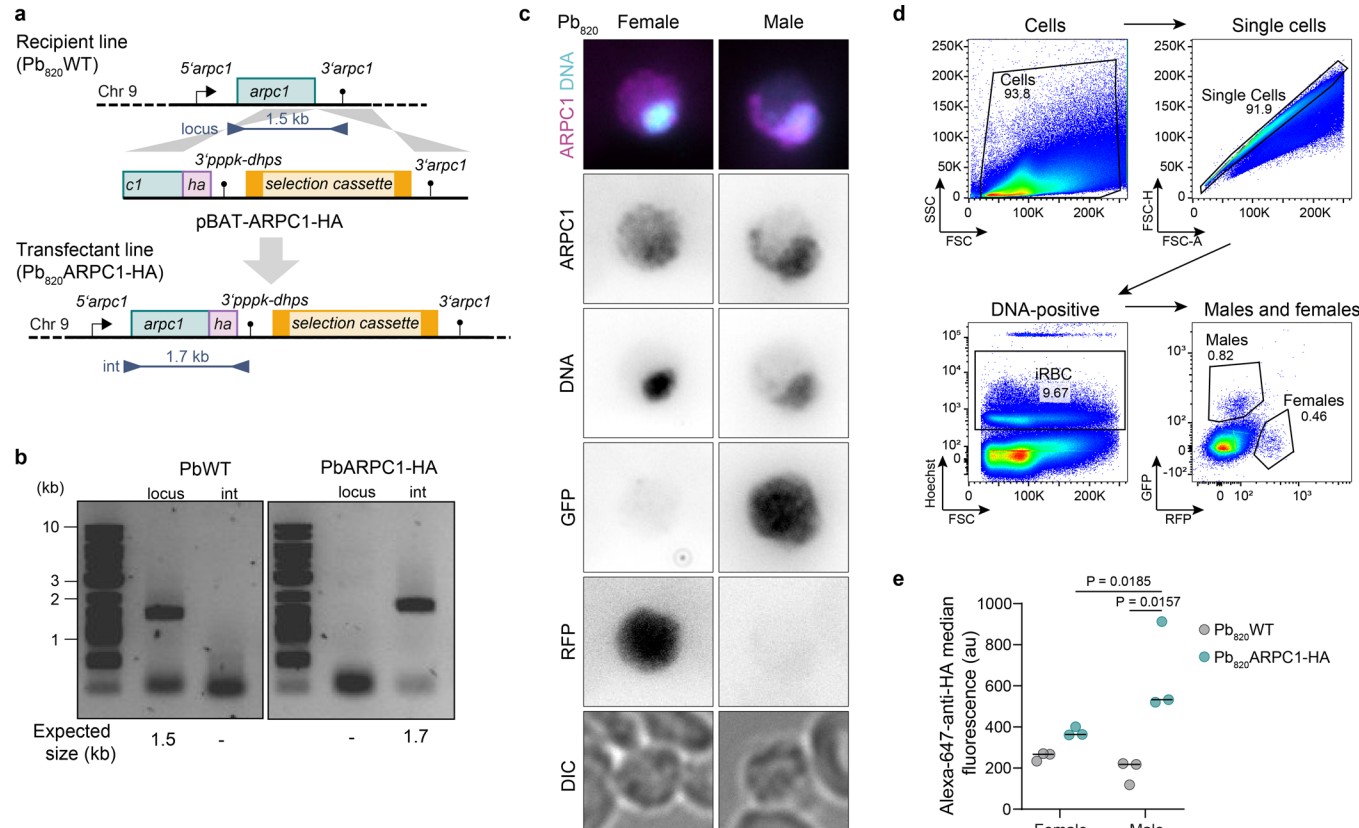

**Extended Data Fig. 4 | Generation of Pb₈₂₀ARPC1-HA and expression of ARPC1-HA. a**, Scheme of genetic strategy. Primers (triangles) and expected amplicon sizes used for genotyping (see **b**) are indicated. Not drawn to scale. **b**, Genotyping PCR. The binding sites of the respective primers are indicated in **a**. The expected size of the product is indicated below the gel images. **c**, ARPC1-HA expression and localisation in female (RFP-positive, first column) and male (GFP-positive, second column) gametocytes. Representative widefield images of at least 10 images taken. Scale bar, 5 µm. **d**, Gating strategy to identify ARPC1-HA expression in male and female gametocytes. **e**, Median fluorescence of males and females stained with Alexa-647-anti-HA as measured by flow cytometry. Each dot represents an independent biological replicate. Statistics: Two-way ANOVA.

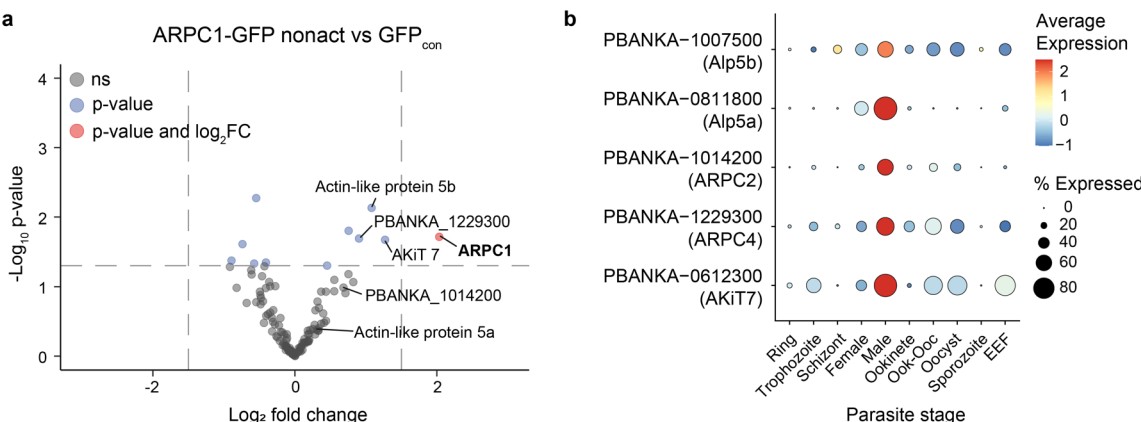

**Extended Data Fig. 5 | Pulldown of ARPC-1 GFP. a**, Enriched proteins after pulldown of ARPC1-GFP from non-activated gametocytes. Note that the actin like protein 5b, PBANKA_1229300 (ARPC4) and AKiT7 are significantly enriched, but below the fold change threshold of 1.5. **b**, Expression of ARPC1-interaction partners across life cycle. Data reanalysed from the malaria cell atlas[38].

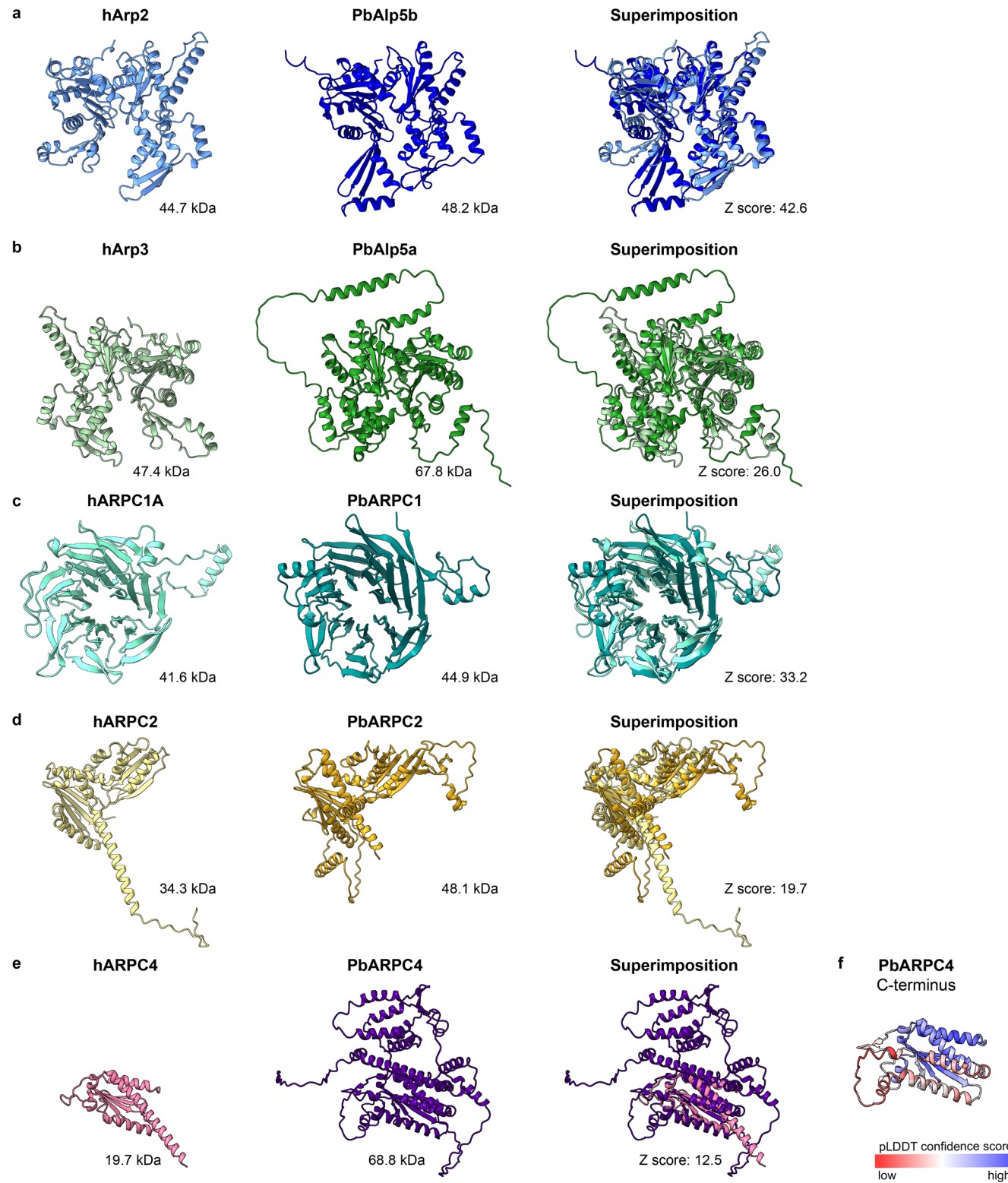

**Extended Data Fig. 6 | Structure prediction and comparison between human and *Plasmodium* subunits. a**, Alp5b/Arp2. **b**, Alp5a/Arp3. **c**, PbARPC1/HsARPC1A. **d**, PbARPC2/HsARPC2. **e**, PbARPC4/HsARPC4. Shown are structural prediction of human (first column) and *P. berghei* (second column) Arp2/3 subunits as well as the superimposition of the two structures (third column).

Protein size in kDa and *Z* scores of DALI structural comparisons are indicated. **f**, PBANKA_1229300 C-terminus structural prediction coloured by pLDDT confidence score indicating high confident prediction in regions aligning to hARPC4.

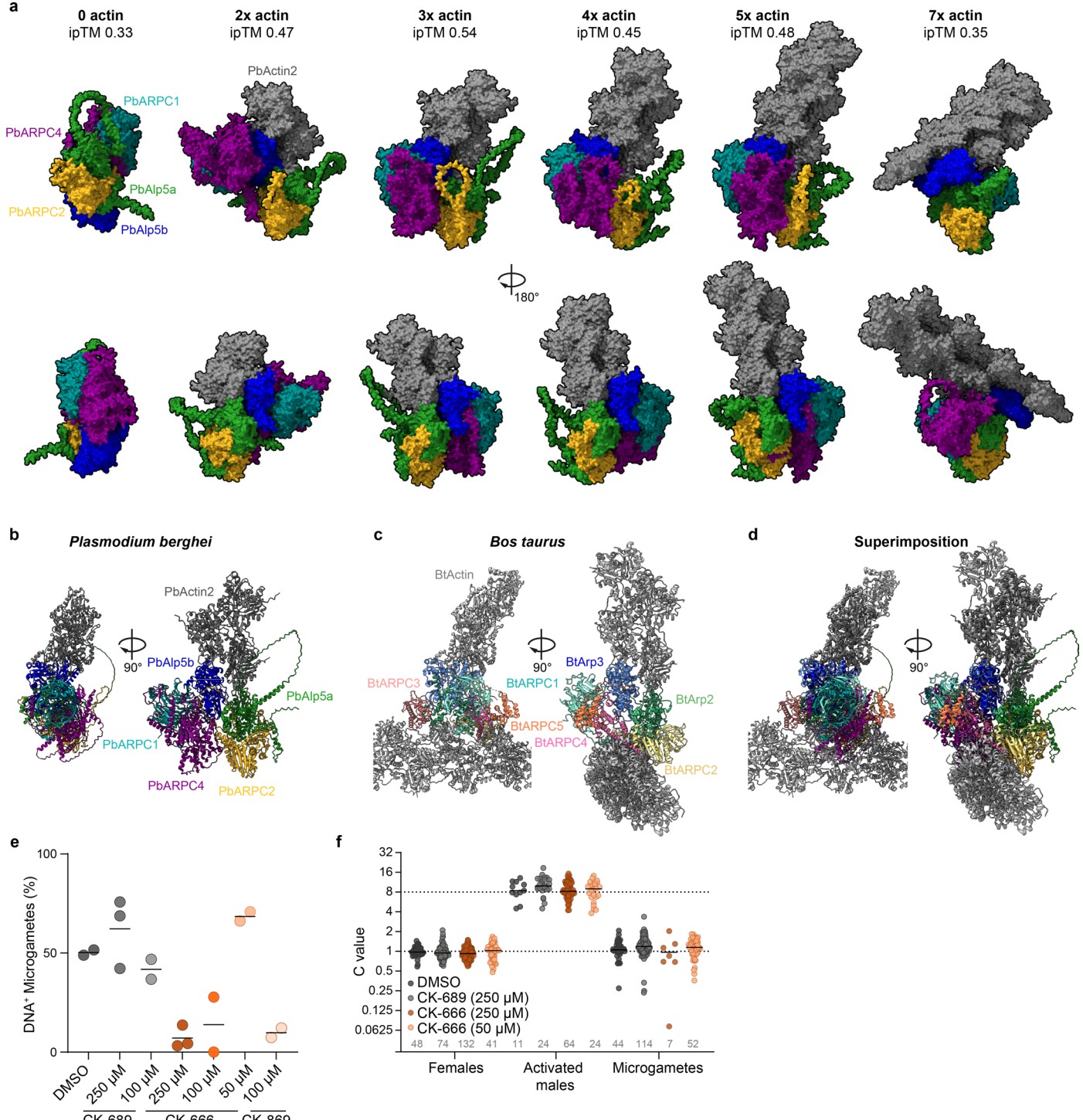

**Extended Data Fig. 7 | The predicted *Plasmodium* Arp2/3 complex structure and DNA segregation in presence of canonical Arp2/3 inhibitors.** **a**, Structure prediction of Arp2/3 subunits together with varying numbers of actin monomers. Prediction scores (idT values) are indicated below each structure. Note how the relative position of subunits differs in predictions with 0, 2 and 7 actin subunits, but is constant in those complexes with highest prediction stores (3, 4 and 5 actin monomers). **b-d**, Additional views of structures depicted in Fig. 4a-c. **b**, First rank of Alphafold 3 multimer prediction of PbAlp5a, PbAlp5b, PbARPC1, PbARPC2, PbARPC4 and three monomers of PbAct2. **c**, Experimentally determined structure of bovine Arp2/3 in the branch junction[45].

**d**, Superimposition of PbArp2/3 prediction and BtArp2/3 structure. **e**, Relative abundance of microgametes with detectable DNA signal after drug treatment with known Arp2/3 inhibitors. Line indicates mean. Each data point represents one independent experiment. **f**, DNA content of females, activated males, and microgametes in presence of DMSO, CK-689 (inactive control), or varying concentrations of CK-666. Values normalised to the mean DNA content of females imaged on the same slide. Dashed lines, expected C value of activated males (8) and microgametes (1). Pooled data from at least 2 independent experiments. Grey numbers above *x* axis indicate total numbers of cells analysed.

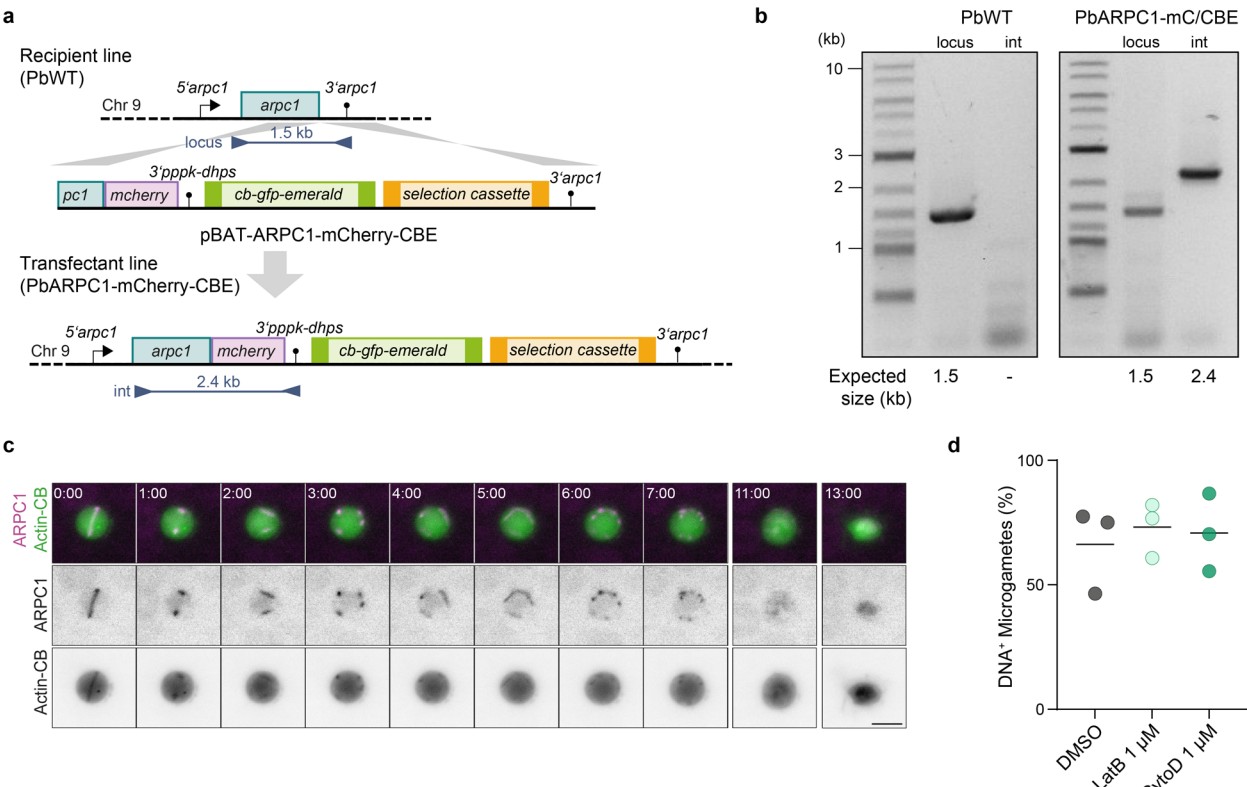

**Extended Data Fig. 8 | F-Actin dynamics during male gametogenesis. a**, Scheme of genetic strategy to generate PbARPC1-mCherry-CBE. Primers (triangles) and expected amplicon sizes used for genotyping (see **b**) are indicated. Not drawn to scale. **b**, Genotyping PCR of PbARPC1-mCherry-CBE. Primers are indicated in **a**. The expected size of the product is indicated below the gel images. **c**, Live cell imaging of activated PbARPC1-mScarlet-CBE gametocyte. Time points in minutes after start of the movie indicated in upper left corner. Scale bar, 5 µm. Representative images from at least 5 movies taken. **d**, Relative abundance of microgametes with detectable DNA signal after drug treatment with known actin inhibitors. Line indicates mean. Each data point represents one independent experiment.

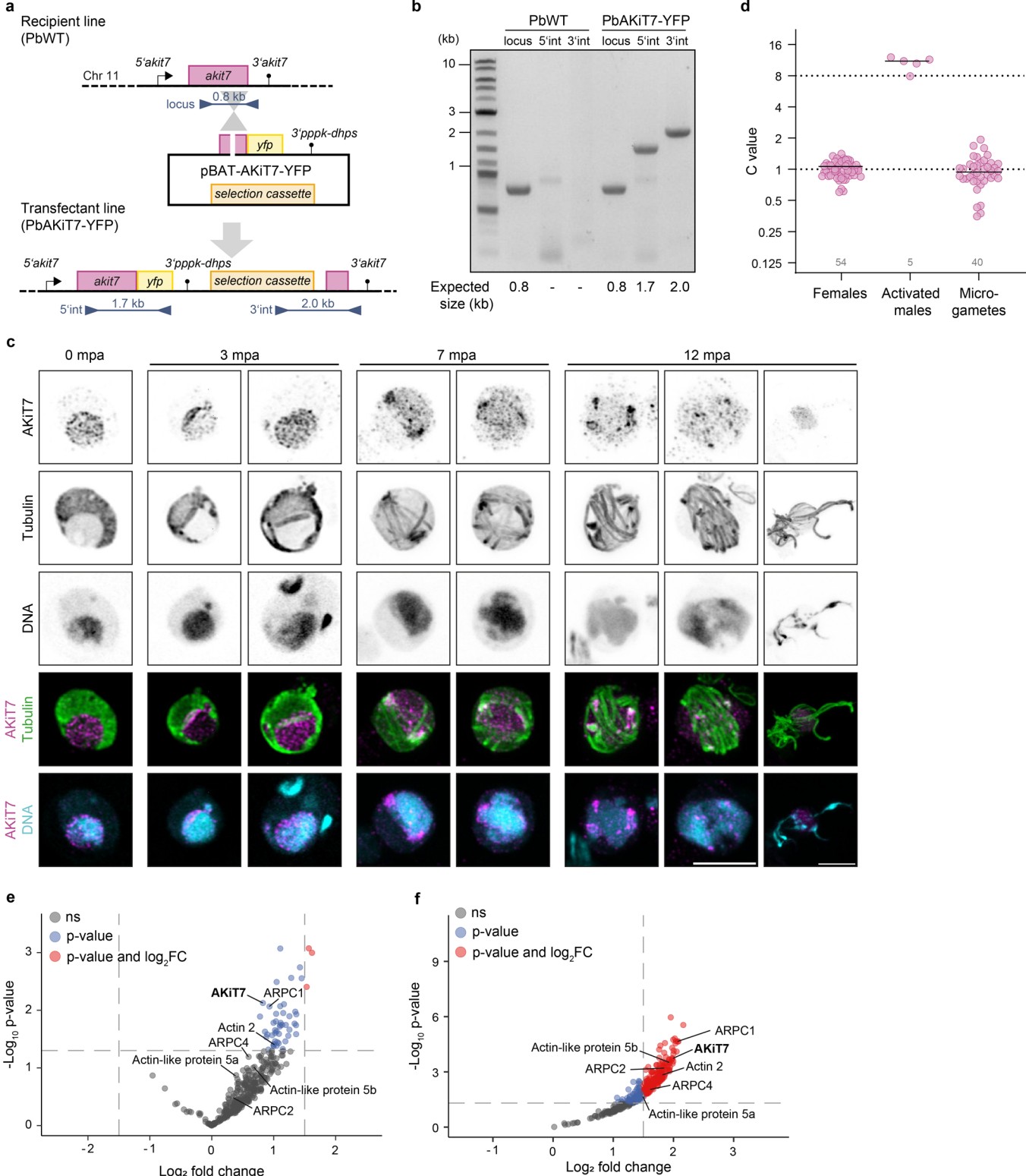

**Extended Data Fig. 9 | Generation and characterisation of AKiT7-YFP and identification of interaction partners. a**, Scheme of genetic strategy to generate PbAKiT7-YFPint. Primers (triangles) and expected amplicon sizes used for genotyping (see **b**) are indicated. Not drawn to scale. **b**, Genotyping PCR. The binding sites of the respective primers are indicated in **a**. **c**, Localisation of AKiT7-YFP in non-activated and activated gametocytes. 0-3 mpa: single slice. 7-12 mpa: maximum Z projection. Scale bars, 5 µm. Representative images from at least 8 images taken. **d**, DNA content of females, activated males, and microgametes of PbAKiT7-YFP. Values normalised to the mean DNA content of females imaged on the same slide. Dashed lines, expected C value of activated males (8) and microgametes (1). Pooled data from at least 2 independent experiments. Grey numbers above x axis indicate total numbers of cells analysed. **e-f**, Enriched proteins after pulldown of ARPC1-GFP from **e**, non-activated or **f**, activated gametocytes in comparison to GFP$_{con}$ gametocytes. Note the presence of all Arp2/3 complex proteins in **f**.

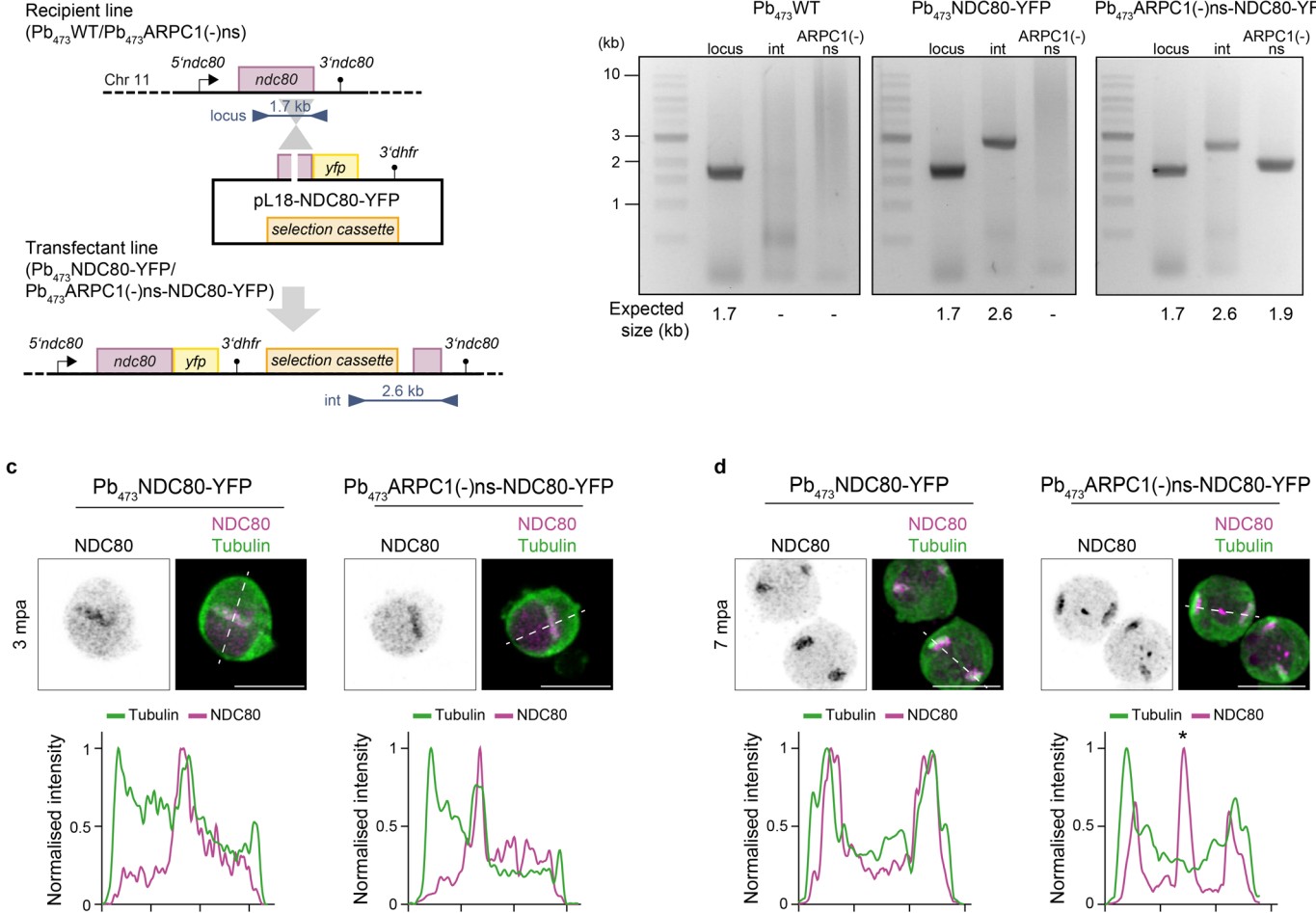

**Extended Data Fig. 10 | Localisation of NDC80-YFP during male gametogenesis. a**, Scheme of genetic strategy to generate Pb$_{473}$NDC80-YFP and Pb$_{473}$ARPC1(-)ns-NDC80-YFP. Primers (triangles) and expected amplicon sizes used for genotyping (see **b**) are indicated. Not drawn to scale. **b**, Genotyping PCRs. The binding sites of the respective primers are indicated in **a**. Binding sites for primers to detect the ARPC1 knockout locus are indicated in Extended Data Fig. 5a. The expected size of the product is indicated below the gel images.

**c,d**, Representative images and corresponding line plots of activated Pb$_{473}$NDC80-YFP and Pb$_{473}$ARPC1(-)ns-NDC80-YFP male gametocytes fixed at (**c**) 3 mpa or (**d**) 7 mpa. White dashed line indicates cross section used to generate the line plot. Intensity was normalized to minimal and maximal value per channel. Asterisk indicates NDC80 focus not associated with a tubulin spindle. **c**, single slice, **d**, Maximum *Z* projection. **c,d**, Representative images of at least 5 images taken per time point. Scale bar, 5 μm.

Dr. Franziska Hentzschel

# Reporting Summary

## Statistics

For all statistical analyses, confirm that the following items are present in the figure legend, table legend, main text, or Methods section.

| n/a | Confirmed | |
|---|---|---|
| ☐ | ☒ | The exact sample size (*n*) for each experimental group/condition, given as a discrete number and unit of measurement |
| ☐ | ☒ | A statement on whether measurements were taken from distinct samples or whether the same sample was measured repeatedly |
| ☐ | ☒ | The statistical test(s) used AND whether they are one- or two-sided<br>*Only common tests should be described solely by name; describe more complex techniques in the Methods section.* |
| ☒ | ☐ | A description of all covariates tested |
| ☐ | ☒ | A description of any assumptions or corrections, such as tests of normality and adjustment for multiple comparisons |
| ☐ | ☒ | A full description of the statistical parameters including central tendency (e.g. means) or other basic estimates (e.g. regression coefficient) AND variation (e.g. standard deviation) or associated estimates of uncertainty (e.g. confidence intervals) |
| ☐ | ☒ | For null hypothesis testing, the test statistic (e.g. *F*, *t*, *r*) with confidence intervals, effect sizes, degrees of freedom and *P* value noted<br>*Give P values as exact values whenever suitable.* |
| ☒ | ☐ | For Bayesian analysis, information on the choice of priors and Markov chain Monte Carlo settings |
| ☒ | ☐ | For hierarchical and complex designs, identification of the appropriate level for tests and full reporting of outcomes |
| ☒ | ☐ | Estimates of effect sizes (e.g. Cohen's *d*, Pearson's *r*), indicating how they were calculated |

*Our web collection on statistics for biologists contains articles on many of the points above.*

## Software and code

Policy information about availability of computer code

| | |
|---|---|
| Data collection | Flow cytometry: BD FACSDiva v8.0.1.1<br>Microscopy: Zeiss ZEN v2.6 |
| Data analysis | Most data was plotted and analysed in GraphPad Prism (v10.0.3), except for mass spectrometry data, which was analysed using Proteome Discoverer 2.5 and Perseus (v2.0.10) and plotted with R (v4.2.2). Flow cytometry data was analysed using FlowJo (v10.7.2). ZEN Blue 3.01 software was used for the post-2D or 3D Airyscan processing with automatically determined default Airyscan settings. All images were further processed in FIJI (v2.14.0). Structures were predicted using Alphafold3 and visualised using UCSF ChimeraX (1.9). EM tomograms were joined and reconstructed using IMOD (v4.11.12). |

For manuscripts utilizing custom algorithms or software that are central to the research but not yet described in published literature, software must be made available to editors and reviewers. We strongly encourage code deposition in a community repository (e.g. GitHub). See the Nature Portfolio guidelines for submitting code & software for further information.

## Data

Policy information about availability of data

All manuscripts must include a data availability statement. This statement should provide the following information, where applicable:
- Accession codes, unique identifiers, or web links for publicly available datasets
- A description of any restrictions on data availability
- For clinical datasets or third party data, please ensure that the statement adheres to our policy

> The Mus musculus and Plasmodium berghei protein databases that were used in this study can be accessed at Uniprot (UP000000589 and UP000074855, respectively). Raw mass spectrometry proteomics data have been deposited to the ProteomeXchange Consortium via the PRIDE partner repository82 with the dataset identifiers PXD046181, PXD051260 and PXD060997. All other data are available included in the main text orarticle and the supplementary information. Source data are provided with this paper. materials. Materials are available upon request.

## Research involving human participants, their data, or biological material

Policy information about studies with human participants or human data. See also policy information about sex, gender (identity/presentation), and sexual orientation and race, ethnicity and racism.

| | |
|---|---|
| Reporting on sex and gender | No research involving human participants, their data, or biological material |
| Reporting on race, ethnicity, or other socially relevant groupings | No research involving human participants, their data, or biological material |
| Population characteristics | No research involving human participants, their data, or biological material |
| Recruitment | No research involving human participants, their data, or biological material |
| Ethics oversight | No research involving human participants, their data, or biological material |

Note that full information on the approval of the study protocol must also be provided in the manuscript.

# Field-specific reporting

Please select the one below that is the best fit for your research. If you are not sure, read the appropriate sections before making your selection.

☒ Life sciences    ☐ Behavioural & social sciences    ☐ Ecological, evolutionary & environmental sciences

For a reference copy of the document with all sections, see nature.com/documents/nr-reporting-summary-flat.pdf

# Life sciences study design

All studies must disclose on these points even when the disclosure is negative.

| | |
|---|---|
| Sample size | No statistical method was used to predetermine sample size. Sample sizes were chosen based on previous experience with similar experiments allowing for robust replication while minimizing usage of rodents. The sample size for mouse experiments was thus 3-5 mice per group. For mosquitoes, 10-30 mosquitoes per timepoint were collected and analysed. For microscopy, 10-25 images per condition/replicate were collected and all cells on each image were quantitatively analysed. Exact sample size is provided in the figure legend of each experiment. |
| Data exclusions | No data was excluded. |
| Replication | Most experiments were performed at least in biological triplicates on different days. Selected experiments were performed in biological duplicates, as indicated in the respective figure legends. No unexpected variation between replicates was observed. |
| Randomization | Mice were age-matched and were allocated randomly to each group. All other samples were allocated in random to experimental groups. |
| Blinding | Given the nature of the experiments, it was not possible to blind investigators to mouse experimental groups during experimental assays and sample generation, as mouse infection and sample processing had to be handled by the same investigator. To avoid bias during quantitative data analysis as much as possible, microscopy images (with the exception of oocyst number and size determination) were blinded after acquisition and thus analysed in a single-blinded fashion. |

# Reporting for specific materials, systems and methods

We require information from authors about some types of materials, experimental systems and methods used in many studies. Here, indicate whether each material, system or method listed is relevant to your study. If you are not sure if a list item applies to your research, read the appropriate section before selecting a response.

## Materials & experimental systems

| n/a | Involved in the study |
|---|---|
| ☐ | ☒ Antibodies |
| ☐ | ☒ Eukaryotic cell lines |
| ☒ | ☐ Palaeontology and archaeology |
| ☐ | ☒ Animals and other organisms |
| ☒ | ☐ Clinical data |
| ☒ | ☐ Dual use research of concern |
| ☒ | ☐ Plants |

## Methods

| n/a | Involved in the study |
|---|---|
| ☒ | ☐ ChIP-seq |
| ☐ | ☒ Flow cytometry |
| ☒ | ☐ MRI-based neuroimaging |

# Antibodies

| | |
|---|---|
| Antibodies used | Immunofluorescence: rabbit-anti-GFP (1:50 dilution, Invitrogen, G10362), mouse-anti-tubulin (1:1000 dilution, Sigma-Aldrich, T5168-2ML), rat-anti-HA (1:1000 dilution, Roche, 11867423001), Alexa-546-anti-rabbit (Invitrogen, A11035), Alexa-488-anti-mouse (Invitrogen, A11029), Alexa-488-anti-rabbit (Invitrogen, A11008), Alexa-647-anti-rat (BD Pharma, 51-9006589), mouse-anti-PbHsp70 (clone 2E6 generated by Tsuji et al., PMID 8153120) |
| Validation | All antibodies used were standard commercial or previously published antibodies and have been validated by the respective manufacturer for specificity. In addition, for all immunofluorescence assays, appropriate controls (untagged wildtype cells or secondary antibody-only controls) were run in parallel to ensure specificity of the signal.

rabbit-anti-GFP (1:50 dilution, Invitrogen, G10362): Validated by manufacturer for use in immunofluorescence assays (https://www.thermofisher.com/antibody/product/GFP-Antibody-Recombinant-Monoclonal/G10362).

mouse-anti-tubulin (1:1000 dilution, Sigma-Aldrich, T5168-2ML): Validated by manufacturer for use in immunofluorescence assays (https://www.sigmaaldrich.com/DE/de/product/sigma/t5168). Used in multiple publications to stain Plasmodium tubulin (e.g. PMID 36154191 or PMID 37704606)

rat-anti-HA (1:1000 dilution, Roche, 11867423001): Validated by manufacturer for use in immunofluorescence assays (https://www.sigmaaldrich.com/DE/de/product/roche/roahaha)

Alexa-546-anti-rabbit (Invitrogen, A11035): Validated by manufacturer for use in immunofluorescence assays (https://www.thermofisher.com/antibody/product/Goat-anti-Rabbit-IgG-H-L-Highly-Cross-Adsorbed-Secondary-Antibody-Polyclonal/A-11035)

Alexa-488-anti-mouse (Invitrogen, A11029): Validated by manufacturer for use in immunofluorescence assays (https://www.thermofisher.com/antibody/product/Goat-anti-Mouse-IgG-H-L-Highly-Cross-Adsorbed-Secondary-Antibody-Polyclonal/A-11029)

Alexa-488-anti-rabbit (Invitrogen, A11008): Validated by manufacturer for use in immunofluorescence assays (https://www.thermofisher.com/antibody/product/Goat-anti-Rabbit-IgG-H-L-Cross-Adsorbed-Secondary-Antibody-Polyclonal/A-11008)

Alexa-647-anti-rat (BD Pharma, 51-9006589): Validated internally by using wildtype cells not expressing the HA - Tag against which the primary antibody was directed. No unspecific signal was observed.

mouse-anti-PbHsp70 (clone 2E6): Validated by Tsuji et al., PMID 8153120, and used in >150 citations. |

# Eukaryotic cell lines

Policy information about cell lines and Sex and Gender in Research

| | |
|---|---|
| Cell line source(s) | Human HepG2, source ATCC, Identifier HB-8065 |
| Authentication | None of the cell lines were authenticated |
| Mycoplasma contamination | Cell lines were not tested for mycoplasma contamination |
| Commonly misidentified lines (See ICLAC register) | No commonly misidentified lines were used in this study. |

# Animals and other research organisms

Policy information about studies involving animals; ARRIVE guidelines recommended for reporting animal research, and Sex and Gender in Research

| | |
|---|---|
| Laboratory animals | Mice (Mus musculus), outbred: Theiler original or CD1, inbred: C57Bl/G ; female; 5-8 weeks old. Mice were kept in groups of 2 to 4 |

| Laboratory animals | mice per cage under specified pathogen-free (SPF) conditions within the animal facilities at Heidelberg University or University of Glasgow on a 12 hour light/dark cycle at 22 °C (± 2 °C) and 50-60% humidity with ad libitum access to food and water. |
| Wild animals | The study did not involve wild animals |
| Reporting on sex | All mice were female, a commonly used model system to study rodent Plasmodium species. The sex of the mice has no impact on the cell biology of the parasite. |
| Field-collected samples | The study did not involve samples collected from the field |
| Ethics oversight | All experiments were performed in accordance with GV-SOLAS and FELASA guidelines and have been approved by the German authorities (Regierungspräsidium Karlsruhe) or according to the guidelines defined by the Home Office and UK Animals (Scientific Procedures) Act 1986 and approved by the UK Home Office (project license P6CA91811) and the University of Glasgow animal welfare and ethical review body. |

Note that full information on the approval of the study protocol must also be provided in the manuscript.

## Plants

| Seed stocks | *Report on the source of all seed stocks or other plant material used. If applicable, state the seed stock centre and catalogue number. If plant specimens were collected from the field, describe the collection location, date and sampling procedures.* |
| Novel plant genotypes | *Describe the methods by which all novel plant genotypes were produced. This includes those generated by transgenic approaches, gene editing, chemical/radiation-based mutagenesis and hybridization. For transgenic lines, describe the transformation method, the number of independent lines analyzed and the generation upon which experiments were performed. For gene-edited lines, describe the editor used, the endogenous sequence targeted for editing, the targeting guide RNA sequence (if applicable) and how the editor was applied.* |
| Authentication | *Describe any authentication procedures for each seed stock used or novel genotype generated. Describe any experiments used to assess the effect of a mutation and, where applicable, how potential secondary effects (e.g. second site T-DNA insertions, mosiacism, off-target gene editing) were examined.* |

## Flow Cytometry

### Plots

Confirm that:

☒ The axis labels state the marker and fluorochrome used (e.g. CD4-FITC).

☒ The axis scales are clearly visible. Include numbers along axes only for bottom left plot of group (a 'group' is an analysis of identical markers).

☒ All plots are contour plots with outliers or pseudocolor plots.

☒ A numerical value for number of cells or percentage (with statistics) is provided.

### Methodology

| Sample preparation | For asexual growth rate and gametocyte production: A drop of blood was collected from the tip of the tail of an infected mouse and stained in 1 ml DRAQ5 (1:1000 diluted in PBS) for 10 min at room temperature. Cells were washed once with PBS before analysis by flow cytometry.<br>For detection of ARPC1-HA: 100 µl blood were collected from infected mice and fixed by adding equal volume of 4% PFA/0.0075% Glutaraldehyde. Cells were stained according to standard immunofluorescence staining procedures with an antibody against HA (Roche, 11867423001) and GFP (Invitrogen, G10362), and with Hoechst at a final concentration of 10 µg/ml. |
| Instrument | For asexual growth rate and gametocyte production: MACSQuant VYB flow cytometer<br>For detection of ARPC1-HA: BD FACSCelesta cell analyzer |
| Software | FACS Diva Software<br>FlowJo V 10.7.2 |
| Cell population abundance | No sorting was done. |
| Gating strategy | For all gating strategies, cells were initially gated on the main population in the FSC/SSC plot to exclude small debris and very large cells, followed by gating on single cells as identified by being on the diagonal of the FSC-H/FSC-A plot. For detection of asexual growth rate and gametocyte production, DRAQ5-positive cells (i.e. all parasites) were further gated on mCherry and GFP (i.e. female and male gametocytes respectively). For quantification of ARPC1-HA expression, parasites were identified by Hoechst-Staining and then gated on mCherry and GFP (i.e. female and male gametocytes, respectively). HA signal was quantified based on median Alexa647 fluorescence intensity in those populations. |

☒ Tick this box to confirm that a figure exemplifying the gating strategy is provided in the Supplementary Information.

