## [Peer Review File · Nature Microbiology]

An atypical Arp2/3 complex is required for Plasmodium DNA segregation and malaria transmission.

Corresponding Author: Dr Franziska Hentzschel

Version 0:

Reviewer comments:

Reviewer #1

(Remarks to the Author)

Review of Hentzschel et al. "A non-canonical Arp2/3 complex is essential for Plasmodium DNA segregation and transmission of malaria".

In this paper, Hentzschel et al lead off with the study of a protein, ARPC1, that eventually leads to identification of a minimal arp2/3 complex (5 units rather than the canonical 7). The study elegantly links the function of this complex to DNA segregation in Plasmodium gamete formation using a suite of cell and genetic techniques. The study is formidable to say the least (I applaud the authors!), extensive and convincing in scope – it challenges a dogma (no Arp2/3 in Apicomplexa!) and in doing so opens up a new vista of biology linking actin and the fascinating world of microgametogenesis. I would certainly expect its publication and believe it will be very favourably received by the microbial cell biology community. My comments below are purposefully pedantic, as I feel a piece of work this high quality deserves to be poured over with a fine-tooth comb. I wouldn't want the authors to think, however, I was being overly negative. Below I've made some comments/suggested changes that I would ask are considered, though this should not preclude publication of the work.

Comments:

1. The male specific phenotype of the ARPC1 KO was reported in the preprint by Sayers et al. 2023: doi: <https://doi.org/10.1101/2023.12.25.572958>, Table S1). Acknowledging the studies were posted at similar times, this is not a suggestion that one preceded or scooped the other, however, it might be good to acknowledge this in the paper – that observations are in line with a genetic screen for fertility genes in Plasmodium. This is also true for Alp5a (PBANKA_0811800), adding further weight to the present study.
2. I found the images in Extended Data Fig. 2c a little underwhelming. With such amazing images throughout the rest of the paper, the presence of a smudge of staining in gametocytes and ookinetes doesn't do justice to the initial "Ah Ha!" moment of ARPC1 being nuclear in just these stages.
3. A minor point in Extended Data Fig. 2, the first column is labelled 'asexuals', with the next column labelled 'schizonts' (also asexual stages). The first column should perhaps be relabelled to trophs or rings?
4. A question. Beginning on line 123 the authors describe crossing ARPC1 knockout mutants with female deficient and male deficient lines, with only the female-deficient lines resulting in normal sized oocysts. This is taken as evidence that "ARPC1 is thus required for male, but not female fertility." Do the authors feel they can fully make this claim if they haven't shown these smaller oocysts (from the male deficient cross) don't produce sporozoites? I.e. as previously shown with the ARPC1 knockout oocysts (Fig. 1b).
5. In Fig. 1k – the statistical impact of this flow data isn't clear to me - I can't seem to locate how this data was obtained? Detail on the number of replicates, statistical significance of it etc is missing. I'd suggest this should be quantified/cleared up. Related, and mirroring point 3, this also raises the unanswered question of why ARPC1 would be present at all in the female gametocyte? Is this just because of timing/shared developmental pathways, or do the authors think ARPC1 may nonetheless have a role in the macrogamete. The data presented make it unclear whether this should be an avenue of investigation or would be a dead end to look down. A comment on this might be worthwhile but left unanswered it feels confusing.
6. Line 171 is perhaps a bit too strong from the evidence provided. Perhaps something like "contributes to" would be more appropriate than "is responsible for" (very definitive singular-sounding statement).
7. Similarly, on line 188, it is not conclusively shown that "ARPC1 is essential for DNA segregation during male gametogenesis", though the data certainly suggests that it influences DNA segregation.

8. Another key observation that I feel is not definitively demonstrated that I suggest is changed, is whether the minimal arp2/3 complex is actually a nucleator. It associates/co-localises with F-actin, it's disruption and the disruption of F-actin phenocopy each other, but this doesn't prove it is a nucleator. Indeed, other actin nucleators [formins] may be present (is MISFIT a nucleator?). Therefore, I feel the statement that it is a nucleator should be tempered to "may be" or "consistent with it being". Definitive proof as a nucleator either requires biochemical validation or the slightly less satisfying experiment where absence of the complex is associated with absence of F-actin. Slightly more convincing, but biochemical would be best. Minimally, I'd suggest tempering the statements (e.g. line 341 – emphatic statement it nucleates actin...).

9. A question: Is there any evidence from imaging (here or elsewhere by the group) of branched actin filaments in gametes/gametocytes after activation? This would be very strong evidence for the complex retaining arp2/3 like nucleator features. In their absence, arp2/3 may actually be an F-actin binding complex (nucleated by a different nucleator – see above) that builds on F-actin present creating the link to the kinetochore etc.

10. In the discussion, line 410, in reference to the puzzling "delayed-death" oocyst phenotype it might be worth saying that this phenomenon is observed with other genes that function in microgametogenesis, e.g. as exemplified by the SUN1-KO (Sayers et al. 2023 preprint, see above) and EBI-KO (Zeeshan et al. 2023, <https://doi.org/10.1038/s41467-023-41395-3>) parasites.

11. Methods (line 508): minor question, to my understanding, PlasmoGEM vectors are typically digested with NotI – the authors say they used NdeI – is this correct?

JB 04/JUN/2024

Reviewer #2

(Remarks to the Author)

Very interesting work. The discovery of an aberrant Arp2/3 complex in Plasmodium is extremely interesting. The novel phenotypes are convincing.

It's not sound yet though. Most importantly - the authors are playing both sides. They're saying that just searching databases is not strong enough to find Arp2/3 complex members as evolutionarily distant as they are in Plasmodium; but they seem to be concluding that two complex members are missing without an exhaustive enough search to justify it that strongly. The title, for example, uses "non-canonical" but it's not proven that the missing two subunits aren't there - they just didn't find anything.

The use of "essential" in the title is a bit over-egged too, the data are more qualified.

My suggestion is moderate the language, to make it clearer what is proven and/or absolute and what is not, and also try more in-depth tests (or argue more cogently...) about the non-canonical nature of the targets.

It's also very densely written in parasite language. I'm an actin biologist and found a lot of the logic difficult. This may be the style and the word limits, but it would be good to try and better explain why the plasmodium-specific experiments lead to the actin conclusions.

But - returning to my starting line - a form of this should be published expeditiously.

=====

Slightly more detail:

- L50 Cite primary literature. Mullins and Machesky, not these very wordy reviews. Also this line has a typo.

- L75 I am not at all sure how "tagging ARPC1 with GFP" allows this conclusion. Either explain better or do a better experiment. Is the GFP knocked directly into the genome so the normal expression is accurately retained? Do the authors know this, and is it accepted in the Plasmodium field? It wouldn't be in mammals/Drosophila/elegans/Dictyostelium.

- L214 I really do not understand this argument. Are the authors trying to say the complex only assembles in activated cells? That's completely unlike what is seen in other organisms. Or are they trying to say the IP doesn't work because there isn't any? It goes back to the issues with the GFP transfection; there may be field-specific meanings but they need a clearer airing.

- L275 the thing I really want colabfold or alphafold 2 to answer is, is this proposed complex structurally reasonable & energetically possible? In particular what does the surface where the missing canonical subunits should be look like? I suspect the standard ("canonical") Arp2/3 complex would not survive loss of 2 subunits because the solution energetics of the exposed hydrophobic surfaces would be awful. Can they model a 5-member complex and show it can work? If you compare it with a canonical one, has there been a clear stabilising change where the other 2 subunits should bind?

- L290 this seems to be using the same result both ways. Is CK666 supposed to bind and work or not? I don't understand what this is trying to say at all.

- L420 also trying to argue both ways – it's noncanonical but none of the subunits were previously found by searching - why so certain the other two components aren't there? ARPC4 is already in the mix – are the other subunits? The "noncanonical" seems to be overreaching. Particularly as L425 "Similar hybrid Arp2/3 complexes have been described to form at points of focal adhesion" they aren't similar at all - they lack p40/p41, the one really conserved member in the Plasmodium, and as far as I can see ARPC4 is in the Chorev complex but not in the one reported here. It's very loose to say two partial complexes are "similar" if they are asserted to lack different subunits (plus as far as I know the Chorev paper has not led to much, either; and the citation of

a Trends article instead of primarily literature is not really appropriate to back up this point).

So overall - I am very keen on the reporting of the identified subunits of the complex and the very interesting roles in segregation and infection; but I'd like to see much more moderation and scholarship in the assessment of whether the complex is truly partial. Some combination of more support and more careful reporting would sort this out.

Reviewer #3

(Remarks to the Author)

In this paper, the authors explore the role of the Plasmodium Arp2/3 complex. In other eukaryotes, this complex plays a major role in the regulation of the actin cytoskeleton. However, its role in Plasmodium is unknown. Using AlphaFold, they first identify ARPC1 as a subunit of a non-canonical Plasmodium Arp2/3 complex. Here, they show that the protein is mainly expressed in male microgametocytes where it shows dynamic localisation around the mitotic spindles. Deletion of *arpc1* leads to the formation of sub-haploid male gametes leading to a complete block in parasite transmission. They further identify five other components of the complex including evolutionary conserved proteins and more apicomplexan-specific proteins. Interestingly, one of these proteins has previously been identified in immunoprecipitations of kinetochore proteins in Plasmodium gametocytes. They also show presence of presence of F-actin in the nucleus during male gametogenesis. As cytochalasin D, an inhibitor of actin polymerisation led to a similar phenotype to that of parasites lacking components of the Arp2/3 complex, the authors propose that the Arp2/3 complex establishes a link between kinetochores and actin II filaments to allow DNA incorporation into the forming male gamete.

I find the paper well written, the experimental design robust and the experiments performed to a high standard.

I however see two major limitations to this study that include i) the robustness of the link between the mitotic spindle, the kinetochores and the ARP2/3 complex and ii) the detailed investigation of the nature of the nuclear actin filaments.

Major comments

1. The link between the Arp2/3 complex with the kinetochore and mitotic spindle

- The link between the Arp2/3 complex with the kinetochores and their attachment to the mitotic spindle requires more attention. The assumption that Akit7 links kinetochores with an actin filament-bound Arp2/3 complex is mainly based and the co-immunoprecipitation of Akit7 with ARPC1 and 2. However Akit7 seemed to be an atypical component of the kinetochore during male gametogenesis in PMID36006241. In the absence of other kinetochore components in immunoprecipitations of Arp2/3 components or reciprocal immunoprecipitation or functional analysis of Akit7, there is a risk that that the author hypothesis on the role of Akit7 is not correct.

-In the same line, the phenotype shown in figure 5a is unclear. What is the exact fate of kinetochores upon ARPC1 deletion? With shown resolution, the Ndc80-YFP signal in absence of ARPC1 is difficult to understand. Are the kinetochores detached from the spindle, stuck onto the spindle and/or unstable in the absence of ARPC1? In the two later cases, this would indicate a different or additional role for the Arp2/3 complex. A phenotyping with a higher resolution such as EM or expansion microscopy may help to address this important issue.

2. The nature and role of nuclear actin filaments during male gametogenesis

- The actin filaments/signal should be localised with a higher resolution. On the images shown here, ARPC1 or the signal from the F-Actin chromobodies does not fully localise to the mitotic spindle and shows a slightly wider distribution. It is also unclear to me where the actin filaments are anchored to and how would this allow tethering of the kinetochores to the microtubules or kinetochore stability during mitosis?

- How does the nuclear actin signal look like in the absence of components of the ARP2/3 complex? Similarly, how does cytochalasin D affect kinetochore assembly/motility along the mitotic spindle mitosis and/or direct incorporation of DNA into the forming gamete? I think that answer these questions are important to support a link between the actin cytoskeleton, the Arp2/3 complex and kinetochores.

Minor comments

I would suggest the authors to be clearer when they use the term DNA/kinetochore segregation. Do they mean segregation along the mitotic spindle or integration into the forming gamete? While the first option likely leads to the second, both processes are nevertheless distinct. As far as I understand both possibilities are alternatively mentioned using the same wording, which makes it difficult to understand the observed phenotypes.

I would suggest the authors to indicate that ARPC1 or 2 interact with a kinetochore component not necessarily with kinetochores

L135 closed mitosis is not a synonymous for endomitosis.

L186 spindle pole body is a terminology used in yeast to describe a different structure. Please avoid using this terminology to avoid any confusion.

Fig 3 despite the figure legend, I do not think that the figure shows a role of ARPC1 in kinetochore-spindle attachment.

Decision Letter:

16th July 2024

Dear Franziska and Freddy,

Thank you for your patience while your manuscript "A non-canonical Arp2/3 complex is essential for Plasmodium DNA segregation and transmission of malaria." was under peer-review at Nature Microbiology. It has now been seen by 3 referees, whose expertise and comments you will find at the end of this email. Although they find your work of some potential interest, they have raised a number of concerns that will need to be addressed before we can consider publication of the work in Nature Microbiology.

In particular, referee #1, while very positive about the study, has some concerns over the insufficient quality of the images in Extended Data Fig. 2c. Referee #2 feels that some of the statements and conclusions reached are currently still contradictory. The referee suggests to moderate the language, to make it clearer what is proven and what is not, and also try more in-depth tests about the non-canonical nature of the targets. Referee #2 also feels that the manuscript could be made more accessible to a broader readership (beyond parasitology experts). Referee #3 states that while the general role of the Arp2/3 complex in male gametogony is well demonstrated, the exact function of the complex at the cellular and molecular levels is unclear. In particular the referee feels that the link between the mitotic spindle, the kinetochores, the Arp2/3 and possible actin filaments is weak, the weakest points being the link between the kinetochore and the Arp2/3 complex and the fate of kinetochores in the absence of the Arp2/3 complex. Furthermore, referee #3 says that the nature of the actin filaments (if any) is unclear and suggests that higher resolution imaging may help to address these issues.

Should further experimental data allow you to address these criticisms, we would be happy to look at a revised manuscript.

Please also note that for a potential second round of review we would also invite an additional referee with expertise in computational biology to specifically assess the AlphaFold-related aspects of the study.

Please include a data availability statement as a separate section after Methods but before references, under the heading "Data Availability". This section should inform readers about the availability of the data used to support the conclusions of your study. This information includes accession codes to public repositories (data banks for protein, DNA or RNA sequences, microarray, proteomics data etc...), references to source data published alongside the paper, unique identifiers such as URLs to data repository entries, or data set DOIs, and any other statement about data availability. At a minimum, you should include the following statement: "The data that support the findings of this study are available from the corresponding author upon request", mentioning any restrictions on availability. If DOIs are provided, we also strongly encourage including these in the Reference list (authors, title, publisher (repository name), identifier, year). For more guidance on how to write this section please see: <http://www.nature.com/authors/policies/data/data-availability-statements-data-citations.pdf>

* If you have not done so already we suggest that you begin to revise your manuscript so that it conforms to our Article format instructions at <http://www.nature.com/nmicrobiol/info/final-submission>. Refer also to any guidelines provided in this letter.

When submitting the revised version of your manuscript, please pay close attention to our [href="https://www.nature.com/nature-portfolio/editorial-policies/image-integrity">Digital Image Integrity Guidelines](https://www.nature.com/nature-portfolio/editorial-policies/image-integrity) and to the following points below:

Link Redacted

Note: This url links to your confidential homepage and associated information about manuscripts you may have submitted or be reviewing for us. If you wish to forward this e-mail to co-authors, please delete this link to your homepage first.

Nature Microbiology is committed to improving transparency in authorship. As part of our efforts in this direction, we are now requesting that all authors identified as 'corresponding author' on published papers create and link their Open Researcher and Contributor Identifier (ORCID) with their account on the Manuscript Tracking System (MTS), prior to acceptance. This applies to primary research papers only. ORCID helps the scientific community achieve unambiguous attribution of all scholarly contributions. You can create and link your ORCID from the home page of the MTS by clicking on 'Modify my Springer Nature account'. For more information please visit www.springernature.com/orcid.

If you wish to submit a suitably revised manuscript we would hope to receive it within 6 months. If you cannot send it within this time, please let us know. We will be happy to consider your revision, even if a similar study has been accepted for publication at Nature Microbiology or published elsewhere (up to a maximum of 6 months).

Yours sincerely,

Reviewer Expertise:

Referee #1: Malaria, Plasmodium biology

Referee #2: Arp2/3 complex, microscopy

Referee #3: Plasmodium cell division

Reviewer Comments:

Reviewer #1 (Remarks to the Author):

Review of Hentschel et al. "A non-canonical Arp2/3 complex is essential for Plasmodium DNA segregation and transmission of malaria".

In this paper, Hentschel et al lead off with the study of a protein, ARPC1, that eventually leads to identification of a minimal arp2/3 complex (5 units rather than the canonical 7). The study elegantly links the function of this complex to DNA segregation in Plasmodium gamete formation using a suite of cell and genetic techniques. The study is formidable to say the least (I applaud the authors!), extensive and convincing in scope – it challenges a dogma (no Arp2/3 in Apicomplexa!) and in doing so opens up a new vista of biology linking actin and the fascinating world of microgametogenesis. I would certainly expect its publication and believe it will be very favourably received by the microbial cell biology community. My comments below are purposefully pedantic, as I feel a piece of work this high quality deserves to be poured over with a fine-tooth comb. I wouldn't want the authors to think, however, I was being overly negative. Below I've made some comments/suggested changes that I would ask are considered, though this should not preclude publication of the work.

Comments:

1. The male specific phenotype of the ARPC1 KO was reported in the preprint by Sayers et al. 2023: doi: <https://doi.org/10.1101/2023.12.25.572958>, Table S1). Acknowledging the studies were posted at similar times, this is not a suggestion that one preceded or scooped the other, however, it might be good to acknowledge this in the paper – that observations are in line with a genetic screen for fertility genes in Plasmodium. This is also true for Alp5a (PBANKA_0811800), adding further weight to the present study.
2. I found the images in Extended Data Fig. 2c a little underwhelming. With such amazing images throughout the rest of the paper, the presence of a smudge of staining in gametocytes and ookinetes doesn't do justice to the initial "Ah Ha!" moment of ARPC1 being nuclear in just these stages.
3. A minor point in Extended Data Fig. 2, the first column is labelled 'asexuals', with the next column labelled 'schizonts' (also asexual stages). The first column should perhaps be relabelled to trophs or rings?

4. A question. Beginning on line 123 the authors describe crossing ARPC1 knockout mutants with female deficient and male deficient lines, with only the female-deficient lines resulting in normal sized oocysts. This is taken as evidence that “ARPC1 is thus required for male, but not female fertility.” Do the authors feel they can fully make this claim if they haven’t shown these smaller oocysts (from the male deficient cross) don’t produce sporozoites? I.e. as previously shown with the ARPC1 knockout oocysts (Fig. 1b).
5. In Fig. 1k – the statistical impact of this flow data isn’t clear to me - I can’t seem to locate how this data was obtained? Detail on the number of replicates, statistical significance of it etc is missing. I’d suggest this should be quantified/cleared up. Related, and mirroring point 3, this also raises the unanswered question of why ARPC1 would be present at all in the female gametocyte? Is this just because of timing/shared developmental pathways, or do the authors think ARPC1 may nonetheless have a role in the macrogamete. The data presented make it unclear whether this should be an avenue of investigation or would be a dead end to look down. A comment on this might be worthwhile but left unanswered it feels confusing.
6. Line 171 is perhaps a bit too strong from the evidence provided. Perhaps something like “contributes to” would be more appropriate than “is responsible for” (very definitive singular-sounding statement).
7. Similarly, on line 188, it is not conclusively shown that “ARPC1 is essential for DNA segregation during male gametogenesis”, though the data certainly suggests that it influences DNA segregation.
8. Another key observation that I feel is not definitively demonstrated that I suggest is changed, is whether the minimal arp2/3 complex is actually a nucleator. It associates/co-localises with F-actin, it’s disruption and the disruption of F-actin phenocopy each other, but this doesn’t prove it is a nucleator. Indeed, other actin nucleators [formins] may be present (is MISFIT a nucleator?). Therefore, I feel the statement that it is a nucleator should be tempered to “may be” or “consistent with it being”. Definitive proof as a nucleator either requires biochemical validation or the slightly less satisfying experiment where absence of the complex is associated with absence of F-actin. Slightly more convincing, but biochemical would be best. Minimally, I’d suggest tempering the statements (e.g. line 341 – emphatic statement it nucleates actin...).
9. A question: Is there any evidence from imaging (here or elsewhere by the group) of branched actin filaments in gametes/gametocytes after activation? This would be very strong evidence for the complex retaining arp2/3 like nucleator features. In their absence, arp2/3 may actually be an F-actin binding complex (nucleated by a different nucleator – see above) that builds on F-actin present creating the link to the kinetochore etc.
10. In the discussion, line 410, in reference to the puzzling “delayed-death” oocyst phenotype it might be worth saying that this phenomenon is observed with other genes that function in microgametogenesis, e.g. as exemplified by the SUN1-KO (Sayers et al. 2023 preprint, see above) and EBI-KO (Zeeshan et al. 2023, <https://doi.org/10.1038/s41467-023-41395-3>) parasites.
11. Methods (line 508): minor question, to my understanding, PlasmoGEM vectors are typically digested with NotI – the authors say they used NdeI – is this correct?

JB 04/JUN/2024

Reviewer #2 (Remarks to the Author):

Very interesting work. The discovery of an aberrant Arp2/3 complex in Plasmodium is extremely interesting. The novel phenotypes are convincing.

It's not sound yet though. Most importantly - the authors are playing both sides. They're saying that just searching databases is not strong enough to find Arp2/3 complex members as evolutionarily distant as they are in Plasmodium; but they seem to be concluding that two complex members are missing without an exhaustive enough search to justify it that strongly. The title, for example, uses "non-canonical" but it's not proven that the missing two subunits aren't there - they just didn't find anything.

The use of "essential" in the title is a bit over-egged too, the data are more qualified.

My suggestion is moderate the language, to make it clearer what is proven and/or absolute and what is not, and also try more in-depth tests (or argue more cogently...) about the non-canonical nature of the targets.

It's also very densely written in parasite language. I'm an actin biologist and found a lot of the logic difficult. This may be the style and the word limits, but it would be good to try and better explain why the plasmodium-specific experiments lead to the actin conclusions.

But - returning to my starting line - a form of this should be published expeditiously.

=====

Slightly more detail:

- L50 Cite primary literature. Mullins and Machesky, not these very wordy reviews. Also this line has a typo.

- L75 I am not at all sure how "tagging ARPC1 with GFP" allows this conclusion. Either explain better or do a better experiment. Is the GFP knocked directly into the genome so the normal expression is accurately retained? Do the authors know this, and is it accepted in the Plasmodium field? It wouldn't be in mammals/Drosophila/elegans/Dictyostelium.

- L214 I really do not understand this argument. Are the authors trying to say the complex only assembles in activated cells? That's completely unlike what is seen in other organisms. Or are they trying to say the IP doesn't work because there isn't any? It goes back to the issues with the GFP transfection; there may be field-specific meanings but they need a clearer airing.

- L275 the thing I really want colabfold or alphafold 2 to answer is, is this proposed complex structurally reasonable & energetically possible? In particular what does the surface where the missing canonical subunits should be look like? I suspect the standard ("canonical") Arp2/3 complex would not survive loss of 2 subunits because the solution energetics of the exposed hydrophobic surfaces would be awful. Can they model a 5-member complex and show it can work? If you compare it with a canonical one, has there been a clear stabilising change where the other 2 subunits should bind?

- L290 this seems to be using the same result both ways. Is CK666 supposed to bind and work or not? I don't understand what this is trying to say at all.

- L420 also trying to argue both ways – it's noncanonical but none of the subunits were previously found by searching - why so certain the other two components aren't there? ARPC4 is already in the mix – are the other subunits? The "noncanonical" seems to be overreaching. Particularly as L425 "Similar hybrid Arp2/3 complexes have been described to form at points of focal adhesion" they aren't similar at all - they lack p40/p41, the one really conserved member in the Plasmodium, and as far as I can see ARPC4 is in the Chorev complex but not in the one reported here. It's very loose to say two partial complexes are "similar" if they are asserted to lack different subunits (plus as far as I know the Chorev paper has not led to much, either; and the citation of a Trends article instead of primary literature is not really appropriate to back up this point).

So overall - I am very keen on the reporting of the identified subunits of the complex and the very interesting roles in segregation and infection; but I'd like to see much more moderation and scholarship in the assessment of whether the complex is truly partial. Some combination of more support and more careful reporting would sort this out.

Reviewer #3 (Remarks to the Author):

In this paper, the authors explore the role of the Plasmodium Arp2/3 complex. In other eukaryotes, this complex plays a major role in the regulation of the actin cytoskeleton. However, its role in Plasmodium is unknown. Using AlphaFold, they first identify ARPC1 as a subunit of a non-canonical Plasmodium Arp2/3 complex. Here, they show that the protein is mainly expressed in male microgametocytes where it shows dynamic localisation around the mitotic spindles. Deletion of *arp1* leads to the formation of sub-haploid male gametes leading to a complete block in parasite transmission. They further identify five other components of the complex including evolutionary conserved proteins and more apicomplexan-specific proteins. Interestingly, one of these proteins has previously been identified in immunoprecipitations of kinetochore proteins in Plasmodium gametocytes. They also show presence of presence of F-actin in the nucleus during male gametogenesis. As cytochalasin D, an inhibitor of actin polymerisation led to a similar phenotype to that of parasites lacking components of the Arp2/3 complex, the authors propose that the Arp2/3 complex establishes a link between kinetochores and actin II filaments to allow DNA incorporation into the forming male gamete.

I find the paper well written, the experimental design robust and the experiments performed to a high standard.

I however see two major limitations to this study that include i) the robustness of the link between the mitotic spindle, the kinetochores and the ARP2/3 complex and ii) the detailed investigation of the nature of the nuclear actin filaments.

Major comments

1. The link between the Arp2/3 complex with the kinetochore and mitotic spindle

- The link between the Arp2/3 complex with the kinetochores and their attachment to the mitotic spindle requires more attention. The assumption that Akit7 links kinetochores with an actin filament-bound Arp2/3 complex is mainly based and the co-immunoprecipitation of Akit7 with ARPC1 and 2. However Akit7 seemed to be an atypical component of the kinetochore during male gametogenesis in PMID36006241. In the absence of other kinetochore components in immunoprecipitations of Arp2/3 components or reciprocal immunoprecipitation or functional analysis of Akit7, there is a risk that that the author hypothesis on the role of Akit7 is not correct.

-In the same line, the phenotype shown in figure 5a is unclear. What is the exact fate of kinetochores upon ARPC1 deletion? With shown resolution, the Ndc80-YFP signal in absence of ARPC1 is difficult to understand. Are the kinetochores detached from the spindle, stuck onto the spindle and/or unstable in the absence of ARPC1? In the two later cases, this would indicate a different or additional role for the Arp2/3 complex. A phenotyping with a higher resolution such as EM or expansion microscopy may help to address this important issue.

2. The nature and role of nuclear actin filaments during male gametogenesis

- The actin filaments/signal should be localised with a higher resolution. On the images shown here, ARPC1 or the signal from the F-Actin chromobodies does not fully localise to the mitotic spindle and shows a slightly wider distribution. It is also unclear to me where the actin filaments are anchored to and how would this allow tethering of the kinetochores to the microtubules or kinetochore stability during mitosis?

- How does the nuclear actin signal look like in the absence of components of the ARP2/3 complex? Similarly, how does cytochalasin D affect kinetochore assembly/motility along the mitotic spindle mitosis and/or direct incorporation of DNA into the forming gamete? I think that answer these questions are important to support a link between the actin cytoskeleton, the Arp2/3 complex and kinetochores.

Minor comments

I would suggest the authors to be clearer when they use the term DNA/kinetochore segregation. Do they mean segregation along the mitotic spindle or integration into the forming gamete? While the first option likely leads to the second, both processes are nevertheless distinct. As far as I understand both possibilities are alternatively mentioned using the same wording, which makes it difficult to understand the observed phenotypes.

I would suggest the authors to indicate that ARPC1 or 2 interact with a kinetochore component not necessarily with kinetochores

L135 closed mitosis is not a synonymous for endomitosis.

L186 spindle pole body is a terminology used in yeast to describe a different structure. Please avoid using this terminology to avoid any confusion.

Fig 3 despite the figure legend, I do not think that the figure shows a role of ARPC1 in kinetochore-spindle attachment.

Version 1:

Reviewer comments:

Reviewer #1

(Remarks to the Author)

I have read the extensive rebuttal, additional data and changes to the manuscript. I am satisfied that the authors have address my specific comments as well as those of other reviewers. The paper is clearly the beginning of a lot more work to come, which I look forward to seeing in time. I have no further requested changes/comments.

Reviewer #2

(Remarks to the Author)

The style of the paper is much better, it's much clearer, the alphafold section is neat, but the authors have not quite got the message. In the abstract they cannot say "Here we discovered an atypical five-subunit Arp2/3 complex in Plasmodium", because they haven't proven it has five subunits, they just only found 5. They also can't say "Unlike the canonical seven-subunit Arp2/3 complex found in other eukaryotes..." because they haven't proven it isn't seven.

As far as I can see on rereading, it's only the abstract that is bad. Perhaps the authors changed the body text and forgot the abstract, but please go one more time through the manuscript and remove any other statements that appear to state they have discovered a 5-member complex (or purify it to visibility by Coomassie, which is fairly easy, but a lot of work for this...).

They could also add a couple of sentences discussing this point in the discussion; it's interesting. I do think there's a good chance the complex is 5-membered, but also a good chance that two subunits diverged for some reason...

Reviewer #3

(Remarks to the Author)

I believe the authors have successfully addressed most of the issues raised by the reviewers and have significantly improved their manuscript.

Decision Letter:

Our ref: NMICROBIOL-24051562A

25th March 2025

Dear Franziska and Freddy,

I hope you are both doing well. I am writing with good news.

Thank you for submitting your revised manuscript "An atypical Arp2/3 complex is required for Plasmodium DNA segregation and transmission of malaria." (NMICROBIOL-24051562A). It has now been seen by the original referees and their comments are below. The reviewers find that the paper has improved in revision, and therefore we'll be happy in principle to publish it in Nature Microbiology, pending minor revisions to satisfy the referees' final requests and to comply with our editorial and formatting guidelines.

We are now performing detailed checks on your paper and will send you a checklist detailing our editorial and formatting requirements in about 1-2 weeks. Please do not upload the final materials and make any revisions until you receive this additional information from us.

Thank you again for your interest in Nature Microbiology. Please do not hesitate to contact me if you have any questions.

Best wishes,

Reviewer #1 (Remarks to the Author):

I have read the extensive rebuttal, additional data and changes to the manuscript. I am satisfied that the authors have address my specific comments as well as those of other reviewers. The paper is clearly the beginning of a lot more work to come, which I look forward to seeing in time. I have no further requested changes/comments.

Reviewer #2 (Remarks to the Author):

The style of the paper is much better, it's much clearer, the alphafold section is neat, but the authors have not quite got the message. In the abstract they cannot say "Here we discovered an atypical five-subunit Arp2/3 complex in Plasmodium", because they haven't proven it has five subunits, they just only found 5. They also can't say "Unlike the canonical seven-subunit Arp2/3 complex found in other eukaryotes..." because they haven't proven it isn't seven.

As far as I can see on rereading, it's only the abstract that is bad. Perhaps the authors changed the body text and forgot the abstract, but please go one more time through the manuscript and remove any other statements that appear to state they have discovered a 5-member complex (or purify it to visibility by Coomassie, which is fairly easy, but a lot of work for this...).

They could also add a couple of sentences discussing this point in the discussion; it's interesting. I do think there's a good chance the complex is 5-membered, but also a good chance that two subunits diverged for some reason...

Reviewer #3 (Remarks to the Author):

I believe the authors have successfully addressed most of the issues raised by the reviewers and have significantly improved their manuscript.

Version 2:

Decision Letter:

25th April 2025

Dear Franziska and Freddy,

I am pleased to accept your Article "An atypical Arp2/3 complex is required for Plasmodium DNA segregation and malaria transmission." for publication in Nature Microbiology. Thank you for having chosen to submit your work to us and many congratulations.

Authors may need to take specific actions to achieve [compliance](https://www.springernature.com/gp/open-research/funding/policy-compliance-faqs) with funder and institutional open access mandates. If your research is supported by a funder that requires immediate open access (e.g. according to [Plan S principles](https://www.springernature.com/gp/open-research/plan-s-compliance)) then you should select the gold OA route, and we will direct you to the compliant route where possible. For authors selecting the subscription publication route, the journal's standard licensing terms will need to be accepted, including [self-archiving policies](https://www.nature.com/nature-portfolio/editorial-policies/self-archiving-and-license-to-publish). Those licensing terms will supersede any other terms that the author or any third party may assert apply to any version of the manuscript.

Congratulations once again and I look forward to seeing the article published.

With kind regards,

P.S. Click on the following link if you would like to recommend Nature Microbiology to your librarian
<http://www.nature.com/subscriptions/recommend.html#forms>

** Visit the Springer Nature Editorial and Publishing website at http://editorial-jobs.springernature.com?utm_source=ejP_NMicro_email&utm_medium=ejP_NMicro_email&utm_campaign=ejp_NMicro for more information about our career opportunities. If you have any questions please click [here](mailto:editorial.publishing.jobs@springernature.com).

Reviewer Comments:

Reviewer #1 (Remarks to the Author):

Review of Hentschel et al. "A non-canonical Arp2/3 complex is essential for Plasmodium DNA segregation and transmission of malaria".

In this paper, Hentschel et al lead off with the study of a protein, ARPC1, that eventually leads to identification of a minimal arp2/3 complex (5 units rather than the canonical 7). The study elegantly links the function of this complex to DNA segregation in Plasmodium gamete formation using a suite of cell and genetic techniques. The study is formidable to say the least (I applaud the authors!), extensive and convincing in scope – it challenges a dogma (no Arp2/3 in Apicomplexa!) and in doing so opens up a new vista of biology linking actin and the fascinating world of microgametogenesis. I would certainly expect its publication and believe it will be very favourably received by the microbial cell biology community. My comments below are purposefully pedantic, as I feel a piece of work this high quality deserves to be poured over with a fine-tooth comb. I wouldn't want the authors to think, however, I was being overly negative. Below I've made some comments/suggested changes that I would ask are considered, though this should not preclude publication of the work.

Heartfelt thanks to this reviewer for his/her enthusiastic appreciation of our work and the wonderfully pedantic (ie very helpful) comments.

Comments:

1. The male specific phenotype of the ARPC1 KO was reported in the preprint by Sayers et al. 2023: doi: <https://doi.org/10.1101/2023.12.25.572958>, Table S1). Acknowledging the studies were posted at similar times, this is not a suggestion that one preceded or scooped the other, however, it might be good to acknowledge this in the paper – that observations are in line with a genetic screen for fertility genes in Plasmodium. This is also true for Alp5a (PBANKA_0811800), adding further weight to the present study.

Thank you for pointing this out, we have added this reference to line 128-129, which now reads:

"ARPC1 is thus required for male, but not female fertility, in line with data from a recent genetic screen for fertility traits in Plasmodium³⁷."

We also added a sentence in line 231-232 to add information about the available data from this screen for the other genes:

"Both Alp5b and AKi7 were recently reported to be required for male fertility, while there is no data available for the other pulldown hits³⁷."

2. I found the images in Extended Data Fig. 2c a little underwhelming. With such amazing images throughout the rest of the paper, the presence of a smudge of staining in gametocytes and ookinetes doesn't do justice to the initial "Ah Ha!" moment of ARPC1 being nuclear in just these stages.

The images of Extended Data Fig 2c are indeed meant only as a starting point as imaged on a standard widefield fluorescence microscope. As such they provide initial information about the dynamics of ARPC1 localization across the blood stage and transmission cycle of the parasite. The precise subcellular localization of ARPC1 in gametocytes is extensively imaged in high resolution in the rest of the paper. We thus believe that there is little added value to repeat the experiment with the ARPC1-GFP line just to obtain new images with higher resolution, and we would prefer not to sacrifice mice for this purpose. To better highlight the nuclear localization of ARPC1, we have now added a dashed line marking the cell circumference in ARPC1 and merged images.

3. A minor point in Extended Data Fig. 2, the first column is labelled 'asexuals', with the next column labelled 'schizonts' (also asexual stages). The first column should perhaps be relabelled to trophs or rings?

We have followed the reviewers suggestions and relabeled the first column in Extended Data Fig. 2c to "Rings/Trophozoites" instead of "Asexuals".

4. A question. Beginning on line 123 the authors describe crossing ARPC1 knockout mutants with female deficient and male deficient lines, with only the female-deficient lines resulting in normal sized oocysts. This is taken as evidence that “ARPC1 is thus required for male, but not female fertility.” Do the authors feel they can fully make this claim if they haven’t shown these smaller oocysts (from the male deficient cross) don’t produce sporozoites? I.e. as previously shown with the ARPC1 knockout oocysts (Fig. 1b).

We have not included sporozoite numbers, as in our laboratory we observe a low but detectable oocyst infection rate of the sex-specific lines on their own, leading to sporozoites detected in the salivary gland (Rebuttal Figure 1A, B). Any sporozoite counts and bite-back experiments are thus confounded by the presence of such breakthrough parasites from the sex-deficient lines. Indeed, we see a minimal number of 250 sporozoites in the *PbARPC1(-)* cross with the *Pb48/45(-)* male-deficient line. That number is even 20 times lower than the number of breakthrough parasites when infecting the male-deficient *Pb48/45(-)* line alone (Rebuttal Figure 1B). The advantage of analyzing the phenotype based on oocyst size is that we can identify those parasites that are at least heterozygous for the ARPC1 knockout by their RFP fluorescence (the sex-deficient lines are in a wildtype background). Notably, the oocysts from the *PbARPC1(-)* x *Pb48/45(-)* cross do not proceed further in their development and are also small at day 12 (Rebuttal Figure 1 C).

Rebuttal Figure 1: Breakthrough infections of sex-deficient lines. **A)** Oocyst numbers per midgut on day 12 after infection. **B)** Sporozoite numbers per mosquito on day 18-19 post infection **C)** Oocyst size on day 12 after infection. Note that the ARPC1(-) data in the first violin graph is taken from Figure 1g in the manuscript. A-C) Data from a single mosquito cage infection.

The data presented here is from a single replicate. As these results are in agreement with our observation that ARPC1 is required for male but not female fertility, we would respectfully refrain from performing additional replicates to save animals.

5. In Fig. 1k – the statistical impact of this flow data isn’t clear to me - I can’t seem to locate how this data was obtained? Detail on the number of replicates, statistical significance of it etc is missing. I’d suggest this should be quantified/cleared up. Related, and mirroring point 3, this also raises the unanswered question of why ARPC1 would be present at all in the female gametocyte? Is this just because of timing/shared developmental pathways, or do the authors think ARPC1 may nonetheless have a role in the macrogamete. The data presented make it unclear whether this should be an avenue of investigation or would be a dead end to look down. A comment on this might be worthwhile but left unanswered it feels confusing.

We have now added a panel e to the Extended Data Fig. 6, quantifying the median fluorescence intensity of males and females stained with Alexa-647-anti-HA, as measured by flow cytometry in three independent biological replicates and performed statistical analysis of the data. The figure legend (line 1197-1199) now reads

*“e, Median fluorescence of males and females stained with Alexa-647-anti-HA as measured by flow cytometry. Each dot represents an independent biological replicate. Statistics: Two-way ANOVA, *, $p < 0.05$.”*

This statistical analysis revealed a significantly increased signal in *Pb₈₂₀ARPC1-HA* males compared to both *Pb₈₂₀WT* males and *Pb₈₂₀ARPC1-HA* females. These data demonstrate the predominant expression of ARPC1 in male gametocytes compared to low levels in females. Indeed, low expression levels in *Pb₈₂₀ARPC1-HA* females were not significantly higher compared to the background signal of *Pb₈₂₀WT* females. Together with the observation that crossing ARPC1(-) with a female-deficient line fully rescues the phenotype, this indicates that the minor residual expression of ARPC1 in females is not of biological relevance. We have now added the following phrase in line 136-140:

“As the ARPC1-HA signal observed in females was not significantly higher than WT background (Extended Data Fig 6e) and crossing ARPC1(-) with a female-deficient line restored the phenotype (Fig. 1j), it is likely that the residual expression of ARPC1 in females reflects the shared developmental pathway with males rather than a specific biological function.”

6. Line 171 is perhaps a bit too strong from the evidence provided. Perhaps something like “contributes to” would be more appropriate than “is responsible for” (very definitive singular-sounding statement).

We have changed the line (now line 179) to:

*“We therefore conclude that deletion of ARPC1 results in sub-haploid microgametes that contribute less than one complete genome to the zygote, suggesting that ARPC1 is **required** for proper DNA segregation into the developing male gametes.”*

7. Similarly, on line 188, it is not conclusively shown that “ARPC1 is essential for DNA segregation during male gametogenesis”, though the data certainly suggests that it influences DNA segregation.

We have changed the line (now line 196) to:

*“Thus, ARPC1 is **required for correct DNA segregation** into male gametes, however subsequent sub-haploid gametes are still fertile and ARPC1(-) parasites can undergo meiosis and ookinete formation.”*

8. Another key observation that I feel is not definitively demonstrated that I suggest is changed, is whether the minimal arp2/3 complex is actually a nucleator. It associates/co-localises with F-actin, it's disruption and the disruption of F-actin phenocopy each other, but this doesn't prove it is a nucleator. Indeed, other actin nucleators [formins] may be present (is MISFIT a nucleator?). Therefore, I feel the statement that it is a nucleator should be tempered to “may be” or “consistent with it being”. Definitive proof as a nucleator either requires biochemical validation or the slightly less satisfying experiment where absence of the complex is associated with absence of F-actin. Slightly more convincing, but biochemical would be best. Minimally, I'd suggest tempering the statements (e.g. line 341 – emphatic statement it nucleates actin...).

We agree that we lack direct evidence that *Plasmodium* Arp2/3 acts as an actin nucleator. Biochemical validation is part of a follow-up grant and will require purification of all complex components in large amounts, or expression of these proteins in heterologous systems, both approaches that are highly challenging for *Plasmodium* and that are out of scope for this manuscript.

We attempted to image actin at the spindle in Pb₄₇₃WT and ARPC1(-) parasites using the chromobody-emerald (CBE) reporter. Using widefield live microscopy, we successfully detected F-actin at the spindle of activated wildtype CBE gametocytes, similar to our findings with the ARPC1-mCherry/CBE line illustrated in Figure 4e of the manuscript. Unfortunately, we were unable to visualize the microtubule spindle itself, as live microtubule dyes such as SirTubulin inhibited exflagellation. Consequently, we could not conclusively demonstrate if the imaged cell was an activated gametocyte, making it challenging to determine the presence or absence of the CBE signal at a spindle (given that we can't detect the spindle). Additionally, the low signal-to-noise ratio of the CBE reporter further complicated reliable imaging.

To address these challenges, we attempted to fix CBE-expressing gametocytes from WT and ARPC1(-) backgrounds. Despite experimenting with various fixation buffers and concentrations of glutaraldehyde and PFA, all efforts resulted in a loss of the CBE signal from the spindle in the wildtype post-activation, possibly reflecting the sensitivity of the actin filaments to chemical fixation. Hence answering this will necessitate cryo electron tomography from lamellae cuts, a highly time consuming approach that we will attempt in collaboration in the future.

Altogether, we are aware of the limitations raised by reviewer 1 (and a similar point raised by reviewer 3) and have explored various approaches to directly demonstrate that the *Plasmodium* Arp2/3 complex acts as an actin nucleator. As technical limitations have so far precluded this, we followed the suggestion of the reviewers and have overall toned down the claims in the manuscript, and now clearly state that direct evidence is still lacking that Arp2/3 acts as a nucleator in line 395-397:

“Arp2/3 subunits colocalise with F-actin, and inhibition of actin polymerisation during gamete formation phenocopies their deficiency. These findings are consistent with a model in which Plasmodium Arp2/3 functions as an actin nucleator, similar to canonical Arp2/3 complexes, however, direct biochemical evidence is still required to support this hypothesis.”

9. A question: Is there any evidence from imaging (here or elsewhere by the group) of branched actin filaments in gametes/gametocytes after activation? This would be very strong evidence for the complex retaining arp2/3 like nucleator features. In their absence, arp2/3 may actually be an F-actin binding complex (nucleated by a different nucleator – see above) that builds on F-actin present creating the link to the kinetochore etc.

We are not aware of any imaging showing branched actin filaments in *Plasmodium*. Indeed, the prevailing view in the field is that *Plasmodium* actin forms linear filaments polymerized by formins. Mammalian and yeast Arp2/3 complexes can also nucleate actin in absence of a mother filament if activated by a nucleation promoting factor (NPF) of the WISH/DIP/SPIN90 family. We have not found homologues to these NPFs in *Plasmodium* (yet), hence it is still an open question if *Plasmodium* Arp2/3 may facilitate branched or linear actin filament formation. Further work (biochemistry and cryo-ET) is required to investigate these open questions. We have highlighted this open question now in lines 397-398:

“It also remains to be investigated if the observed F-actin filaments represent linear or branched actin filaments.”

10. In the discussion, line 410, in reference to the puzzling “delayed-death” oocyst phenotype it might be worth saying that this phenomenon is observed with other genes that function in microgametogenesis, e.g. as exemplified by the SUN1-KO (Sayers et al. 2023 preprint, see above) and EBI-KO (Zeeshan et al. 2023, <https://doi.org/10.1038/s41467-023-41395-3>) parasites.

Thanks for pointing this out. Indeed, we have noted several previously published genes that have a similar phenotype as Arp2/3 components, i.e. paternal phenotype followed by early oocyst arrest, as listed in the previous paragraph (lines 456-459). We have now extended the list by including SUN1 from Sayers et al. We have also highlighted once more that this phenotype is likely not specific to Arp2/3, but a general phenotype often observed with defects in microgametogenesis by adding the following sentence to the paragraph (lines 477-478):

“Finally, the Arp2/3 complex could have a separate function during oocyst development. Yet, the observation of a similar delayed-death like phenotype in other parasite mutants, such as knockout lines of EB1 or SUN1, points towards a more general phenomenon.”

11. Methods (line 508): minor question, to my understanding, PlasmogEM vectors are typically digested with NotI – the authors say they used NdeI – is this correct?

Thanks for noting, that was a mix-up from our side. The vector was digested with NotI, as all PlasmogEM vectors. The respective sentence in the methods section has now been corrected.

JB 04/JUN/2024

Reviewer #2 (Remarks to the Author):

Very interesting work. The discovery of an aberrant Arp2/3 complex in *Plasmodium* is extremely interesting. The novel phenotypes are convincing.

Thank you for the positive evaluation of our work.

It's not sound yet though. Most importantly - the authors are playing both sides. They're saying that just searching databases is not strong enough to find Arp2/3 complex members as evolutionarily distant as they are in *Plasmodium*; but they seem to be concluding that two complex members are missing without an exhaustive enough search to justify it that strongly. The title, for example, uses "non-canonical" but it's not proven that the missing two subunits aren't there - they just didn't find anything.

The use of "essential" in the title is a bit over-egged too, the data are more qualified.

My suggestion is moderate the language, to make it clearer what is proven and/or absolute and what is not, and also try more in-depth tests (or argue more cogently...) about the non-canonical nature of the targets.

We have now moderated the language throughout the manuscript, as exemplified by the title which now reads *"An atypical Arp2/3 complex is required for Plasmodium DNA segregation and transmission of malaria"*

We also removed the term non-canonical from everywhere in the manuscript and now explicitly point out in the discussion that the two missing subunits might be too divergent to identify easily:

"While the canonical Arp2/3 complex consists of seven subunits, we have so far only identified five subunits. We did not find evidence for putative Plasmodium ARPC3 and ARPC5 subunit orthologs in any of our three pulldowns or by structure-based search. Their existence cannot be excluded, as they may exhibit weaker interactions with the complex and thus be undetectable in our experimental conditions and be too divergent for identification by structure-based search."

It's also very densely written in parasite language. I'm an actin biologist and found a lot of the logic difficult. This may be the style and the word limits, but it would be good to try and better explain why the plasmodium-specific experiments lead to the actin conclusions.

We indeed aim at talking to both cell biologists and parasitologist in as concise a form as possible or dictated by word limits (which we admittedly stretched). Throughout the manuscript we carefully rewrote parts where too much parasitology terms might confuse by either cutting them, simplifying them and/or explaining them briefly.

For example, the sentence

"Male gametogenesis is a particularly complex, fast, and intriguing process, in which within just 15 minutes eight flagellated microgametes are formed simultaneously from a progenitor gametocyte with a single, octoploid nucleus."

now reads

"Male gametogenesis is a particularly complex, fast, and intriguing process: within just 15 minutes a progenitor gametocyte undergoes three rounds of endomitosis, resulting in the formation of an octoploid nucleus that is then divided as eight flagellated microgametes emerge simultaneously from the mother cell" (line 45-48)

and

"In WT gametocytes, [...] 20% of cells were exflagellating, with DNA localising to the budding flagella."

now reads

"In WT gametocytes, [...] 20% of cells were exflagellating (gametes beating their flagella and leaving the mother cell), with DNA localising to the budding flagella." (lines 155-159).

But - returning to my starting line - a form of this should be published expeditiously.

Thanks for this generous assessment.

=====

Slightly more detail:

- L50 Cite primary literature. Mullins and Machesky, not these very wordy reviews. Also this line has a typo.

We have now added the references Mullins et al., 1997, Welch et al. 1997 and Machesky et al 1997 to the first sentence. Given that parasitologists not necessarily have a deep background in Arp2/3 biology, we decided to keep the reviews as references for additional information.

- L75 I am not at all sure how "tagging ARPC1 with GFP" allows this conclusion. Either explain better or do a better experiment. Is the GFP knocked directly into the genome so the normal expression is accurately retained? Do the authors know this, and is it accepted in the Plasmodium field? It wouldn't be in mammals/Drosophila/elegans/Dictyostelium.

To characterize ARPC1 expression and localization across the *Plasmodium* life cycle, we tagged the endogenous ARPC1 itself with GFP at the C-terminus, so that it is expressed from the endogenous locus under the ARPC1 promoter. This is indeed the standard way of characterizing protein expression in the *Plasmodium* field. It conveniently also allows to check if adding a GFP tag has a detrimental function by following the parasite across the life cycle and compare rates of infection to the wild type. In contrast to the ARPC1(-), the ARPC1-GFP line did not exhibit any phenotypic defects and formed normal oocysts, indicating that the tag does not interfere with protein expression or function. The observed ARPC1 localisation at the male mitotic spindle also “matches” to the phenotype of the ARPC1 knockout. To better highlight this, we have now added in the text the following sentences (line 77-82);

“Tagging ARPC1 endogenously at the C-terminus with GFP in P. berghei wildtype (WT) parasites (Extended Data Fig. 2a, b) revealed that the protein is predominantly expressed in gametocytes and ookinetes (the motile zygote that forms in the mosquito midgut after a blood meal) and localises to the nucleus (Extended Data Fig. 2c). A weak ARPC1-GFP signal was also observed in late-stage oocysts. Notably, ARPC1-GFP exhibited no phenotypic defects, indicating that the C-terminal tag did not affect the function of ARPC1.”

- L214 I really do not understand this argument. Are the authors trying to say the complex only assembles in activated cells? That's completely unlike what is seen in other organisms. Or are they trying to say the IP doesn't work because there isn't any? It goes back to the issues with the GFP transfection; there may be field-specific meanings but they need a clearer airing.

Indeed, for each of the three proteins we used as bait (ARPC1, ARPC2, AKi7), we found significant enrichment of the other Arp2/3 components only in IPs from activated gametocytes (based on our thresholds for defining a hit). In IPs from non-activated gametocytes, we generally did not observe many significant hits, and these did not include other Arp2/3 components. We have no reason to assume that the IP only works in activated cells, so it is indeed tempting to speculate that (unlike in other organisms) the Arp2/3 complex in *Plasmodium* only assembles upon activation. However, we agree that this hypothesis remains speculative and requires further investigation that is beyond the scope of this manuscript. Hence, we do not wish to make any claims about Arp2/3 assembly in this manuscript. To clarify that we just describe an observed result at line 214 (now line 222), we have changed the text to:

“In pulldowns from non-activated ARPC1-GFP gametocytes, we did not identify any significantly enriched hits besides ARPC1 itself. In activated gametocytes, however, we found five additional proteins to be enriched in ARPC1-GFP compared to PbGFP_{con}.”

- L275 the thing I really want colabfold or alphafold 2 to answer is, is this proposed complex structurally reasonable & energetically possible? In particular what does the surface where the missing canonical subunits should be look like? I suspect the standard (“canonical”) Arp2/3 complex would not survive loss of 2 subunits because the solution energetics of the exposed hydrophobic surfaces would be awful. Can they model a 5-member complex and show it can work? If you compare it with a canonical one, has there been a clear stabilising change where the other 2 subunits should bind?

We have now used Alphafold3 to model the five so far identified Arp2/3 subunits together with *Plasmodium* actin 2. We observed that the arrangement of the Arp2/3 subunits depends on the number of actin monomers in the model. Interestingly, in those predictions with the highest score (with three, four or five actin monomers), Arp2/3 subunits locate in a similar arrangement as their counterparts in the canonical Arp2/3 complex (Rebuttal Figure 2 A, B), giving some support to the hypothesis that the identified five subunits form an Arp2/3-like ensemble. However, to draw any firm conclusions about the structure of the complex would clearly require more experimental evidence.

We have also investigated the surfaces on those sites where the missing ARPC3 and ARPC5 subunits would be located (Rebuttal Figure 2 C, D). We frankly are unable to judge if there is a clear hydrophobic patch in the bovine Arp2/3 complex where ARPC3 or ARPC5 bind, but at least it appears that there is no extensive hydrophobic patch on the surface of *Plasmodium* ARP2/3. Interestingly, our modeling suggests that the extended domain of PbARPC4 could take the place of where ARPC5 binds in canonical Arp2/3 complexes.

Rebuttal Figure 2: Modelled surface hydrophobicity of bovine and *Plasmodium* Arp2/3 complexes. **A, B)** Two views of the bovine Arp2/3 complex in the branch junction (experimentally determined structure) and the AlphaFold 3 multimer prediction of the putative *Plasmodium* Arp2/3 complex. Views are selected to show (A) ARPC3 and (B) ARPC5 in the bovine complex. **C, D)** Surface hydrophobicity of (B) ARPC3-facing side and (C) ARPC5-facing side of the (predicted) Arp2/3 structures. **C, D)** From left to right: Surface plot of entire bovine Arp2/3 complex, surface plot with (C) ARPC3 or (D) ARPC5 highlighted as stick model, surface plot of Arp2/3 complex without (C) ARPC3 or (D) ARPC5, surface plot of equally oriented *Plasmodium* Arp2/3 complex. For D: Last plot indicates *Plasmodium* Arp2/3 subunit with PbARPC4 highlighted as stick model. Red circle indicates (theoretical) position of subunit of interest.

We have thus added predicted PbArp2/3 structures together with actin and the corresponding structure from *Bos taurus* to Fig 4 and Extended Data Fig 13. We also describe and discuss these results with the appropriate caveats in lines 284-296 and x of the revised manuscript. In light of the inconclusive result and the uncertainty of the modeling of the *Plasmodium* Arp2/3 complex, we do not think that including the surface hydrophobicity maps shown above will benefit the manuscript.

Note that in that context, we have updated all structural modeling, also of the single subunits (Fig 3b and Extended Data Fig 10) to AlphaFold 3 predictions.

- L290 this seems to be using the same result both ways. Is CK666 supposed to bind and work or not? I don't understand what this is trying to say at all.

We apologize for the confusion. We hypothesized that CK-666 and CK-869 would affect DNA segregation by inhibiting the *Plasmodium* Arp2/3 complex. However, our data revealed that microgametes treated with either inhibitor contained a full haploid genome set, suggesting that these inhibitors do not block the *Plasmodium* Arp2/3 complex itself, possibly because it is structurally too different to be bound. However, we observed an overall drop in microgamete formation, which is likely an off-target effect given that we do not observe this phenotype in the Arp2/3 knockout lines. We have now rephrased this section to clarify.

Lines 296-307: *"Given the predicted structural conservation of the Plasmodium Arp2/3 complex, we hypothesised that it could be inhibited by the known Arp2/3 inhibitors CK-666 or CK-869⁴⁹. However, upon drug treatment of gametocytes during activation we observed a different phenotype: CK-666 and CK-869 both impaired overall exflagellation rates in a titratable manner compared to the non-active control CK-689, and the few gametes that formed often lacked DNA (Extended Data Fig. 14a). Nevertheless, those microgametes that were DNA-positive contained a complete haploid genome, indicating that neither CK-666 nor CK-869 did affect DNA segregation itself (Fig. 4d and Extended Data Fig. 14b). The phenotypic difference between drug treatment and knockout of Arp2/3 subunits suggests that CK-666 and CK-869 do not inhibit the Plasmodium Arp2/3 complex itself, possibly because the binding site for these drugs in the Plasmodium complex is not conserved. The impact of CK-666 and CK-869 on exflagellation may instead be an off-target effect."*

Note that we moved this section to improve the overall flow of the manuscript.

- L420 also trying to argue both ways – it's noncanonical but none of the subunits were previously found be found by searching - why so certain the other two components aren't there? ARPC4 is already in the mix – are the other subunits? The "noncanonical" seems to be overreaching. Particularly as L425 "Similar hybrid Arp2/3 complexes have been described to form at points of focal adhesion" they aren't similar at all - they lack p40/p41, the one really conserved member in the Plasmodium, and as far as I can see ARPC4 is in the Chorev complex but not in the one reported here. It's very loose to say two partial complexes are "similar" if they are asserted to lack different subunits (plus as far as I know the Chorev paper has not led to much, either; and the citation of a Trends article instead of primarily literature is not really appropriate to back up this point).

Thanks, point well taken. As mentioned above, we have moderated the language throughout the manuscript to highlight the uncertainties regarding the presence or absence of the ARPC3 and ARPC5 subunits. Line 484-488 now reads:

"We did not find evidence for putative Plasmodium ARPC3 and ARPC5 subunit orthologs in any of our three pulldowns or by structure-based search. Their existence cannot be excluded, as they may exhibit weaker interactions with the complex and thus be undetectable in our experimental conditions and be too divergent for identification by structure-based search."

As correctly pointed out, the previously described hybrid Arp2/3 complex is not similar in composition nor function to our here discovered *Plasmodium* Arp2/3 complex. We have thus removed the respective sentence from the discussion.

So overall - I am very keen on the reporting of the identified subunits of the complex and the very interesting roles in segregation and infection; but I'd like to see much more moderation and scholarship in the assessment of whether the complex is truly partial. Some combination of more support and more careful reporting would sort this out.

We hope our revised manuscript satisfies the concerns of the reviewer and thank him/her again for the constructive comments. Naturally this work opens up many questions that will be pursued by the first author in her own independent lab.

Reviewer #3 (Remarks to the Author):

In this paper, the authors explore the role of the Plasmodium Arp2/3 complex. In other eukaryotes, this complex plays a major role in the regulation of the actin cytoskeleton. However, its role in Plasmodium is unknown. Using AlphaFold, they first identify ARPC1 as a subunit of a non-canonical Plasmodium Arp2/3 complex. Here, they show that the protein is mainly expressed in male microgametocytes where it shows dynamic localisation around the mitotic spindles. Deletion of *arpc1* leads to the formation of sub-haploid male gametes leading to a complete block in parasite transmission. They further identify five other components of the complex including evolutionary conserved proteins and more apicomplexan-specific proteins. Interestingly, one of these proteins has previously been identified in immunoprecipitations of kinetochore proteins in Plasmodium gametocytes. They also show presence of presence of F-actin in the nucleus during male gametogenesis. As cytochalasin D, an inhibitor of actin polymerisation led to a similar phenotype to that of parasites lacking components of the Arp2/3 complex, the authors propose that the Arp2/3 complex establishes a link between kinetochores and actin II filaments to allow DNA incorporation into the forming male gamete.

I find the paper well written, the experimental design robust and the experiments performed to a high standard.

I however see two major limitations to this study that include i) the robustness of the link between the mitotic spindle, the kinetochores and the ARP2/3 complex and ii) the detailed investigation of the nature of the nuclear actin filaments.

We thank the reviewer for the overall positive evaluation of our manuscript. We agree that the discovery of the *Plasmodium* Arp2/3 complex opens up many new questions regarding the molecular mechanism of its function. Please find below our detailed response to the individual points.

Major comments

1. The link between the Arp2/3 complex with the kinetochore and mitotic spindle

- The link between the Arp2/3 complex with the kinetochores and their attachment to the mitotic spindle requires more attention. The assumption that Akit7 links kinetochores with an actin filament-bound Arp2/3 complex is mainly based and the co-immunoprecipitation of Akit7 with ARPC1 and 2. However Akit7 seemed to be an atypical component of the kinetochore during male gametogenesis in PMID36006241. In the absence of other kinetochore components in immunoprecipitations of Arp2/3 components or reciprocal immunoprecipitation or functional analysis of Akit7, there is a risk that that the author hypothesis on the role of Akit7 is not correct.

We thank the reviewer for this suggestion, which led us to perform several new experiments to further test the hypothesis of a direct Arp2/3-kinetochore interaction that indeed resulted in a correction of the model. To test if AKi7 links the Arp2/3 complex to the kinetochore, we tagged AKi7 with YFP and performed immunoprecipitation. Importantly, the endogenous tagging of AKi7 with YFP did not affect male DNA segregation, thus, functionally important protein-protein interactions are likely not interrupted by the tag. The IP confirmed the interaction of AKi7 with the components of the Arp2/3 complex, and - somewhat surprisingly and contradicting previous publications - did not reveal any interaction with known kinetochore proteins. Instead, we found several proteins involved in chromatin organization and DNA replication, such as the MCM complex, DNA polymerases and histones. Given that we do not see an effect of Arp2/3 on DNA replication, we currently assume that these hits reflect the more pan-nuclear localization of AKi7. The precise functional role of AKi7 thus remains to be addressed and will be the scope of future studies.

We have added the data in the new Extended Data Fig. 16 and present them in the results in lines 338-353. We have also adapted the final model (new Fig. 6) and rephrased the abstract and discussion (lines 421-423) to clarify that there is no evidence for a direct interaction of the Arp2/3 complex with the kinetochore. However, please note that our data also does not rule out such an interaction, e.g. a transient or low affinity one.

-In the same line, the phenotype shown in figure 5a is unclear. What is the exact fate of kinetochores upon ARPC1 deletion? With shown resolution, the Ndc80-YFP signal in absence of ARPC1 is difficult to understand. Are the kinetochores detached from the spindle, stuck onto the spindle and/or unstable in the absence of ARPC1? In the two later cases, this would indicate a different or additional role for the Arp2/3 complex. A phenotyping with a higher resolution such as EM or expansion microscopy may help to address this important issue.

In absence of Arp2/3, kinetochores detach from the spindle and are found free within the nucleoplasm, indicating that Arp2/3 is important for maintaining kinetochore-spindle connections. We have now imaged NDC80-YFP at 7 minutes post activation, where two spindles are present. Interestingly, we already observe the partial detachment of the kinetochores from the spindle at this earlier time point (new Extended Data Fig 17 c,d and 18). To better visualize the localization of NDC80, we have added line profiles to our microscopy images, demonstrating that in the WT, NDC80 colocalises with the tubulin signal. In contrast, this colocalisation is only partial in the ARPC1(-), and free NDC80 foci can be found within the nucleus (Fig. 5b, Extended Data Fig 17 c,d). In this process, we have also added a third, independent biological replicate to the experiment and added it to Figure 5c. The new results are described in lines 358-372, and the model (now Figure 6) was adapted to include the partial loss of kinetochore-spindle attachment even before exflagellation.

We indeed tried EM and expansion microscopy, but both did not yield any clear answers. Likely EM will only be informative from high pressure freezing and serial sectioning.

2. The nature and role of nuclear actin filaments during male gametogenesis

- The actin filaments/signal should be localised with a higher resolution. On the images shown here, ARPC1 or the signal from the F-Actin chromobodies does not fully localise to the mitotic spindle and shows a slightly wider distribution. It is also unclear to me where the actin filaments are anchored to and how would this allow tethering of the kinetochores to the microtubules or kinetochore stability during mitosis?

We acknowledge that the current resolution of the actin signal could be enhanced. As stated in the response to reviewer 1, we have made extensive attempts to achieve higher resolution imaging of F-actin through confocal live imaging or by testing various fixation methods. However, we were only able to observe the chromobody localization to the spindle using widefield microscopy. Under confocal live imaging conditions, gametocytes did not activate, likely due to the laser affecting cell viability. Upon fixation, the chromobody signal was lost, possibly because the actin filaments are sensitive to cell fixation. We are actively working on addressing these challenges, but we do not anticipate resolving them in the near future.

In light of the difficulties in imaging F-actin, we are also unable to answer the question where actin filaments are anchored to, and how this precisely supports kinetochore-spindle tethering. The detailed molecular mechanism of these processes is the focus ongoing work beyond the scope of this study as it will include cryo electron tomography studies.

- How does the nuclear actin signal look like in the absence of components of the ARP2/3 complex? Similarly, how does cytochalasin D affect kinetochore assembly/motility along the mitotic spindle mitosis and/or direct incorporation of DNA into the forming gamete? I think that answer these questions are important to support a link between the actin cytoskeleton, the Arp2/3 complex and kinetochores.

We followed the reviewer's suggestion to determine kinetochore localisation in presence of CytoD. This experiment revealed that CytoD treatment, i.e. inhibition of actin polymerization, results in the same phenotype as the knockout of ARPC1: NDC80 does not localize exclusively to the spindle and/or eight distinct foci, but we find a rather dispersed localization throughout the nucleus. Example images and a quantification of the phenotype have been added to Figure 5c, d and Extended Data Figure 18 and the following lines were added to the text:

"Notably, we observed the same dispersed NDC80 signal when treating WT gametocytes with the actin polymerisation inhibitor cytochalasin D, which also resulted in aberrant DNA condensation and segregation during exflagellation (Fig. 5d,e, Extended Data Fig. 18b)."

We agree that it is important to investigate if nuclear actin localizes to the spindle in absence of the Arp2/3 complex. Unfortunately, investigating nuclear actin signal in ARPC1(-) parasites was unsuccessful due to the above

mentioned challenges in fixing the F-actin chromobody signal in activated gametocytes. Live imaging conditions did not allow for spindle visualization, as the use of live microtubule stains such as SirTubulin inhibited exflagellation. Without clear spindle visualization, it was impossible to definitively identify an activated gametocyte during imaging and thus determine the presence or absence of the F-actin chromobody signal at the spindle. We have now toned down the conclusions throughout the manuscript and state that while data fits with a model in which *Plasmodium* Arp2/3 nucleates actin filaments, more biochemical work is needed to support this hypothesis (lines 395-397)

Minor comments

I would suggest the authors to be clearer when they use the term DNA/kinetochore segregation. Do they mean segregation along the mitotic spindle or integration into the forming gamete? While the first option likely leads to the second, both processes are nevertheless distinct. As far as I understand both possibilities are alternatively mentioned using the same wording, which makes it difficult to understand the observed phenotypes.

Sorry for the confusion. We here refer to DNA segregation as the process of genome integration into the forming gamete. As there is no classical DNA condensation happening through the three rounds of mitosis during gametocyte activation, clear DNA segregation along the mitotic spindle is not visible by microscopy. We have now edited the manuscript to clarify this point, e.g. in line 386-388 which now reads:

“In this study we present the discovery of an atypical Plasmodium Arp2/3 complex which is required for maintaining kinetochore-spindle attachment during male gametogenesis in the mosquito and consequently correct DNA segregation into emerging gametes.” (instead of *“DNA segregation during male gametogenesis”*)

I would suggest the authors to indicate that ARPC1 or 2 interact with a kinetochore component not necessarily with kinetochores

We agree with the reviewer on this point, especially in the light of the new interactome of AKi7, which does not support a direct interaction of the Arp2/3 complex with the kinetochore itself. We have rephrased this throughout the manuscript, for example in the abstract which now reads “

L135 closed mitosis is not a synonym for endomitosis.

We have removed the term closed mitosis from the sentence, which now reads (line 143-144):

*“In the mosquito midgut, male gametocytes undergo gametogenesis. During this process, they replicate their DNA three times by **endomitosis** and form eight axonemes in the cytoplasm.”*

L186 spindle pole body is a terminology used in yeast to describe a different structure. Please avoid using this terminology to avoid any confusion.

We have rephrased the respective line 194-196 to:

*“However, the reconstruction demonstrated that duplication of the **centrosomes** and spindle formation has occurred in ARPC1(-) ookinetes”*

Fig 3 despite the figure legend, I do not think that the figure shows a role of ARPC1 in kinetochore-spindle attachment.

This is of course true and an oversight from our side as a leftover of an older version of that figure. We have removed that part of the figure legend and it now reads just *“ARPC1 constitutes part of an atypical Arp2/3 complex”*.

Response to Reviewers

Reviewer #1:

Remarks to the Author:

I have read the extensive rebuttal, additional data and changes to the manuscript. I am satisfied that the authors have address my specific comments as well as those of other reviewers. The paper is clearly the beginning of a lot more work to come, which I look forward to seeing in time. I have no further requested changes/comments.

We thank the reviewer for his positive feedback.

Reviewer #2:

Remarks to the Author:

The style of the paper is much better, it's much clearer, the alphafold section is neat, but the authors have not quite got the message. In the abstract they cannot say "Here we discovered an atypical five-subunit Arp2/3 complex in Plasmodium", because they haven't proven it has five subunits, they just only found 5. They also can't say "Unlike the canonical seven-subunit Arp2/3 complex found in other eukaryotes..." because they haven't proven it isn't seven.

As far as I can see on rereading, it's only the abstract that is bad. Perhaps the authors changed the body text and forgot the abstract, but please go one more time through the manuscript and remove any other statements that appear to state they have discovered a 5-member complex (or purify it to visibility by Coomassie, which is fairly easy, but a lot of work for this...).

They could also add a couple of sentences discussing this point in the discussion; it's interesting. I do think there's a good chance the complex is 5-membered, but also a good chance that two subunits diverged for some reason...

Thank you for your overall positive feedback. We have adjusted the abstract to remove all statements that suggest that we have discovered a five-member complex. We have also specified a statement in the discussion (line 356) which now reads:

*"Intriguingly, Plasmodium ARPC4 contains a long, N-terminal domain that does not align to any Arp2/3 subunit. This extension may compensate for the **possibly** missing Arp2/3 subunits ARPC3 or ARPC5,..."*

We would like to point out that we discuss the number of subunits of the atypical Plasmodium Arp2/3 complex in the discussion (line 343-348), reading *"While the canonical Arp2/3 complex consists of seven subunits, we have so far only identified five subunits. We did not find evidence for putative Plasmodium ARPC3 and ARPC5 subunit orthologs in any of our three pulldowns or by structure-based search. Their existence cannot be excluded, as they may exhibit weaker interactions with the complex and thus be undetectable in our experimental conditions and/or be too divergent for identification by structure-based search"*

Reviewer #3:

Remarks to the Author:

I believe the authors have successfully addressed most of the issues raised by the reviewers and have significantly improved their manuscript.

We thank the reviewer for his positive feedback.